# Contrasting thinning patterns between lake- and land-terminating glaciers in the Bhutan Himalaya

Shun Tsutaki[1,a], Koji Fujita[1], Takayuki Nuimura[1,b], Akiko Sakai[1], Shin Sugiyama[2], Jiro Komori[1,3,c], and Phuntsho Tshering[1,3,d]

[1]Graduate School of Environmental Studies, Nagoya University, Nagoya, Japan

[2]Institute of Low Temperature Science, Hokkaido University, Sapporo, Japan

[3]Department of Geology and Mines, Ministry of Economic Affairs, Thimphu, Bhutan

[a]now at: Atmosphere and Ocean Research Institute, The University of Tokyo, Kashiwa, Japan

[b]now at: Chiba Institute of Science, Choshi, Japan

[c]now at: Department of Modern Life, Teikyo Heisei University, Tokyo, Japan

[d]now at: Cryosphere Services Division, National Center for Hydrology and Meteorology, Thimphu, Bhutan

*Correspondence to*: Shun Tsutaki (tsutshun@frontier.hokudai.ac.jp) and Koji Fujita (cozy@nagoya-u.jp)

**Abstract.** Despite the importance of glacial lake development in ice dynamics and glacier thinning, in situ and satellite-based measurements from lake-terminating glaciers are sparse in the Bhutan Himalaya, where a number of proglacial lakes exist. We acquired in situ and satellite-based observations across a lake- and a land-terminating debris-covered glacier in the Lunana region, Bhutan Himalaya. A repeated differential global positioning system survey reveals that thinning of the debris-covered ablation area of the lake-terminating Lugge Glacier (4.67 ± 0.07 m a$^{-1}$) is more than three times greater than that of the land-terminating Thorthormi Glacier (1.40 ± 0.07 m a$^{-1}$) for the 2004–2011 period. The surface flow velocities decrease down-glacier along Thorthormi Glacier, whereas they increase from the upper part of the ablation area to the terminus of Lugge Glacier. Numerical experiments using a two-dimensional ice flow model demonstrate that the rapid thinning of Lugge Glacier would be driven by both negative surface mass balance and dynamically induced ice thinning. However, the thinning of Thorthormi Glacier is suppressed by a longitudinally compressive flow regime. The magnitude of dynamic thickening compensates for approximately one-thirds of the negative surface mass balance of Thorthormi Glacier. Multiple supraglacial ponds on Thorthormi Glacier have been expanding since 2000 and merged into a single proglacial lake, with the glacier

terminus detaching from its terminal moraine in 2011. Numerical experiments suggest that the thinning of Thorthormi Glacier will be accelerated with continued proglacial lake development.

## 1 Introduction

The spatially heterogeneous shrinkage of Himalayan glaciers has been revealed by in situ measurements (Yao et al., 2012; Azam et al., 2018), satellite-based observations (Bolch et al., 2012; Kääb et al., 2012; Brun et al., 2017), mass balance and climate models (Fujita and Nuimura, 2011; Mölg et al., 2014), and a compilation of multiple methods (Cogley, 2016). Glaciers in Bhutan in the southeastern Himalayas have experienced significant shrinkage and thinning over the past four decades. For example, the glacier area loss in Bhutan was $13.3 \pm 0.1\%$ between 1990 and 2010, based on repeated decadal glacier inventories (Bajracharya et al., 2014). Multitemporal digital elevation models (DEMs) revealed that the glacier-wide mass balance of Bhutanese glaciers was $-0.17 \pm 0.05$ m w.e. $a^{-1}$ during 1974–2006 (Maurer et al., 2016) and $-0.22 \pm 0.12$ m w.e. $a^{-1}$ during 1999–2010 (Gardelle et al., 2013). Bhutanese glaciers are inferred to be particularly sensitive to changes in air temperature and precipitation because they are affected by monsoon-influenced, humid climate conditions (Fujita and Ageta, 2000; Fujita, 2008; Sakai and Fujita, 2017). Mass loss of Gangju La Glacier in central Bhutan was much greater than those of glaciers in the eastern Himalaya and southeastern Tibet for the recent decade (Tshering and Fujita, 2016). It is crucial to investigate the mechanisms driving the mass loss of Bhutanese glaciers to provide more insight for the glacier mass balance (Zemp et al., 2015) and improve projections of global sea level rise and glacier evolution (Huss and Hock, 2018).

In recent decades, glacial lakes have formed and expanded at the termini of retreating glaciers in the Himalayas (Ageta et al., 2000; Komori, 2008; Fujita et al., 2009; Hewitt and Liu, 2010; Sakai and Fujita, 2010; Gardelle et al., 2011; Nie et al., 2017). Proglacial lakes can be form by expansion and coalescence of suplaglacial ponds, which are formed in topographic hollows formerly occupied with ice, by being fed with both precipitation and glacial meltwater. Proglacial lakes are dammed by terminal and lateral moraines, or stagnant ice masses at the glacial front (Sakai, 2012; Carrivick and Tweed, 2013). The formation and expansion of proglacial lakes accelerates glacier retreat through flotation of the terminus, increased calving, and ice flow (e.g., Funk and Röthlisberger, 1989; Warren and Kirkbride, 2003; Tsutaki et al., 2013). The ice thinning rates of lake-terminating glaciers are generally greater than those of neighbouring land-terminating glaciers in the Nepal and Bhutan Himalayas (Nuimura et al., 2012; Gardelle

et al., 2013; Maurer et al., 2016; King et al., 2017). Increases in ice discharge and surface flow velocity at the glacier terminus cause rapid thinning due to longitudinal stretching, known as dynamic thinning. For example, dynamic thinning accounted for 17 % of the total ice thinning at lake-terminating Yakutat Glacier, Alaska, during 2007–2010 (Trüssel et al., 2013). Therefore, it is important to quantify the contributions of dynamic thinning and surface mass balance (SMB) to evaluate ongoing mass loss and predict the future evolution of lake-terminating glaciers in Bhutan.

To investigate the contribution of dynamically induced changes in ice thickness to glacier thinning, it is beneficial to compute the ice flow velocity field of a lake-terminating glacier using an ice flow model. Two-dimensional ice flow models have been utilised to investigate the dynamic thinning of marine-terminating outlet glaciers (Benn et al., 2007a; Vieli and Nick, 2011), which require the ice flow velocity field and glacier thickness. In Bhutan, ice flow velocity measurements have been carried out via remote sensing techniques with optical satellite images (Kääb, 2005; Bolch et al., 2012; Dehecq et al., 2015) and in situ global positioning system (GPS) surveys (Naito et al., 2012), but no ice thickness data are available. Another approach to investigate the relative importance of ice dynamics in glacier thinning is to compare lake- and land-terminating glaciers in the same region. This method has been applied to neighbouring lake- and land-terminating glaciers in Nepal and other regions (Nuimura et al., 2012; Trüssel et al., 2013; King et al., 2017).

Widespread thinning of Himalayan glaciers has been revealed by differencing multitemporal DEMs constructed from satellite image photogrammetry because the surface of debris-covered glaciers can be highly variable, making access difficult to get large amounts of data (e.g., Gardelle et al., 2013; Maurer et al., 2016; Brun et al., 2017). In particular, unmanned autonomous vehicle (UAVs) is a powerful tool to obtain higher-resolution imagery than satellite, and thus resolves the highly variable topography and thinning rates of debris-covered surface more accurately (e.g., Immerzeel et al., 2014; Vincent et al., 2016). Repeated differential GPS (DGPS) measurements, which are acquired with centimetre-scale accuracy, also enable us to evaluate elevation changes of several metres (e.g., Fujita et al., 2008). Although their temporal and spatial coverage can be limited, the repeated DGPS measurements have been successfully acquired to investigate the surface elevation changes of debris-free glaciers in Bhutan (Tshering and Fujita, 2016) and the Inner Tien Shan (Fujita et al., 2011).

This study aims to quantify the contributions of ice dynamics and SMB to the thinning of adjacent land- and lake-terminating glaciers. To investigate the importance of glacial lake formation and expansion on glacier thinning, we measured surface elevation changes on a lake- and a land-terminating glacier in the Lunana region, Bhutan

Himalaya. Following a previous report of surface elevation measurements from a DGPS survey (Fujita et al., 2008),
we repeated the DGPS survey on the lower parts of the land-terminating Thorthormi Glacier as well as the adjacent
lake-terminating Lugge Glacier. Thorthormi and Lugge glaciers were selected for analysis because they have
contrasting termini, grounding and fully contacting lake at similar elevations. These contrasting conditions at the
similar elevations make them suitable for evaluating the contribution of ice dynamics to the observed ice thickness
changes. The glaciers are also suitable for field measurements because of their relatively safe ice-surface conditions
and proximity to trekking routes. We also performed numerical simulations to evaluate the contributions of SMB
and ice dynamics to surface elevation changes.
**2 Study site**
This study focuses on two debris-covered glaciers (Thorthormi and Lugge glaciers) in the Lunana region of
northern Bhutan (Fig. 1a, 28°06' N, 90°18' E). Thorthormi Glacier covers an area of 13.16 km$^2$, based on a satellite
image from 17 January 2010 (Table S1, Nagai et al., 2016). The ice flows to the south in the upper part and to the
southwest in the terminal part of the glacier at rates of 60–100 m a$^{-1}$ (Bolch et al., 2012). The surface is almost flat
(< 1°) within 3000 m of the glacier terminus. The ablation area of the glacier thinned at a rate of 3 m a$^{-1}$ during the
2000–2010 period (Gardelle et al., 2013). Large supraglacial lakes, which are inferred to possess a high potential
for outburst flood (Fujita et al., 2008, 2013), have formed along the western and eastern lateral moraines in the
ablation area by merging multiple supraglacial ponds since the 1990s (Ageta et al., 2000; Komori, 2008). The front
of Thorthormi Glacier was still in contact with the terminal moraine during our field campaign in September 2011,
but the glacier was completely detached from the moraine in the Landsat 7 image of 2 December 2011. Thorthormi
Glacier is therefore termed as a land-terminating glacier here since the glacier terminus was grounded during the
studied period of 2004–2011.
Lugge Glacier is a lake-terminating glacier with an area of 10.93 km$^2$ in May 2010 (Table S1, Nagai et al., 2016).
The mean surface slope is 12° within 3000 m of the glacier terminus. A moraine-dammed proglacial lake has
expanded since the 1960s (Ageta et al., 2000; Komori, 2008), and the glacier terminus retreated by ~1 km during
1990–2010 (Bajracharya et al., 2014). Lugge Glacier thinned near the terminus at a rate of 8 m a$^{-1}$ during 2000–
2010 (Gardelle et al., 2013). On 7 October 1994, an outburst flood, with a volume of 17.2 × 10$^6$ m$^3$, occurred from
Lugge Glacial Lake (Fujita et al., 2008). The depth of Lugge Glacial Lake was 126 m at its deepest location, with a
mean depth of 50 m, based on a bathymetric survey in September 2002 (Yamada et al., 2004).

110       Although the debris thickness was not measured during the field campaigns, there were regions of debris-free

surface across the ablation areas of Thorthormi and Lugge glaciers (Fig. S1). Debris cover is therefore considered
to be thin and sparse across the study area. Furthermore, few supraglacial ponds and ice cliffs were observed across
the glaciers. Satellite imagery show that the surface is heavily crevassed in the lower ablation areas, suggesting that
glacier meltwater immediately drain into the interior of the glaciers.

115       Meteorological and glaciological in situ observations were acquired across the glaciers and lakes in the Lunana

region from 2002 to 2004 (Yamada et al., 2004). Naito et al. (2012) reported changes in surface elevation and ice
flow velocity along the central flowline in the lower parts of Thorthormi and Lugge glaciers for the 2002–2004
period. The ice thinning rate at Lugge Glacier was ∼5 m a$^{-1}$ during 2002–2004, which is much higher than that at
Thorthormi Glacier (0–3 m a$^{-1}$). The surface flow velocities of Thorthormi Glacier decrease down-glacier from
∼90 to ∼30 m a$^{-1}$ at 2000–3000 m from the terminus, while the surface flow velocities of Lugge Glacier are nearly
uniform at 40–55 m a$^{-1}$ within 1500 m of the terminus (Naito et al., 2012).

**3 Data and methods**

**3.1 Surface elevation change**

124       We surveyed the surface elevations in the lower parts of Thorthormi and Lugge glaciers from 19 to 22 September

2011, and then compared them with those observed from 29 September to 10 October 2004 (Fujita et al., 2008). We
used dual- and single-frequency carrier phase GPS receivers (GNSS Technologies, GEM-1, and MAGELLAN
ProMark3). One receiver was installed 2.5 km west of the terminus of Thorthormi Glacier as a reference station
(Fig. 1a), whose location was determined by an online precise point positioning processing service
(https://webapp.geod.nrcan.gc.ca/geod/tools-outils/ppp.php?locale=en, last accessed: 21 October 2018), which
provided standard deviations of < 4 mm for both the horizontal and vertical coordinates after one week of
continuous measurements in 2011. Observers walked on/around the glaciers with a GPS receiver and antenna fixed
to a frame pack. The height uncertainty of the GPS antenna during the survey was < 0.1 m (Tsutaki et al., 2016).
We neglected influence of change in debris thickness in the DGPS surveys because the debris cover across the
glaciers is sparse and thin (Fig. S1), and we therefore could walk on the ice surface across most of the surveyed
area. The DGPS data were processed with RTKLIB, an open source software for GNSS positioning
(http://www.rtklib.com/, last accessed: 21 October 2018). Coordinates were projected onto a common Universal
Transverse Mercator projection (UTM zone 46N, WGS84 reference system). We generated DEMs with 1 m
resolution by interpolating the surveyed points with an inverse distance weighted method, as used in previous
studies (e.g., Fujita and Nuimura, 2011; Tshering and Fujita, 2016). The 2004 survey data were calibrated with four
benchmarks around the glaciers (Fig. 1a) to generate a 1 m resolution DEM. Details of the 2004 and 2011 DGPS
surveys, along with their respective DEMs, are summarised in Table S1. The surface elevation changes between
2004 and 2011 were computed at points where data were available for both dates. Elevation changes were obtained
at 431 and 248 DEM grid points for Thorthormi and Lugge glaciers, respectively (Table 1).
To evaluate the spatial representativeness of the change in glacier surface elevation change derived from DGPS
measurements, we compared the elevation changes derived from the DGPS-DEMs and Advanced Spaceborne
Thermal Emission and Reflection Radiometer (ASTER) DEMs acquired on 11 October 2004 and 6 April 2011
(Table S2), respectively, which cover a similar period to our field campaigns (2004–2011). The 30 m ASTER-
DEMs were provided by the ASTER-VA (https://gbank.gsj.jp/madas/map/index.html, last accessed: 21 October
2018) and used to compute the surface elevation change. The ASTER-DEM elevations were calibrated using the
DGPS data on the off-glacier terrain in 2011. The vertical coordinates of the ASTER-DEMs were then corrected for
the corresponding bias, with the elevation change over the glacier surface computed as the difference between the
calibrated DEMs.
The horizontal uncertainty of the DGPS survey was evaluated by comparing the positions of four benchmarks
installed around Thorthormi and Lugge glaciers (Fig. 1a). Although previous studies dealing with satellite-based
DEMs have adopted standard error as vertical uncertainty ($\sigma_e$) (e.g., Berthier et al., 2007; Bolch et al., 2011;
Maurer et al., 2016), we used the standard deviation of the elevation difference on the off-glacier terrain in the
DGPS surveys because the off-glacier points in our DGPS-DEM survey is so many ($n = 3893$) and then the
standard error could be too small.
**3.2 Surface flow velocities**
We calculated surface flow velocities by processing ASTER images (15 m resolution, near infrared (NIR), near
nadir 3N band) with the COSI-Corr feature tracking software (Leprince et al., 2007), which is commonly adopted
in mountainous terrain to measure surface displacements with an accuracy of one-fourth to one-tenth of the pixel
size (e.g., Heid and Kääb, 2012; Scherler and Strecker, 2012; Lamsal et al., 2017). Orthorectification and
coregistration of the images were performed by Japan Space Systems before processing. The orthorectification and
coregistration accuracies were reported as 16.9 m and 0.05 pixel, respectively. We selected five image pairs from
seven scenes between 22 October 2002 and 12 October 2010, with temporal separations ranging from 273 to 712
days (Table S3), to obtain annual surface flow velocities of the glaciers. It should be noted that the aim of our flow
velocity measurements is to investigate the mean surface flow regime of the glaciers rather than its interannual
variability. The subpixel displacement of features on the glacier surface was recorded at every fourth pixel in the
orthorectified ASTER images, providing the horizontal flow velocities at a 60 m resolution (Scherler et al., 2011).
We used a statistical correlation mode, with a correlation window size of 16 × 16 pixels and a mask threshold of 0.9
for noise reduction (Leprince et al., 2007). The obtained ice flow velocity fields were filtered to remove residual
attitude effects and miscorrelations (Scherler et al., 2011; Scherler and Strecker, 2012). We applied two filters to
eliminate those flow vectors that deviated in magnitude (greater than $\pm 1\,\sigma$) or direction ($> 20°$) from the mean
vector within the neighbouring 21 × 21 data points.
**3.3 Glacial lake area**
We analysed the areal variations in the glacial lake area in Thorthormi and Lugge glaciers using 12 satellite
images acquired by the Landsat 7 ETM+ between November 2000 (distributed by the United States Geological
Survey, http://landsat.usgs.gov/, last accessed: 21 October 2018). We selected images taken in either November or
December with the least snow and cloud cover. We also analysed multiple ETM+ images acquired from the
October to December timeframe of each year to avoid the scan line corrector-off gaps. Glacial lakes were manually
delineated on false colour composite images (bands 3–5, 30 m spatial resolution). Following previously proposed
delineation methods (e.g., Bajracharya et al., 2014; Nuimura et al., 2015; Nagai et al., 2016), marginal ponds in
contact with bedrock/moraine ridge were included in the glacial lake, whereas small supraglacial ponds surrounded
by ice were excluded. The accuracy of the outline mapping is equivalent to the image resolution (30 m). The
coregistration error in the repeated images was ±30 m, based on visual inspection of the horizontal shift of a stable
bedrock and lateral moraines on the coregistered imagery. The user-induced error was estimated to be 5% of the
lake area delineated from the Landsat images (Paul et al., 2013). The total error of the area analysis was less than
±0.14 and ±0.08 km$^2$ for Thorthormi and Lugge glaciers, respectively.

## 3.4 Mass balance of the debris-covered surface

SMB is an essential component of ice thickness change, but no in situ SMB data are available in the Lunana region. Therefore, the spatial distributions of the SMB on the debris-covered Thorthormi and Lugge glaciers were computed with a heat and mass balance model, which quantifies the spatial distribution of the mean SMB for each glacier.

Thin debris accelerates ice melt by lowering surface albedo, while thick debris (generally more than ~5 cm) suppresses ice melt and acts as an insulating layer (Østrem, 1959; Mattson et al., 1993). To obtain the spatial distributions of debris thickness and SMB, we estimated the thermal resistance from remotely sensed data and reanalysis climate data (Suzuki et al., 2007a; Zhang et al., 2011; Fujita and Sakai, 2014). The thermal resistance ($R_T$, m$^2$ K W$^{-1}$) is defined as follows:

$$R_T = \frac{h}{\lambda} \tag{1}$$

where $h$ and $\lambda$ are debris thickness (m) and thermal conductivity (W m$^{-1}$ K$^{-1}$), respectively. This method has been applied to reproduce debris thickness and SMB in southeastern Tibet (Zhang et al., 2011) and glacier runoff in the Nepal Himalaya (Fujita and Sakai, 2014). Assuming no heat storage, a linear temperature profile within the debris layer, and the melting point temperature at the ice–debris interface ($T_i$, 0 °C), the conductive heat flux through the debris layer ($G_d$, W m$^{-2}$) and the heat balance at the debris surface are described as follows:

$$G_d = \frac{(T_s - T_i)}{R_T} = (1 - \alpha_d)R_{Sd} + R_{Ld} - R_{Lu} + H_S + H_L \tag{2}$$

where $\alpha_d$ is the debris surface albedo; $R_{Sd}$, $R_{Ld}$, and $R_{Lu}$ are the downward short wave radiation, and downward and upward long wave radiation, respectively (positive sign, W m$^{-2}$); and $H_S$ and $H_L$ are the sensible and latent heat fluxes (W m$^{-2}$), respectively, which are positive when the fluxes are directed toward the ground. Both turbulent fluxes were ignored in the original method to obtain the thermal resistance based on a sensitivity analysis and field measurements (Suzuki et al., 2007a). However, we improved the method by taking the sensible heat into account because several studies have indicated that ignoring the sensible heat can result in an underestimation of the

thermal resistance (e.g., Reid and Brock, 2010). Using eight ASTER images (90 m resolution, Level 3A1 data)
obtained between October 2002 and October 2010 (Table S4), along with the NCEP/NCAR reanalysis climate data
(NCEP-2, Kanamitsu et al., 2002), we calculated the distribution of mean thermal resistance on the two target
glaciers. Surface albedo is calculated using three-visible near-infrared sensors (VNIR; bands 1–3), and surface
temperature is obtained from an average of five sensors in the thermal infrared (TIR; bands 10–14). Automatic
weather station (AWS) observations from the terminal moraine of Lugge Glacial Lake (4524 m a.s.l., Fig. 1a)
showed that the annual mean air temperature during 2002–2004 was ~0 °C, and annual precipitation was 900 mm
in 2003 (Suzuki et al., 2007b). The air temperature at the AWS elevation was estimated using the pressure level
atmospheric temperature and geopotential height (Sakai et al., 2015), and then modified for each 90 × 90 m mesh
grid points using a single temperature lapse rate (0.006 °C km$^{-1}$). The wind speed was assumed to be 2.0 m d$^{-1}$,
which is the two-years average of the 2002–2004 AWS record (Suzuki et al., 2007b). The uncertainties in the
thermal resistance and albedo were evaluated as 107 and 40%, respectively, by taking the standard deviations
calculated from multiple images at the same location (Fig. S2).
The SMB of the debris-covered ablation area was calculated by a heat and mass balance model that included
debris-covered effects (Fujita and Sakai, 2014). First, the surface temperature is determined to satisfy Eq. (2) using
the estimated thermal resistance and an iterative calculation, and then, if the heat flux toward the ice–debris
interface is positive, the daily amount of ice melt beneath the debris mantle ($M_d$, kg m$^{-2}$ d$^{-1}$) is obtained as follows:

$$M_d = \frac{t_D G_d}{l_m} \tag{3}$$


where $t_D$ is the length of a day in seconds (86400 s), and $l_m$ is the latent heat of fusion of ice (3.33 × 105 J kg$^{-1}$).
Annual mass balance of debris-covered part ($b$, m w.e. a$^{-1}$) is expressed as:

$$b = \sum_{D=1}^{365} \left( P_s + P_r + \frac{t_D H_L}{l_m}_{\ for\ debirs} + \frac{t_D H_L}{l_m}_{\ for\ snow} - D_d - D_s \right) / 1000 \tag{4}$$


here $P_s$ and $P_r$ are snow and rain, respectively, which are distinguished from precipitation depending on air
temperature. Evaporation from debris and snow surfaces is expressed in the same formula but they are calculated in
different schemes because temperature and saturation conditions of the debris and snow surfaces are different. $D_d$
and $D_s$ are the daily discharge from the debris and snow surfaces, respectively. Discharge and evaporation from the
snow surface was calculated only when snow layer was formed on the debris. Because snow layer does not exist at
the end of melting season in the current climate condition and at the elevation of debris-covered area, snow
accumulation ($P_s$) is compensated with evaporation and discharge from snow surface during a calculation year.
Discharge from debris ($D_d$) is expressed as:

$$D_d = M_d + P_r + \frac{t_D H_L}{l_m}_{\ for\ debirs} \tag{5}$$


and then the mass balance can be simplified as:

$$b = -\sum_{D=1}^{365} M_d / 1000 \tag{6}$$


This implies that the mass balance of debris covered area is equivalent to the ice melting under the debris. Further
details on the equations and methodology used in the model are described by Fujita and Sakai (2014). The mass
balance was calculated at $90 \times 90$ m mesh grid points on the ablation area of the two glaciers using 38 years of
ERA-Interim reanalysis data (1979–2017, Dee et al., 2011), with the results given in metres of water equivalent
(w.e.). The meteorological variables in the ERA-Interim reanalysis data (2002–2004) were calibrated with in situ
meteorological data (2002–2004) from the terminal moraine of Lugge Glacier (Fig. S3). The ERA-Interim wind
speed was simply multiplied by 1.3 to obtain the same average as in the observational data. The SMBs calculated
with the observed and calibrated ERA-Interim data for 2002–2004 were compared with those from the entire 38-
year ERA-Interim data set. The SMBs for 2002–2004 (from both the observational and ERA-Interim data sets)
show no clear anomaly against the long-term mean SMB (1979–2017) (Fig. S4).
The sensitivity of the simulated meltwater was evaluated against the meteorological parameters used in the SMB
model. We chose meltwater instead of SMB to quantify the uncertainty because the SMB uncertainty cannot be
evaluated as absolute value. The tested parameters are surface albedo, air temperature, precipitation, relative
humidity, solar radiation, thermal resistance and wind speed. The thermal resistance and albedo uncertainties were

based on the standard deviations derived from the eight ASTER images used to estimate these parameters (Fig. S2). Each meteorological variable uncertainty, with the exceptions of the thermal resistance and albedo uncertainties, was assumed to be the root mean square error (RMSE) of the ERA-Interim reanalysis data against the observational data (Fig. S3). The simulated meltwater uncertainty was estimated as the variation in meltwater within a possible parameter range via a quadratic sum of the results from each meteorological parameter.

## 3.5 Ice dynamics

### 3.5.1 Model descriptions

To investigate the dynamically induced ice thickness change, numerical experiments were carried out by applying a two- dimensional ice flow model to the longitudinal cross sections of Thorthormi and Lugge glaciers. The aim of the experiments was to investigate whether the ice thickness changes observed at the glaciers were affected by the presence of proglacial lakes.

The model was developed for a land-terminating glacier (Sugiyama et al., 2003, 2014), and is applied to the lake-terminating glacier in this study. Taking the $x$ and $z$ coordinates in the along flow and vertical directions, the momentum and mass conservation equations in the $x$–$z$ plane are:

$$\frac{\partial \sigma_{xx}}{\partial x} + \frac{\partial \sigma_{xz}}{\partial z} = 0 \tag{7}$$

$$\frac{\partial \sigma_{zx}}{\partial x} + \frac{\partial \sigma_{zz}}{\partial z} = \rho_i g \tag{8}$$

$$\frac{\partial u_x}{\partial x} + \frac{\partial u_z}{\partial z} = 0 \tag{9}$$

where $\sigma_{ij}$ ($i,j = x,z$) are components of the Cauchy stress tensor, $\rho_i$ is the density of ice (910 kg m$^{-3}$), $g$ is the gravitational acceleration vector (9.81 m s$^{-2}$), and $u_x$ and $u_z$ are the horizontal and vertical components of the flow velocity vector, respectively. The stress in Eqs. (8) and (9) is linked to the strain rate via the constitutive equation given by Glen's flow law (Glen, 1955):


$$\dot{\varepsilon}_{ij} = A\tau_e^{n-1}\tau_{ij} \tag{10}$$


where $\dot{\varepsilon}_{ij}$ and $\tau_{ij}$ are the components of the strain rate and deviatoric stress tensors, respectively, and $\tau_e$ is the
effective stress, which is described as

$$\tau_e = \frac{1}{2}(\tau_{xx}^2 + \tau_{zz}^2) + \tau_{xz}^2 \tag{11}$$


The rate factor ($A$) and flow law exponent ($n$) are material parameters. We used the commonly accepted value of
$n = 3$ for the flow law exponent and employed a rate factor of $A = 75 \text{ MPa}^{-3}\text{a}^{-1}$, which was previously used to
model a temperate valley glacier (Gudmundsson, 1999). We assumed the glaciers were temperate because there
was no available information on the thermal states of the studied glaciers.
Model domain was within 5100 m and 3500 m from the termini of Thorthormi and Lugge glaciers, respectively
(white lines in Fig. 1b), including the ablation area and the lower accumulation area. We only interpret results in the
ablation area (0–4200 and 700–2500 m from the termini of Thorthormi and Lugge glaciers, respectively), where
surface flow velocity was obtained from ASTER imagery. The lower accumulation area was included in the
domain to supply ice into the studied area, thus excluded from analysis presentation of the results. The surface
elevation of the model domain ranges from 4442 to 4813 m for Thorthormi Glacier, and from 4530 to 5244 m for
Lugge Glacier. The surface geometry was obtained from the 90-m-grid ASTER GDEM version 2 obtained in
January 2001 after filtering the elevations with a smoothing routine at a bandwidth of 1000 m. The ice thickness
distribution was estimated from a method proposed for alpine glaciers (Farinotti et al., 2009). We applied the same
local regression filter to smooth the estimated bedrock geometry. The bedrock elevation of Thorthormi Glacier was
estimated from bathymetry data acquired in September 2011 at 1400 m from the terminus. For Lugge Glacier, the
bed elevation at the glacier front was estimated from the bathymetric map of Lugge Glacial Lake, surveyed in
September 2002 (Yamada et al., 2004). To solve Eqs. (8) and (9) for $u_x$ and $u_z$, the modelled domain was
discretised with a finite element mesh. The mesh resolution was 100 m in the horizontal direction, and several
metres near the bed and 10–28 m near the surface in the vertical direction. The total numbers of elements were 612
and 420 for Thorthormi and Lugge glaciers, respectively.
The upper surface of the domain was assumed to be stress free, through which the ice flux was prescribed from
the surface velocity obtained by the satellite analysis. We assume no basal sliding and quadratic function (4th
order) for the velocity profile from the surface to the bed. The basal sliding velocity ($u_b$) was given as a linear
function of the basal shear traction ($\tau_{xz,b}$):

$$u_b = C\tau_{xz,b} \tag{12}$$


where $C$ is the sliding coefficient. We used constant sliding coefficients of $C = 766$ and $125$ m a$^{-1}$ MPa$^{-1}$ over the
entire domains of Thorthormi and Lugge glaciers, respectively. These parameters were obtained by minimising the
RMSE between the modelled and measured surface flow velocities over the entire model domains (Fig. S5).
**3.5.2 Experimental configurations**
To quantify the effect of glacier dynamics on ice thickness change, we performed two experiments for Thorthormi
and Lugge glaciers. Experiment 1 was performed to compute the ice flow velocity fields under the present terminus
conditions. In this experiment, Thorthormi Glacier was treated as a land-terminating glacier by prescribing zero
horizontal velocity at the glacier front, whereas Lugge Glacier was treated as a lake-terminating glacier by applying
hydrostatic pressure at the front as a function of water depth. A stress-free boundary condition was given to the
calving front above the lake level. We employed glacier surface elevation in 2001 and water level of supraglacial
ponds and proglacial lake observed in 2004 as boundary conditions (Fujita et al., 2008).
Experiment 2 was designed to investigate the influence of proglacial lakes on glacier dynamics. For Thorthormi
Glacier, we simulated a calving front with thickness of 106 m. Position of the hypothetical calving front was
determined at the place where only one lake depth was acquired from bathymetry survey in September 2011. The
surface level of the proglacial lake was assumed to be 4432 m a.s.l., which is the mean surface level of the
supraglacial ponds measured in September 2004 (Fujita et al., 2008). Hydrostatic pressure and stress-free
conditions were applied to the lower boundary below and above the lake level, respectively. For Lugge Glacier, we
simulated a lake-free situation, with ice flowing to the contemporary terminal moraine, so that the glacier
terminates on land. Bedrock topography is derived from the bathymetric map (white lines in Fig. 1b, Yamada et al.,
2004). The surface topography is linearly extrapolated from the surface elevations at the calving front in 2002, and
zero flow velocity was assumed at the terminus. In the experiment, we used 444 and 684 elements for Thorthormi
and Lugge glaciers, respectively.
**3.6 Simulated ice thickness change**
To compare the influence of ice dynamics on glacier thinning in the lake- and land-terminating glaciers, we
calculated the emergence velocity ($v_e$) as follows:

$$v_e = v_z - v_h \tan \alpha \qquad (13)$$


where $v_z$ and $v_h$ are the vertical and horizontal flow velocities, respectively, and $\alpha$ is the surface slope (Cuffey and
Paterson, 2010). The surface slope $\alpha$ was obtained every 100 m from the surface topography of the ice flow model.
The surface elevation change over time ($dh/dt$, m a$^{-1}$), which is caused by the imbalance of the emergence
velocity and ice equivalent SMB ($b_{ie}$) along the central flowline, is calculated as:

$$\frac{dh}{dt} = b_{ie} + v_e \qquad (14)$$


The $b_{ie}$ was converted from SMB ($b$, m w.e. a$^{-1}$) using densities of ice (910 kg m$^{-3}$) and water (1000 kg m$^{-3}$), for
comparison with the emergence velocity. The magnitude of the emergence velocity is approximately proportional
to the horizontal flow velocity (Truffer et al., 2009). Assuming this relationship, the emergence velocity uncertainty
($\sigma_{ve}$) was estimated as:

$$\sigma_{ve} = \frac{v_e}{u_{model}} \times \sigma_{u\_model} \qquad (15)$$


where $u_{model}$ is the simulated horizontal flow velocity and $\sigma_{u\_model}$ is the mean uncertainty of the simulated
surface flow velocity, which is estimated by quadratic sum of accuracy of velocity measurements, interannual
variability in measured surface velocity over the period of 2002–2010, and RMSE between modelled and measured
surface velocities. Uncertainty in the rate of simulated ice thickness change was estimated from the sum of
uncertainties in the simulated ice-equivalent SMB and emergence velocity ($\sigma_{ve}$).
**4 Results**
**4.1 Surface elevation change**
Figure 1a shows the rate of surface elevation change ($dh/dt$, hereafter) of Thorthormi and Lugge glaciers from
2004 to 2011 derived from DGPS-DEMs. The rates for Thorthormi Glacier range from −3.37 to +1.14 m a$^{-1}$, with a
mean rate of −1.40 m a$^{-1}$ (Table 1). These rates show large variability within the limited elevation band (4410–
4450 m a.s.l., Fig. 2b). No clear trend is observed at 1000–3000 m from the terminus (Fig. 2c). The rates for Lugge
Glacier range from −9.13 to −1.30 m a$^{-1}$, with a mean rate of −4.67 m a$^{-1}$ (Table 1). The most negative values (−9
m a$^{-1}$) are found in the lower elevation band (4560 m a.s.l., Fig. 2b), which corresponds to 1300 m from the 2002
terminus position (Fig. 2c). The RMSE between the surveyed positions (five measurements in total, with one or two
measurements for each benchmark) is 0.21 m in the horizontal direction. The mean elevation difference between
the 2004 and 2011 DGPS-DEMs is 0.48 m with a standard deviation of 1.91 m (Fig. 2a), which results in the
vertical uncertainty of 0.27 m a$^{-1}$. Vertical uncertainties of ASTER-DEMs in the same manner are estimated to be
2.75 m a$^{-1}$ from the standard deviations of 2004 (15.73 m) and 2011 (8.43 m) DEMs (Fig. S6). Given the ASTER-
DEM uncertainties, the DGPS-DEMs and ASTER-DEMs yield a similar $dh/dt$ that falls within the uncertainty
range in scatter plots (Fig. S7) and in elevation distribution (Fig. S8), thus supporting the applicability of the DGPS
measurements to the entire ablation area.
**4.2 Surface flow velocities**
Figure 1b shows the surface flow velocity field from 30 January 2007 to 1 January 2008 (337 days). On Thorthormi
Glacier, the flow velocities decrease down-glacier, ranging from ~110 m a$^{-1}$ at the foot of the icefall to < 10 m a$^{-1}$
at the terminus (Fig. 3a). The flow velocities of Lugge Glacier increase down-glacier, ranging from 20–60 to 50–80
m a$^{-1}$ within 2000 m of the calving front (Fig. 3b). The flow velocity uncertainty was estimated to be 12.1 m a$^{-1}$, as
given by the mean off-glacier displacement from 3 February 2006 to 30 January 2007 (362 days) (Fig. S9).

### 4.3 Changes in glacial lake area

The supraglacial pond area near the front of Thorthormi Glacier progressively increased from 2000 to 2017, at a mean rate of 0.09 km$^2$ a$^{-1}$ while Lugge Glacial Lake also expanded from 2000 to 2017, at a mean rate of 0.03 km$^2$ a$^{-1}$ (Fig. 4). The total area changes from 2000 to 2017 are 1.79 km$^2$ and 0.46 km$^2$ for Thorthormi and Lugge glaciers, respectively.

### 4.4 Mass balance of the debris-covered surface

The simulated SMBs over the ablation area were $-7.36 \pm 0.12$ m w.e. a$^{-1}$ for Thorthormi Glacier and $-5.25 \pm 0.13$ m w.e. a$^{-1}$ for Lugge Glacier (Fig. 1c, Table 1). The debris-free surface has a more negative SMB than the debris-covered regions of the glaciers. The mean SMBs of the debris-free and debris-covered surfaces in the ablation area of Thorthormi Glacier are $-9.31 \pm 0.68$ and $-7.30 \pm 0.13$ m w.e. a$^{-1}$, respectively, while those of Lugge Glacier are $-7.33 \pm 0.41$ m w.e. a$^{-1}$ and $-5.41 \pm 0.18$ m w.e. a$^{-1}$, respectively (Table 1). The sensitivity of simulated meltwater in the SMB model was evaluated as a function of the RMSE of each meteorological variable across the debris-covered area (Fig. S10). Ice melting is more sensitive to solar radiation and thermal resistance. The influence of thermal resistance on meltwater formation is considered to be small since the debris cover is sparse over the glaciers. The estimated meltwater uncertainty is < 50% across most of Thorthormi and Lugge glaciers (Fig. S11).

### 4.5 Numerical experiments of ice dynamics

The ice thinning of Lugge Glacier was three times faster than that of Thorthormi Glacier. However, the mean SMB was 1.4 times more negative at Thorthormi Glacier, suggesting a substantial influence of glacier dynamics on ice thickness change. To quantify the contribution of ice dynamics to the ice thickness change, we performed numerical experiments with the present (Experiment 1) and prescribed (Experiment 2) glacier geometries.

### 4.5.1 Experiment 1 – present terminus conditions

Modelled results for the present geometry show significantly different flow velocity fields for Thorthormi and Lugge glaciers (Figs. 5c and 5d). Thorthormi Glacier flows faster (> 150 m a$^{-1}$) in the upper reaches, where the surface is steeper than the other regions (Fig. 5c). Down-glacier of the icefall, where the glacier surface is flatter,

the ice motion slows in the down-glacier direction, with the flow velocities decreasing to < 10 m a$^{-1}$ near the
terminus (Fig. 5e). Ice flows upward relative to the surface across most of the modelled region (Fig. 5c). In contrast
to the down-glacier decrease in the flow velocities at Thorthormi, the computed velocities of Lugge Glacier are up
to ~40 m a$^{-1}$ within 500–2200 m of the terminus, and it sharply increases to ~80 m a$^{-1}$ at the calving front (Fig. 5f).
Ice flow is nearly parallel to the glacier surface, except for the more downward motion near the calving front (Fig.
5d). Within 3000 m of the terminus of Thorthormi Glacier, the modelled surface flow velocities are in good
agreement with the satellite-derived flow velocities (Fig. 5e). The calculated surface flow velocities of Lugge
Glacier agreed with the satellite-derived flow velocities within 16 % (Fig. 5f).
**4.5.2 Experiment 2 – reversed terminus conditions**
Figure 6c shows the flow velocities simulated for the lake-terminating boundary condition of Thorthormi Glacier,
in which the flow velocities within 200 m of the calving front are three to four times faster than those of
Experiment 1 (Figs. 5c and 6c). The mean vertical surface flow velocity within 2000 m of the front is still positive
(0.9 m a$^{-1}$), but is smaller than that for the land-terminating condition (1.6 m a$^{-1}$). The modelled result demonstrates
significant acceleration as the glacier dynamics change from a compressive to stretching flow regime after
proglacial lake formation. For Lugge Glacier, the flow velocities decrease over the entire glacier in comparison
with Experiment 1 (Figs. 5d and 6d). The upward ice motion appears within 3000 m of the terminus. The numerical
experiments demonstrate that the formation of a proglacial lake causes significant changes in ice dynamics.
**4.5.3 Simulated surface flow velocity uncertainty**
Basal sliding accounts for 97 % and 75 % of the simulated ice flow velocity in the ablation area of Thorthormi
and Lugge glaciers, respectively (Figs. 5e and 5f), suggesting that ice deformation is insufficient to represent the ice
flow regardless of assumption of ice temperature. Standard deviations of ASTER-derived surface velocities are 2.9
and 6.7 m a$^{-1}$ for Thorthormi and Lugge glaciers, respectively, which are considered as interannual variability in
the measured surface velocities (Fig. 3). The RMSEs between the modelled and measured flow velocities were
computed as a measure of the model performance. For Thorthormi Glacier, the model exhibits similar sensitivities
to the sliding coefficient and ice thickness while the model is more sensitive to the ice thickness than the sliding
coefficient for Lugge Glacier (Fig. S5). Uncertainty of the emergence velocity is affected by factors such as
accuracy of flow velocity measurement, interannual variability, and RMSE between modelled and measured flow
velocity so that we performed sensitivity test by changing ±30 % of ice thickness and sliding coefficient. It shows
that the simulated surface flow velocities of Thorthormi Glacier vary by ±30 % when the constant sliding
coefficient ($C$) and ice thickness are varied by ±30 % (Fig. S12). For Lugge Glacier, the simulated flow velocities
vary by 22 and 65 % when the sliding coefficient and ice thickness are varied by ±30 %, respectively. The mean
uncertainty of the simulated surface flow velocity ($\sigma_{u\_model}$) is 20.7 and 26.9 m a$^{-1}$ for Thorthormi and Lugge
glaciers, respectively. Finally the mean uncertainty is estimated to be 0.2 m a$^{-1}$ for both glaciers.

**4.6 Simulated ice thickness change**

Figure 7a shows the computed emergence velocity and SMB along the central flowlines of the glaciers. Given
the computed surface flow velocities from Experiment 1, the emergence velocity of Thorthormi Glacier was 3.21 ±
0.21 m a$^{-1}$ within 4200 m of the terminus that increases to > 10 m a$^{-1}$ in the upper reaches of the glacier (Fig. 7a).
In contrast, the emergence velocity of Lugge Glacier was −1.69 ± 0.18 m a$^{-1}$ within 700–2500 m of the terminus
(Fig. 7a) and greatly negative value at the calving front (−10 m a$^{-1}$) due to the increase in surface flow velocities
toward the glacier front (Fig. 5f). Under the Experiment 1 conditions, the estimated $dh/dt$ are −4.88 ± 0.34 m a$^{-1}$
within 4200 m of Thorthormi Glacier and −7.46 ± 0.32 m a$^{-1}$ within 700–2500 m from the calving front of Lugge
Glacier (Fig. 7).
The emergence velocity computed under contrasting geometries (Experiment 2) varies from that with the present
geometries (Experiment 1) for both Thorthormi and Lugge glaciers. For the lake-terminating condition of
Thorthormi Glacier, the mean emergence velocity turns into negative (−1.37 ± 0.23 m a$^{-1}$) within 3700 m of the
terminus. The mean emergence velocity of Lugge Glacier computed with the land-terminating condition is less
negative (−0.78 ± 0.28 m a$^{-1}$) within 700–2500 m of the terminus. Given the same SMB distribution, the mean
$dh/dt$ is computed as −9.46 ± 0.36 m a$^{-1}$ for Thorthormi Glacier with the lake-terminating condition and −6.55 ±
0.42 m a$^{-1}$ for the land-terminating Lugge Glacier (Table 1).

## 5 Discussion

### 5.1 Glacier thinning

The repeated DGPS surveys revealed rapid thinning of the ablation area of Lugge Glacier between 2004 and 2011. The mean $dh/dt$ ($-4.67 \pm 0.27$ m a$^{-1}$) is comparable to that for the 2002–2004 period ($-5$ m a$^{-1}$, Naito et al., 2012) while it is more than twice as negative as that derived from ASTER-DEMs for the 2004–2011 period ($-2.24 \pm 2.75$ m a$^{-1}$). The results suggest that Lugge Glacier is thinning more rapidly than neighbouring glaciers in the Nepal and Bhutan Himalayas. The mean $dh/dt$ was $-0.50 \pm 0.14$ m a$^{-1}$ in the ablation area of Bhutanese glaciers for the period 2000–2010 (Gardelle et al., 2013), and $-2.30 \pm 0.53$ m a$^{-1}$ for debris-free glaciers in eastern Nepal and Bhutan during 2003–2009 (Kääb et al., 2012). Maurer et al. (2016) reported that $dh/dt$ for Lugge Glacier during 1974–2006 ($-0.6 \pm 0.2$ m a$^{-1}$) was greater than those for other Bhutanese lake-terminating glaciers ($-0.2$ to $-0.4$ m a$^{-1}$). The $dh/dt$ of Thorthormi Glacier derived from DGPS-DEMs ($-1.40 \pm 0.27$ m a$^{-1}$) and ASTER-DEMs ($-1.61 \pm 2.75$ m a$^{-1}$) from 2004 to 2011 are comparable with previous measurements, which range from $-3$ to 0 m a$^{-1}$ for the period 2002–2004 (Naito et al., 2012). The mean rate across Thorthormi Glacier was $-0.3 \pm 0.2$ m a$^{-1}$ during 1974–2006 (Maurer et al., 2016), which is a typical rate in the Bhutan Himalaya.

Lugge Glacier is thinning more rapidly than Thorthormi Glacier, which is consistent with previous satellite-based studies. For example, the $dh/dt$ of the lake-terminating Imja and Lumding glaciers ($-1.14$ and $-3.41$ m a$^{-1}$, respectively) were ~4 times greater than those of the land-terminating glaciers (approximately $-0.87$ m a$^{-1}$) in the Khumbu region of the Nepal Himalaya (Nuimura et al., 2012). King et al. (2017) measured the $dh/dt$ of the lower parts of nine lake-terminating glaciers in the Everest area (approximately $-2.5$ m a$^{-1}$), which was 67% more negative than that of 18 land-terminating glaciers (approximately $-1.5$ m a$^{-1}$). The $dh/dt$ of lake-terminating glaciers in Yakutat ice field, Alaska ($-4.76$ m a$^{-1}$) was ~30% more negative than the neighbouring land-terminating glaciers (Trüssel et al., 2013). It should be noted that the difference in $dh/dt$ between Lugge and Thorthormi glaciers derived from DGPS-DEMs (3.3 times) is much greater than the numbers previously reported in the Nepal Himalaya, suggesting that ice dynamics play a more significant role here.

## 5.2 Influence of ice dynamics on glacier thinning

The estimated $dh/dt$ are 3.5 time more negative than the DPGS observation for Thorthormi Glacier and 60 % more negative than the DGPS observation for Lugge Glacier, respectively (Table 1). However, differences in $dh/dt$ between the glaciers are similar (Lugge < Thorthormi by 3.27 m a$^{-1}$ in the observation and 2.58 m a$^{-1}$ in the Experiment 1). Although both SMB and emergence velocity could have large uncertainties, the discrepancy between observation and estimation would be resulted from an overestimation of ice melting in the SMB model, and the contrasting emergence velocity among the glaciers would be plausible. The mean SMB of Thorthormi Glacier is 40 % more negative than that of Lugge Glacier. Since debris cover looks sparse across the ablation area of both glaciers (Fig. S1), the more negative SMB of Thorthormi Glacier could be explained by the glacier situated at lower elevations (Fig. 2b). The calculated SMBs (Thorthomi < Lugge) and observed $dh/dt$ (Lugge < Thorthormi) suggest that contribution of glacier dynamics is substantially different in the two glaciers. The horizontal flow velocities of Lugge Glacier are nearly uniform along the central flowline, with ice flow parallel to the glacier surface (Fig. 5d), suggesting that the dynamically induced ice thickness change is small. The computed emergence velocity is negative (−1.69 ± 0.18 m a$^{-1}$), which means the ice dynamics rather accelerates glacier thinning. In contrast to Lugge, the flow velocities of Thorthormi Glacier decrease toward the terminus (Fig. 5c), resulting in thickening under a longitudinally compressive flow regime. The emergence velocity of Thorthormi Glacier is positive (+3.21 ± 0.21 m a$^{-1}$), indicating a vertically extending strain regime. The calculated $dh/dt$ of Thorthormi Glacier is equivalent to 60 % of the negative SMB, implying that one-third of the surface ablation is counterbalanced by ice dynamics. In other words, dynamically induced ice thickening partly compensates for the negative SMB.

Experiment 1 demonstrates that the difference in emergence velocity between the land- and lake-terminating glaciers leads to contrasting thinning patterns. On the other hand, Experiment 2 demonstrates that the emergence velocity was less negative (−0.78 ± 0.28 m a$^{-1}$) in the absence of a glacial lake in Lugge Glacier, resulting in a decrease in the thinning rate by 12 % as compared with the lake-terminating condition. The negative emergence velocity suggests that Lugge Glacial Lake could have been inevitably formed, and the more negative emergence velocity caused by the development of the lake would have accelerated the thinning of Lugge Glacier. For Thorthormi Glacier, the emergence velocity under the lake-terminating condition is negative (−1.37 ± 0.23 m a$^{-1}$), resulting in a doubled thinning rate (4.88 to 9.46 m a$^{-1}$, Table 1). Our ice flow modelling demonstrates that thinning

will be accelerated in association with the development of a supraglacial lake in the terminal part of Thorthormi
Glacier.
Contrasting patterns of glacier thinning and horizontal flow velocities between the land- and lake-terminating
glaciers are consistent with satellite-based observations over lake or ocean-terminating glaciers and neighbouring
land-terminating glaciers in the Nepal Himalaya (King et al., 2017) and Greenland (Tsutaki et al., 2016). A
decrease in the down-glacier flow velocities over the lower reaches of land-terminating glaciers suggests a
longitudinally compressive flow regime, which would result in a positive emergence velocity and therefore
thickening to compensate for the negative SMB. Conversely, for lake-terminating glaciers, an increase in the down-
glacier flow velocities suggests a longitudinally stretching flow regime, which would yield a negative emergence
velocity, resulting in accelerated ice thinning. Contrasting flow regimes modelled in this study suggest that the
mechanisms would not only be applicable to Thorthormi and Lugge glaciers, but to other lake- and land-
terminating glaciers worldwide where contrasting thinning patterns are observed.
The thinning rate calculated from the model is ~5 m $a^{-1}$ more negative than the observation over the entire
domain of Lugge Glacier and also the lower part of Thorthormi Glacier (Fig. 7b), which is probably due to the
uncertainties in the estimated ice thickness and basal sliding conditions and SMB. The two-dimensional feature is
another reason for the insufficient modelled results because the model neglects drag from the side walls and
changes in glacier width. The SMB uncertainty is < 50% over a large portion of Thorthormi and Lugge glaciers
(Fig. S11). Nevertheless, our numerical experiments suggest that dynamically induced ice thickening compensates
the negative SMB in the lower part of land-terminating glacier, resulting in less ice thinning in contrast to the lake-
terminating glacier. Further measurements of the spatial distributions of ice thickness and SMB will help in
deriving more accurate estimates of the effect of ice dynamics on glacier thinning.
**5.3 Proglacial lake development and glacier retreat**
Lugge Glacial Lake has expanded continuously and at a nearly constant rate from 2000 to 2017 (Fig. 4).
Bathymetric data suggest that glacier ice below the lake level accounted for 89% of the full ice thickness at the
calving front in 2002 (Fig. 5b). If lake level is close to the ice flotation level, where the basal water pressure equals
the ice overburden pressure, calving caused by ice flotation regulates glacier front position (van der Veen, 1996),
and glacier could rapidly retreat (e.g., Motyka et al., 2002; Tsutaki et al., 2011). Moreover, retreat could be
accelerated when the glacier terminus is situated on a reversed bed slope (e.g., Nick et al., 2009). A recent

numerical study estimated overdeepening of Lugge Glacier within 1500 m of the 2009 terminus (Linsbauer et al., 2016), which could cause further rapid retreat in the future. Recent glacier inventories indicate that Lugge Glacier has a smaller accumulation area than Thorthormi Glacier (Nuimura et al., 2015; Nagai et al., 2016), also suggesting that a less ice flux supplied cannot counterbalance the ongoing ice thinning.

After progressive mass loss since 2000, the front of Thorthormi Glacier detached from the terminal moraine and retreated further from November 2010 to December 2011 (Fig. 4a). The glacier ice was still in contact with the moraine during the field campaign in September 2011, but the glacier was completely detached from the moraine on the 2 December 2011 Landsat 7 image. Satellite images taken after 2 December 2011 show a large number of icebergs floating in the lake, suggesting rapid calving due to ice flotation. A numerical study suggested that lake water currents driven by valley wind over the lake surface could enhance thermal undercutting and then calving when a proglacial lake expands to a certain longitudinal length (Sakai et al., 2009). A previous study estimated that the overdeepening of Thorthormi Glacier extends for > 3000 m from the terminal moraine (Linsbauer et al., 2016), which suggests that continued glacier thinning will lead to rapid retreat of the entire section of the terminus as the ice thickness reaches flotation.

Experiment 2 simulates a significant increase in surface flow velocity at the lower part of Thorthormi Glacier when a proglacial lake forms (Fig. 6e). Previous studies reported the speed up and rapid retreat of glaciers after detachment from a terminal ridge or bedrock bump (e.g., Boyce et al., 2007; Sakakibara and Sugiyama, 2014; Trüssel et al., 2015). In addition to the reduction in back stress, thinning itself decreases the effective pressure, which enhances basal ice motion and increases the flow velocity (Sugiyama et al., 2011). A decrease in the effective pressure also reduces shear strength of the water saturated till layer beneath the glacier (Cuffey and Paterson, 2010), though little information is available on subglacial sedimentation in the Himalayas. Acceleration near the terminus results in ice thinning and a decrease in effective pressure, which in turn leads to further acceleration of glacier flow (e.g., Benn et al., 2007b). At the calving front of the glacier, no clear acceleration was observed during 2002–2011 (Fig. 3a), it is likely that the thinning and retreat of Thorthormi Glacier will be accelerated in the near future due to the formation and expansion of the proglacial lake.

## 6 Conclusions

To better understand the importance of glacial lake formation on rapid glacier thinning, we carried out field and satellite-based measurements across the lake-terminating Lugge Glacier and the land-terminating Thorthormi Glacier in the Lunana region, Bhutan Himalaya. Surface elevations were surveyed in 2011 by differential GPS (DGPS) across the lower parts of the glaciers and compared with a 2004 DGPS survey. Changes in surface elevation were also measured by differencing satellite-based DEMs. The flow velocity and area of glacial lake were determined from optical satellite images. We also performed numerical experiments to quantify the contributions of surface mass balance (SMB) and ice dynamics in relation to the observed ice thinning.

Lugge Glacier has experienced rapid ice thinning which is 3.3 times greater than Thorthormi Glacier, even though the SMB was less negative. The numerical modelling results, using the present glacier geometries, demonstrate that Thorthormi Glacier is subjected to a longitudinally compressive flow regime, suggesting that dynamically induced vertical extension compensates for the negative SMB, and thus results in less ice thinning than at Lugge Glacier. Conversely, the computed negative emergence velocity suggests that the rapid thinning of Lugge Glacier was driven by both surface melt and ice dynamics. This study reveals that contrasting ice flow regimes cause different ice thinning observations between the lake- and land-terminating glaciers in the Bhutan Himalaya.

Thorthormi Glacier has been retreating since 2000, resulting in the detachment of the glacier front from the terminal moraine and the formation of a proglacial lake in 2011. Ice flow modelling with the lake-terminating boundary condition indicates a significant increase in surface flow velocities near the calving front, which leads to continued glacier retreat. This positive feedback will be activated in Thorthormi Glacier with the expansion of the proglacial lake, causing further thinning and retreat in the near future.

*Data availability.* The ALOS satellite data are available for purchase from the Remote Sensing Technology Center of Japan (https://www.restec.or.jp/en/). The Landsat 7 ETM+ satellite data are distributed by the United States Geological Survey (http://landsat.usgs.gov/). ASTER-DEM data are distributed by the National Institute of Advanced Industrial Science and Technology (https://gbank.gsj.jp/madas/?lang=en). High Mountain Asia 8-metre DEM data are distributed by the NASA National Snow and Ice Data Center Distributed Active Archive Center.

*Author contributions.* KF and AS designed the study. KF, JK, TN, PT, and ST conducted the field survey in 2011. KF analysed the DGPS survey data in 2004 and 2011, and simulated the surface mass balance. TN calculated the

satellite-based surface flow velocities. SS provided ice flow models. ST analysed the data. ST and KF wrote the
paper, with contributions from AS and SS.

*Competing interests.* The authors declare that they have no conflict of interest.

*Acknowledgement.* We thank the Department of Geology and Mines, Bhutan, for providing the opportunity and
permission to conduct the field observations. We thank S. Takenaka, M. Sano, A. Sasaki, K. Ghallay and logistic
members for their support during the field campaign. We appreciate F. Pellicciotti, M. Truffer and four anonymous
referees for their thoughtful and constructive comments. This research was supported by the Science and
Technology Research Partnership for Sustainable Development (SATREPS), supported by the Japan Science and
Technology Agency (JST) and the Japan International Cooperation Agency (JICA). Support was also provided by
the Funding Program for Next Generation World Leading Researchers (NEXT Program, GR052), and JSPS-
KAKENHI (grant numbers 26257202 and 17H06104).

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

**Table 1:** Observed rate of elevation changes ($dh/dt$), calculated surface mass balance (SMB), and simulated
emergence velocity ($v_e$) and $dh/dt$ for the ablation area of Thorthormi and Lugge glaciers in the Lunana region,
Bhutan Himalaya. $b_{ie}$ enotes ice-equivalent SMB.

| Glacier | | Thorthormi | Lugge |
|---|---|---|---|
| DGPS $n$ | | 431 | 248 |
| $dh/dt$ (m a$^{-1}$) | DGPS | −1.40 ± 0.27 | −4.67 ± 0.27 |
| | ASTER | −1.61 ± 2.75 | −2.24 ± 2.75 |
| SMB (m w.e. a$^{-1}$) | Ablation area | −7.36 ± 0.12 | −5.25 ± 0.13 |
| | Debris-covered area | −7.30 ± 0.13 | −5.41 ± 0.18 |
| | Debris-free area | −9.31 ± 0.68 | −7.33 ± 0.41 |
| Exp. 1 (m a$^{-1}$) | $b_{ie}$ | −8.09 ± 0.13 | −5.77 ± 0.14 |
| | $v_e$ | +3.21 ± 0.21 | −1.69 ± 0.18 |
| | $dh/dt$ | −4.88 ± 0.34 | −7.46 ± 0.32 |
| Exp. 2 (m a$^{-1}$) | $b_{ie}$ | −8.09 ± 0.13 | −5.77 ± 0.14 |
| | $v_e$ | −1.37 ± 0.23 | −0.78 ± 0.28 |
| | $dh/dt$ | −9.46 ± 0.36 | −6.55 ± 0.42 |


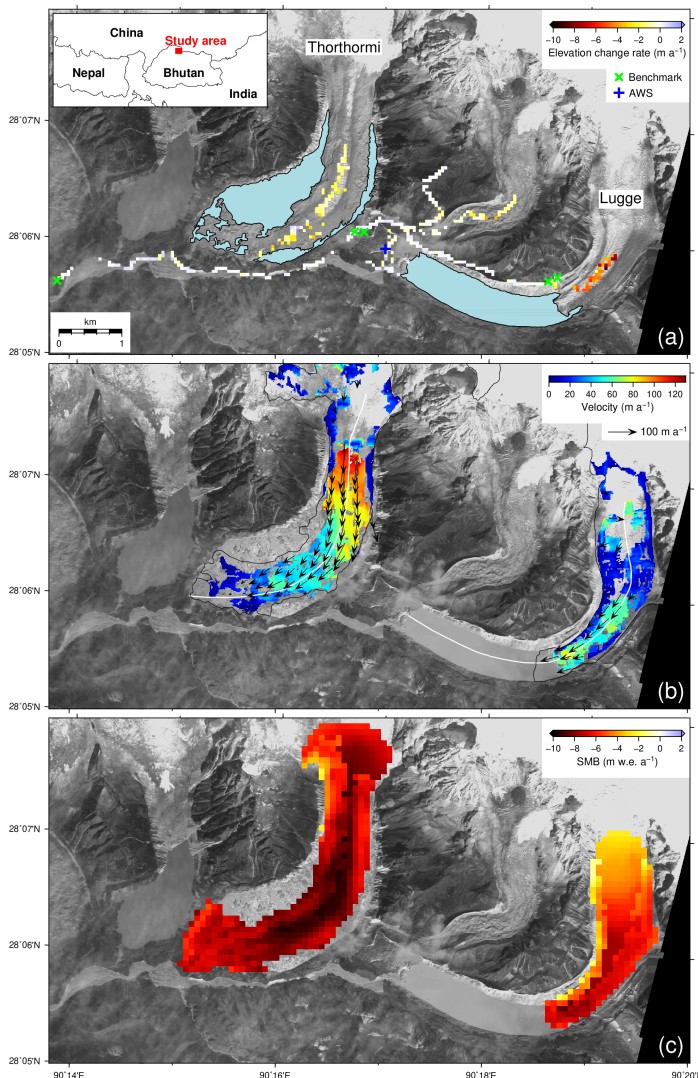


**Figure 1:** Glaciers and glacial lakes in the Lunana region, Bhutan Himalaya, superimposed with (a) the rate of elevation change ($dh/dt$) for the 2004–2011 period derived from DGPS-DEMs, (b) surface flow velocities (arrows) with magnitude (colour scale), between 30 January 2007 and 1 January 2008, and (c) simulated surface mass balance (SMB) for the 1979–2017 period. The inset in (a) shows the location of the study site. The $dh/dt$ in (a) is depicted on a 50 m grid, which is averaged from the differentiated 1 m DEMs. Light green and blue crosses are the benchmark locations used for the GPS surveys in 2004 and 2011 and of the automatic weather station (AWS) installed in 2002. Light blue hatches indicate glacial lakes in December 2009. The background image is an ALOS PRISM scene from 2 December 2009 (Ukita et al., 2011; Nagai et al., 2017). White lines in (b) indicate the central flowline of each glacier.


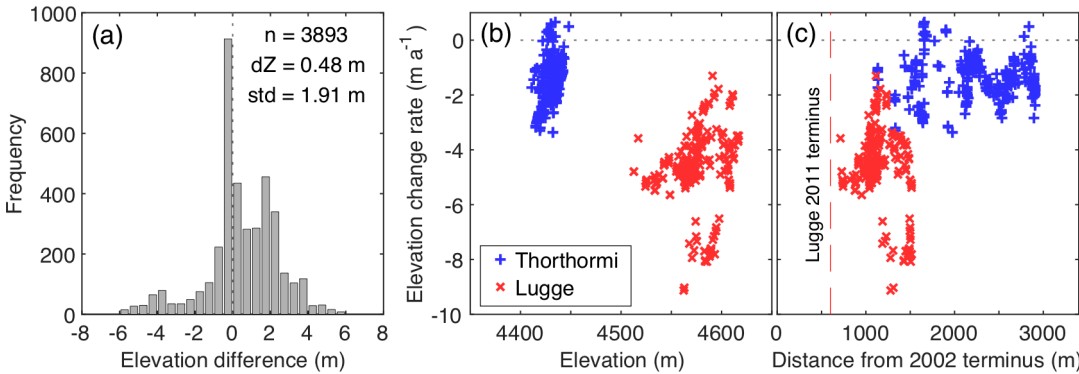


**Figure 2:** (a) Histogram of elevation differences over off-glacier at 0.5 m elevation bins. The rate of elevation
change for Thorthormi (blue) and Lugge (red) glaciers is compared with (b) elevation in 2011, and (c) distance
from the glacier terminus in 2002 along the central flowlines (Fig. 1b). The red dashed line in (c) denotes the
location of the calving front of Lugge Glacier in 2011.

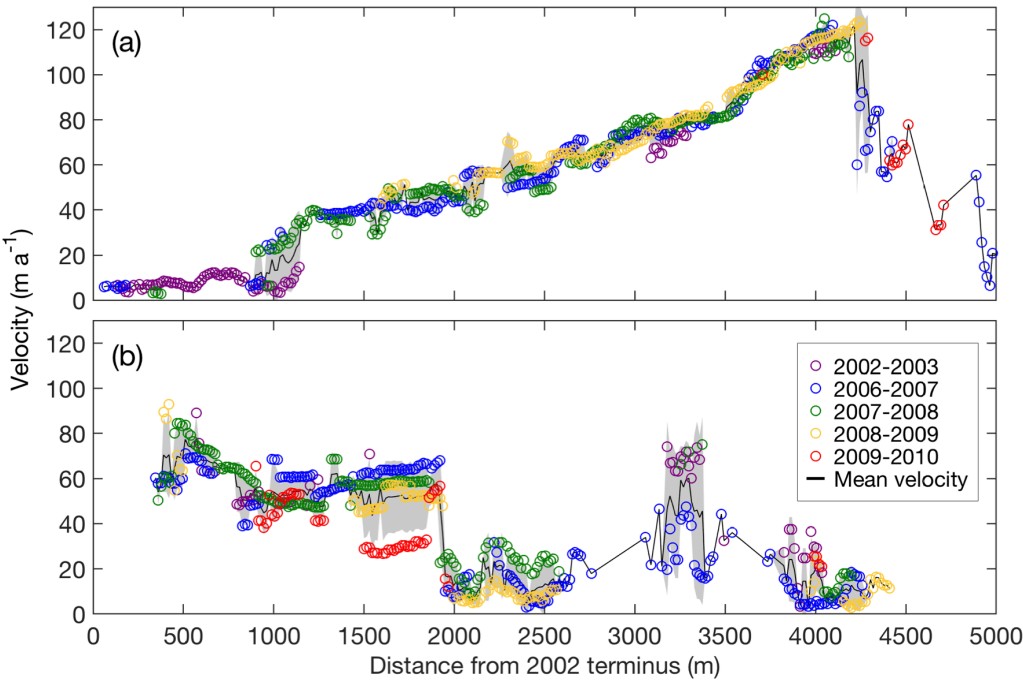


**Figure 3:** Surface flow velocities along the central flowlines of (a) Thorthormi and (b) Lugge glaciers for the
2002–2010 study period. The black lines are the mean flow velocities from 2002 to 2010, with the shaded grey
regions denoting the standard deviation. The distance from each respective 2002 glacier terminus is indicated on
the horizontal axis.


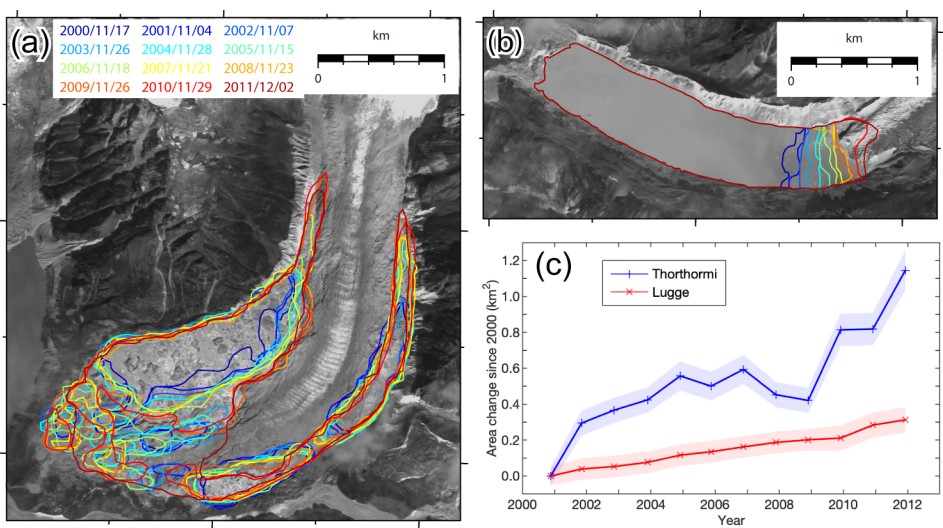


**Figure 4:** Glacial lake boundaries in (a) Thorthormi and (b) Lugge glaciers from 2000 to 2012, and (c) cumulative
lake area changes of the glaciers since 17 November 2000. The background image is an ALOS PRISM image
acquired on 2 December 2009.


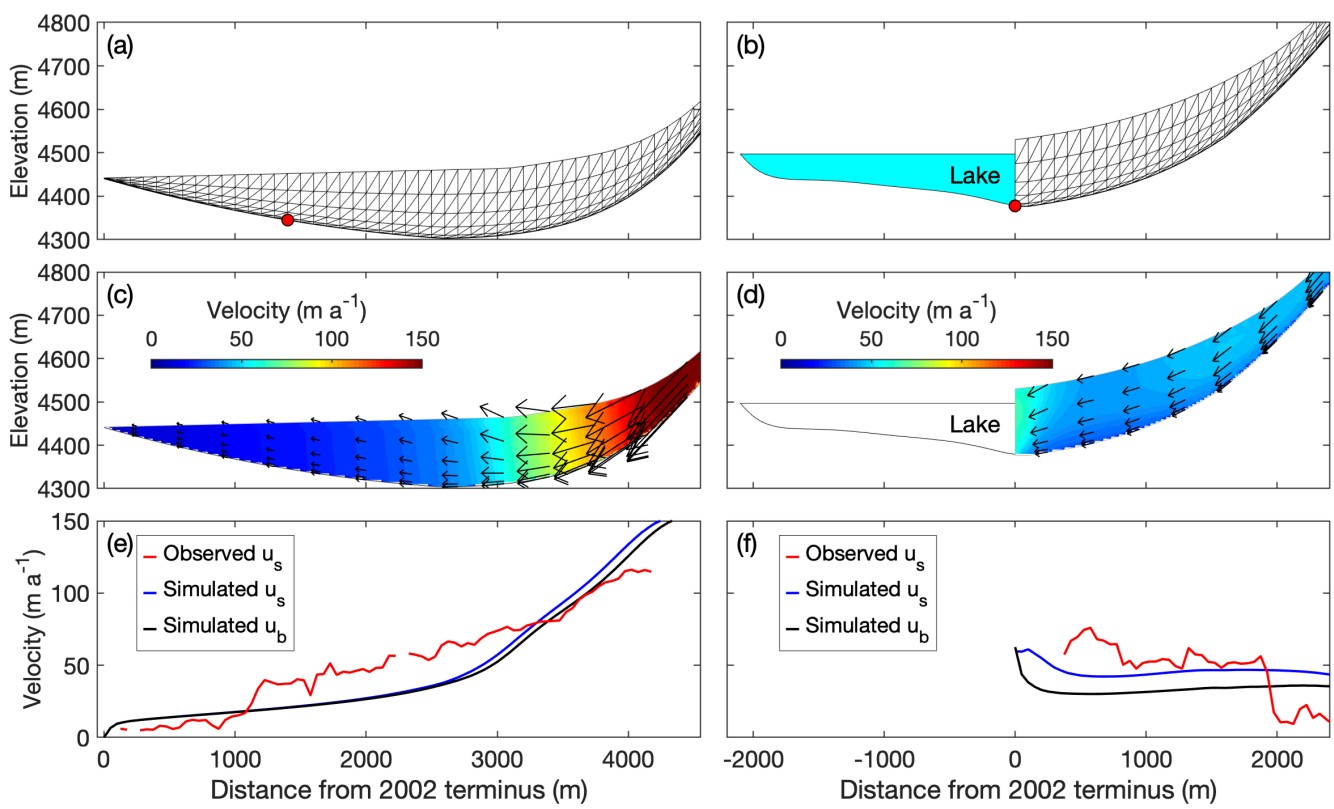

**Figure 5:** Ice flow simulations in longitudinal cross sections of Thorthormi (left panels) and Lugge (right panels) glaciers, with the present geometries of the glaciers employed in the models. (a and b) Finite element meshes used for the simulations, with red markers indicating the bedrock elevation based on a bathymetric survey. The light blue shading in (b) indicates Lugge Glacial Lake. Simulated (c and d) two-dimensional flow vectors (magnitude and direction) and (e and f) horizontal components of the flow velocity. The blue and black curves are the simulated surface ($u_s$) and basal velocities ($u_b$), respectively. The red curves are the observed surface flow velocities for 2002–2010.

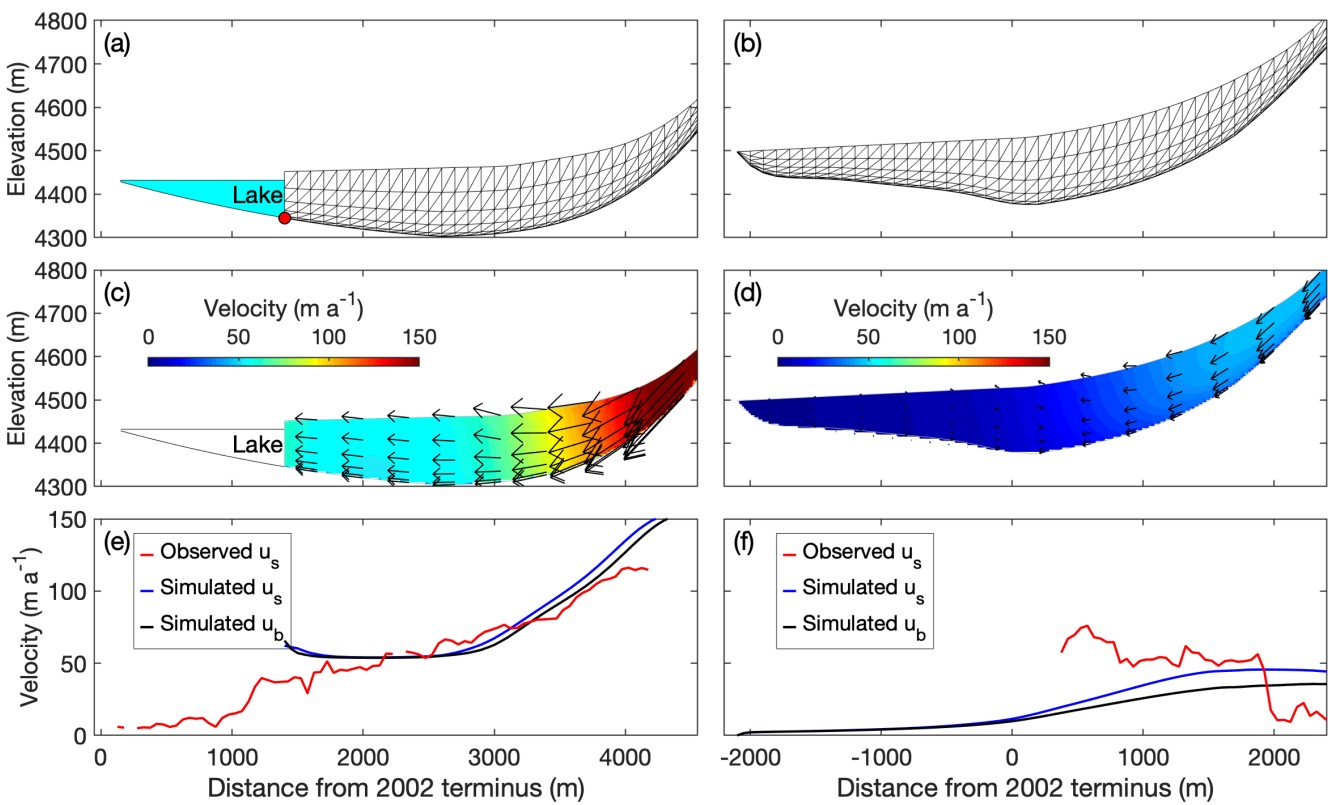

873

**Figure 6:** Ice flow simulations in longitudinal cross sections of Thorthormi Glacier under the lake-terminating condition (left panels), and Lugge Glacier under the land-terminating condition (right panels). (a and b) Finite element meshes used for the simulation. The light blue shading in (a) indicates the proglacial lake in front of Thorthormi Glacier. Simulated (c and d) two-dimensional flow vectors (magnitude and direction) and (e and f) horizontal components of the flow velocity. The blue and black curves are the simulated surface ($u_s$) and basal velocities ($u_b$), respectively. The red curves are the observed surface flow velocities for 2002–2010.

880

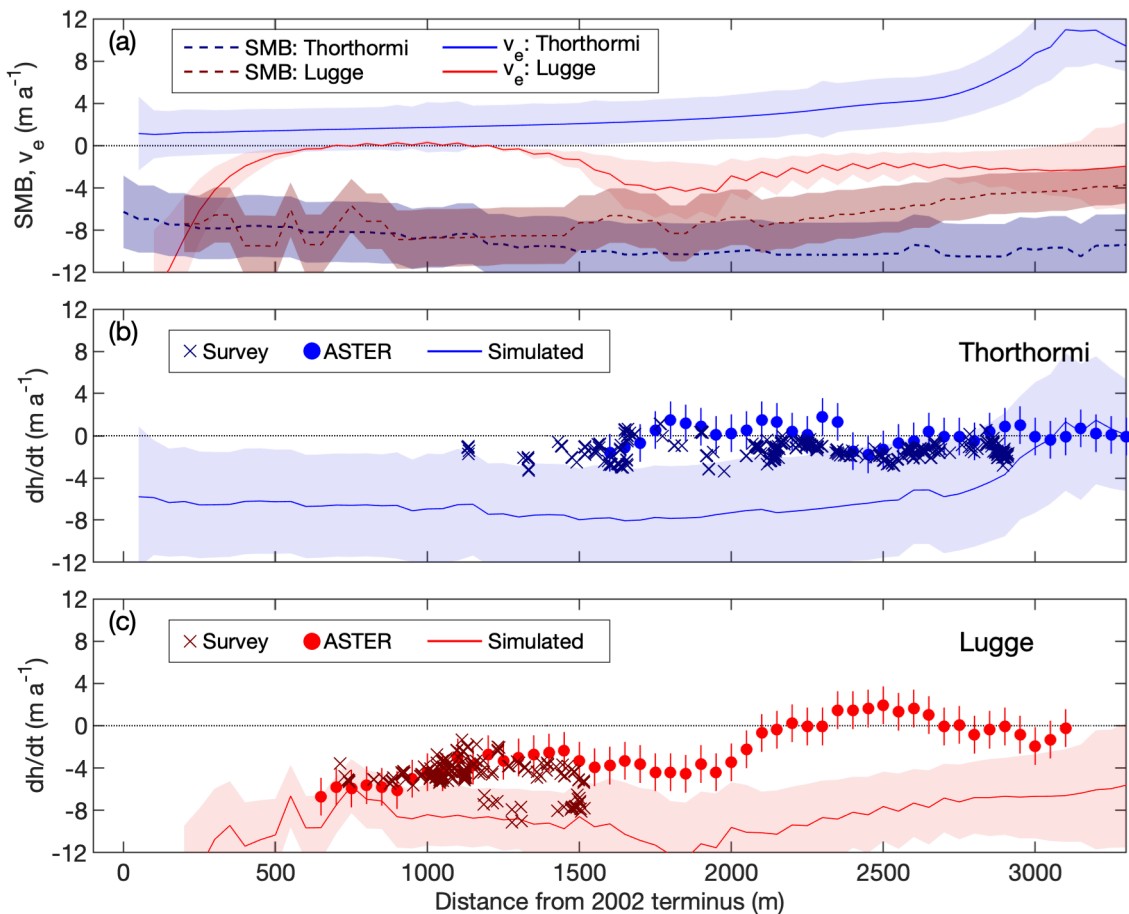

881

**Figure 7:** (a) Simulated surface mass balance (SMB) and emergence velocity ($v_e$) calculations along the central flowlines of Thorthormi and Lugge glaciers. Rate of elevation change ($dh/dt$), from survey and ASTER-DEMs during 2004–2011, and model simulations for (b) Thorthormi and (c) Lugge glaciers. Shaded regions denote the model uncertainties for each calculation.

886