# Peer review of "Contrasting thinning patterns between lake- and land-terminating glaciers in the Bhutan Himalaya"

_The Cryosphere, 2018_

## Referee Comment (RC1) · Anonymous Referee #1 · 13 Mar 2018

General comments

In this study, the authors combine field measurements (DGPS), satellite image analysis (debris thickness estimation, surface velocity fields) and modelling (2D flow model, surface mass balance (SMB) simulated from an energy balance model) to assess how sensitive the thinning rate of two glaciers in Bhutan is to the presence or not of a proglacial lake. This question is important because many studies have already observed higher thinning rates for lacustrine terminating glaciers than for land terminating glaciers (a complete list of references addressing such observations is available line 37). But the reasons for this different behavior are still not entirely clear (accelerated

flow and calving, flotation).

This study includes different steps: 1. Thinning rates estimation using DGPS measurements; 2. Debris thickness estimation using thermal ASTER images 3. SEB modelling and 4. Flow modelling. My main concern comes from the fact that each step mentioned above has large uncertainties (see my comments below) that are not possible to quantify because there is almost no validation (except surface velocity fields derived from optical satellite images). As a consequence, the results are rather subjective and are, in my opinion, not supported. This is a pity, because the study is interesting and exhaustive, but in state, it looks like more a theoretical numerical exercise than a field case study. I have some suggestions below to try to evaluate (only qualitatively though) the reliability of some results, but without validation dataset, I doubt that the results can be fully supported. I found interesting the experimental strategy (starting from the present state for experiment 1, and exploring the opposite situation removing or adding a proglacial lake for experiment 2), but according to me, given that the debris thickness spatial variability is likely to be badly reproduced, the SMB is highly uncertain, or the bedrock topography extremely simplified, such experimental exercise concerns more synthetic glaciers than true case studies. If there is no possibility to validate the results at each step, I recommend to stick with the theoretical approach (applying the flow model to a synthetic glacier with and without proglacial lake, prescribing a vertical mass balance gradient observed on Himalayan glaciers, including a debris cover part such as Chhota Shigri Glacier, India for instance – Azam et al, Annals Glaciol. 57, 328–338, 2016) than trying to relate this study to a true case study.

Surface elevation changes have been obtained by interpolating points surveyed by DGPS in the field with obviously a limited number of points (approx. 5000 to 26000 surveyed points over glaciers TabS1). Given that those glaciers are rather large (approx. 13, 11 and 3 km2, respectively for thorthormi, Luge and Lugge II), the number of points is not very large (corresponding to a relative coverage <1% of the total glacier surface). These glaciers are also heavily debris covered (with supra glacial lakes and

likely cliffs, the latter not being mentioned in the text though), and in turn with a large variability of their thinning rates (e.g., Immerzeel et al., Remote Sensing Env., 150 (2014) 93–103). Consequently, using an interpolation technique to derive the glacier surface thinning rate is questionable. The expected accuracy is therefore probably very bad, and I doubt that the standard errors displayed in table 1 (a few cm) obtained from the surveyed points can be applied to the whole glacier surface. The authors should comment on this, and should explore how sensitive the results of their study are to these glacier surface thinning rates, which are likely to be very different from their point thinning rates (with a difference potentially as high as a few meters in some areas i.e. cliffs, ponds...). In my opinion, the authors should compare their DEM with DEM obtained from satellite images.

SMB simulations depend on the debris thickness (obtained from ASTER thermal imagery known to be potentially inaccurate), as well as a surface energy balance model based on a large set of hypothesis and parameters (i.e. T=0°C at the ice-debris interface, linear debris temperature profile within the debris (lines 162-63), surface roughness, albedo of the debris, or bare ice to list only some sensitive parameters – see table 1 of Fujita and Sakai, 2014 for a complete list of parameters). Even though there is no information regarding the used parameter set, I presume that most of these parameters have been taken from a previous study conducted on Tso Rolpa catchment in Nepal (Fujita and Sakai, 2014) where the surface energy balance has been validated using hydrological and meteorological observations. We do not know if the parameters used in Fujita and Sakai (2014) are transferable to this present catchment in Bhutan. In short, there are a large amount of sources of uncertainties (not discussed in this present study), which prevent the results from being reliable if not validated. Looking at results of SMB (Fig 1c), point surface mass balance are very negative. The authors compute SMB of -7 m w.e./a over debris cover areas (section 4.4). To my knowledge, such very negative values of point SMB have never been observed in the Himalayas beneath debris. Plausibly, such values could correspond to very thin debris cover (a few mm or cm, before the maximum of the Ostrem curve) but given the location of these

areas (in the lower part of the glaciers where the debris thickness is expected to be the largest), it is highly unlikely. Moreover, the studied glaciers are debris covered, with potentially cliffs and ponds at their surface (is it true? No information regarding cliffs in this study) so the SMB spatial variability is supposed to be very high (e.g., Immerzeel et al, Remote Sensing Env., 150 (2014) 93–103; Buri et al., Ann Glaciol. 57(71), 199–211, 2016, Miles et al, Ann glaciol., 57(71), 29–40,2016) although the SMB map displayed in Fig1c does not show large spatial heterogeneities. In order to evaluate the reliability of the SMB results, a map showing the debris thickness over the 3 glaciers would be necessary. It would be useful also to show the SMB gradient as a function of elevation. And a sensitivity test including all parameters is necessary to test the reliability of the results.

The application of the debris flow model in 2 opposite configurations (experiments 1 and 2) is interesting but the bedrock topography is potentially very different from reality. Either the authors stick with a theoretical case (using an idealized synthetic glacier with a prescribed bedrock topography) or they make a sensitivity analysis using different bedrock topographies, sliding coefficients... A sensitivity test has been performed (section 5.2) but I believe that the explored range of ice thickness (+/-10 m) or sliding coefficient (+/-10%) should be much wider.

May be I missed something but SMB is estimated over the period 2002-2004, elevation change over the period 2004-2011 and flow velocities are simulated over the period 2002-2010 (but it is not clear for the latter). The periods do not match although results of thinning rate, or SMB are compared each other. How are data/results extrapolated in time? This might bring another layer of uncertainty to the results.

I found it confusing to have a study focusing on 2 glaciers (Thorthormi, Lugge) but including in fact 3 glaciers (the 2 previous glaciers and Lugge II). I know that finally most of the study is based on the comparison between Thorthormi and Lugge, because the flow model has not been applied on Lugge II, but finally why? Would the results have been different? With this partial study on Lugge II, I do not see any added value.

Specific comments

Line 19: Mölg instead of Mörg, same in the reference list

Line 21-22: images used by Bajracharya et al (2014) in 1980 to quantify the area reduction of Himalayan glaciers were full of snow, and in turn the area reduction from 1980 to 2010 is likely to be exaggerated. I recommend to report here the area reduction from 1990 to 2010, likely more accurate. This comment is valid for every places where this study is cited (section study area). It might be useful to compare the glacier area reduction obtained in this present study (section 3.3 – period 2000-2011) with the results of Bajracharya et al (2014) for the period 2000-2010.

L24: -0.22 +/- 0.12 m w.e./a (Gardelle et al, 2013) is not restricted to the ablation area but for the entire glaciers: this figure corresponds to the region-wide mass balance. Same comment for Maurer et al (2016), -0.17 m w.e./a is the glacier wide mass balance, not the ablation area

L30: may be worth updating the reference and citing Huss and Hock, 2018 here (Nature Climate Change, VOL 8 | FEBRUARY 2018 | 135–140 | www.nature.com/natureclimatechange)

L54: I disagree with this statement, DEM differencing using satellite images do allow extracting signals of a few meters, especially with the new generation of satellite images i.e. Pléiades, World view... The best proof of this are the references just cited above.

L56-58: in Nepal, Vincent et al (2016) show that the repeated DGPS profiles performed in the field were accurate enough to extract a thinning rate along the considered profile, but more importantly, they also said that this thinning rate along the profile is not representative of the whole glacier surface, or cannot be extrapolated in space given that the spatial variability of this thinning rate is extreme over debris covered tongues, due to the large variability of debris thickness and heterogeneity, presence of ponds or cliffs. Therefore, using remote sensing techniques (satellite, UAV) to obtain a thinning

rate over the debris cover tongue is more accurate than performing sporadic repeated DGPS profiles.

L67: it might be worth including the elevation range of each glacier, at least to have an idea of their maximum elevation, and the fact that they are potentially cold or polythermal. This issue is important for flow modelling

L100: the benchmarks for DGPS measurements are indicated in fig 1 (4 green crosses) but there is no benchmark visible on Fig 1 2.5 km from Thorthormi snout. Did you relate benchmarks indicated in Fig 1 to the benchmark obtained with PPP processing?

L174 details without e

L200: are the glaciers of this study temperate?

L203: what is the elevation at 5100 and 3500 m of the termini of Thorthormi and Lugge glaciers, respectively?

L207: strange to see the appearance of Fig 6 right after fig1

L255-59: not very consistent to say earlier that the inter annual variability is somehow questionable (l129) and then to discuss here this interannual variability! Is it truly significant?

Fig 4a: it is strange and not very consistent to see the annual glacier outlines crossing each other, as if from one year to the following, some areas of the glacier were expanding while some others were shrinking. This is likely not to be realistic.

Line 268-69: on fig 4b, we observe the opposite, with the northern half retreating less rapidly than the southern half

L281: given that the uncertainty on the SMB difference between both glaciers is expected to be very high (see general comment), the result "substantial influence of glacier dynamics on ice thickness change" is not supported as long as there is no sensitivity test on the SMB results, or any additional information to validate SMB simulations.

L292-94: I do not agree with the authors when they are mentioning that the agreement between observed and simulated surface velocities are good (fig 6e and f, lines red and blue, respectively). Looking at fig 7f, it is hard to believe that there is no more than 7% difference between observations and simulations: how is it obtained? More importantly, the velocity depends on the bedrock topography, obtained from Farinotti et al (2009). How reliable is it? how sensitive is the bedrock topography on velocity fields?

L327: "over recent decades" give the exact period to facilitate the comparison with the period 1974-2006.

Table1: I am confused about the periods: dh or dh/dt are obtained during 2004-2011, but SMB are obtained during 2002-04 and simulated dh/dt during 2002-2010. Not all periods match which makes also the comparison not very reliable. Another question regarding SMB in table 1, over which area of the glacier is it calculated?

L332-33: Gardelle, Brun and Kaab studies cover more or less the same period i.e. 1999-2001; 2000-2016 and 2003-2008 respectively (with Kaab study being shorter though) and the results are not always significantly different (i.e. Brun and Kaab) so I agree that we can say that the mass loss is intensified since 2000, but only based on the comparison of these 3 studies with Maurer's covering 1974-2006. I also totally agree that this acceleration is potentially not significant as stated lines 327-328

L344-47: somehow senseless and not very relevant to compare SMB and thinning rates over disconnected periods (2002-04 and 2004-11, respectively) especially because SMB may have large inter-annual variability.

L355: the emergence velocity obtained from equation 11 is very sensitive to the choice of the surface slope alpha. How is it obtained? From a DEM, which resolution?

L358: negative emergence velocity is submergence velocity?

L374-76: the mismatch between model and observation may have other origins than only the ice thickness or sliding coefficient: other sources of uncertainties may come from SEB computation affecting the model results or interpolation of DGPS measurements impacting thinning rate observations. A systematic sensitivity analysis is needed. Farther in the text, the authors claim that the SEB uncertainty is 11% based on fig S1b which shows the standard deviation of the thermal resistance. Actually, there are much more sources of uncertainties and the SEB uncertainty is likely much higher. (see general comments)
* * *

---

## Referee Comment (RC2) · Anonymous Referee #2 · 13 Mar 2018

This manuscript presents measurements of areal and surface elevation change, satellite-derived surface velocity data and modelled mass balance and ice dynamics data for three glaciers in the Bhutan Himalaya. One of these glaciers is land-terminating, another is transitioning between land-terminating and lake-terminating, and the third is lake-terminating. The ultimate goal is to be able to test whether proglacial lake development leads to increased glacier thinning rates. The conclusion is that it does, and that the glacier transitioning from land- to lake-terminating will accelerate and thin further as the proglacial lake develops. The manuscript is well-written, appropriately and clearly structured and the figures are good quality, but further work is required before it can be published in The Cryosphere.

[Figure]

Major comments:

1. My main concern relates to the lack of any real sensitivity testing to the many components that are assumed or estimated in the modelling – particularly relating to the surface mass balance. The stated uncertainty in the thermal resistance calculations are > 60 % alone. . . As a minimum it would be helpful to see the output from the debris thickness modelling to see if it is realistic. There are further assumptions relating to the linear temperature profile and albedo, for example, that need to be accounted for since the estimated mass balances are very negative compared with previous studies. How much impact are these terms having on the results? The ice flow modelling is simple, which is not a problem in itself, but certainly it would help to see some of the input datasets such as the ice thickness map to convince the reader it is somewhat realistic. And what impact does the chosen sliding coefficients have on the modelled results (beyond figure S3)?

2. The main conclusion of the manuscript is, as I understand it, that lake development does impact ice dynamics, and therefore thinning rates. I didn't get this from first reading, mainly because the two glaciers on which the manuscript focuses (Thorthormi and Lugge) are not easy to compare – they have very different geometries, different debris distributions, and different flow regimes (even before accounting for lake vs no-lake). Given this, perhaps spending a bit more time looking at the lake- vs no-lake simulations for Lugge Glacier might help (the latter of which is given little attention at present). And/or looking further at what has happened at Thorthormi following lake development (see point 4 below). There are also several statements about the low impact of ice dynamics on the thinning rates of Lugge Glacier, yet a final forecast of rapid changes at Thorthormi Glacier once the lake develops – how can these two assertions be reconciled? Is it that emergence velocity at Lugge would be (more) positive in the absence of a lake? Overall, spending some further time sharpening the take-home message would be beneficial.

3. Somewhere it needs to be explicitly acknowledged that this is a very (very) small

sample. While the field data clearly cannot be replicated, an abundance of satellite remote sensing data are available to test some of these ideas across the broader Lunana area. I acknowledge this would require significant further data processing, but augmenting the dataset would certainly give the study more substance.

4. The forecast for an impact on ice dynamics at Thorthormi is interesting, but represents a missed opportunity I think. Why not test this prediction, using velocity (and perhaps also surface elevation) data derived from more recent satellite imagery (it has been 7 years since detachment from the terminus). If this analysis does indeed show that the glacier has accelerated and thinned, it would add great weight to the existing conclusions.

Minor comments (per line number)

1-5: these two sentences are almost identical. Suggest rewording one or the other.

5: spell out GPS in full

6: move 'for the 2004-2011 period' to end of sentence

12: does it really 'more than offset' glacier thinning? Surely this would result in thickening? Suggest 'compensates'. . .

24: insert 'particularly' before 'sensitive' given that all glaciers are impacted by changes in temperature and precipitation

28: remove 'therefore' given this sentence is not substantiated by preceding text

29: what is meant by 'mechanisms' – this is rather vague. . .

47: spell out GPS at first use in main text

54: I'm not sure remote sensing methods can't measure several metres of change. How about lidar? Suggest change to 'small' changes in surface elevation.

55: change 'sub-metre' to 'centimetric'?

57: change 'performed' to 'acquired'

59: remove 'rapid' since no results have been presented at this stage of the manuscript

63-64: yes, but the glaciers are entirely different in geometry – some better justification for site selection is required here

65: using 'dynamic thinning' here is pre-emptive – it could be thickening too... maybe change to 'dynamics'?

65-66: change 'the surveyed glacier thinning' to 'changes in glacier surface elevation'

72: is this thinning rate a mean value for the ablation area? Needs specifying.

75: is this what defines a land-terminating glacier? Does whether it is grounded or floating not represent a better criterion?

101 and elsewhere: I'm not sure what TCD protocol is for referencing web pages but this is awkward – can the full url not be put in the reference list?

112: very few points of elevation change are shown in Figure 1... where can I see these 431, 248 and 258 points?

114: 'off-glacier' should be hyphenated

119: specify the sample number is 'n'

125: comment on the quality of the co-registration?

131: how can a single window size of 16 x 16 pixels be multi-scale?

136: replace 'aerial' with 'areal'

141-143: why exclude the ponds? Would these not have ice beneath or do you think they have melted down to bedrock? Does this explain the very odd digitising of glacier area presented in Figure 4a?

152: change 'calculated' to 'estimated' given there are many uncertainties in the modelling

159: that's a large uncertainty. How does it propagate through for the rest of the modelling?

230: make it clear here that you're simulating a lake-free Lugge Glacier – I read this that at present the lake is frozen! Suggest 'For Lugge Glacier, we simulate a lake-free situation, with ice flowing to the contemporary terminal moraine' or similar

315-316: are these both '-3 to 0 m a-1' by coincidence or is there a typo?

341-342: but you go on to show that dynamics only play a minor role in thinning at Lugge... are you suggesting dynamics were more important following initial lake development?

344: specify this is 'simulated' SMB...

427: does this statement that dynamic thinning is small at Lugge not undermine the main take-home message of the manuscript?

537: replace 'Mörg' with 'Mölg'...

Figure 1: can you indicate the ponds that ultimately coalesce into a lake on Thorthormi?

Figure 3: can you be sure these data towards the terminus of Lugge are not tracking the recession of the ice-front? How do you avoid matching the ice-front (i.e. the dominant feature) in these locations?

Figure 4: how were these outlines derived? They look very odd to me, with no obvious distinction in the debris-cover around any of the digitised outlines...

---

## Referee Comment (RC3) · Anonymous Referee #3 · 28 Mar 2018

The manuscript by Tsutaki et al. presents a comparison of three glaciers in the Bhutan Himalaya. Two of the three glaciers are studied to determine differences in glacier dynamics, retreat and mass wastage between land-terminating and lake-terminating glaciers, and whether the presence of a proglacial lake increases dynamics and ice wastage. To do this the authors: (a) present in situ measurements of surface elevation made using DGPS in 2004 and 2011 and compare them with remotely sensed elevation changes reported in literature; (b) derive glacier surface flow velocities using feature tracking on ASTER satellite imagery; (c) manually delineate retreat of the tongues using Landsat 7 imagery; (d) model surface mass balance of the debris covered glaciers, and (e) present a two-dimensional ice dynamics model and two model

experiments. The results show that the lake-terminating glacier (Lugge) has consider-able higher thinning rates than the land-terminating glacier (Thorthormi), but that this is mainly caused by differences in ice dynamics and not by differences in surface mass balance. A strong emergence is present for Thorthormi due to its longitudinally com-pressive flow regime that offsets its much more negative surface mass balance, and this is largely absent for Lugge. The manuscript in generally well-written besides some style issues, and the subject is of interest to the readers of the Cryosphere. There are, however, some technical issues and uncertainties with the modelling. At least moderate revisions are required before the manuscript can be published.

Major comments:

The authors present three glaciers in the manuscript: Thorthormi, Lugge and Lugge II. Lugge II was measured using the DGPS, was included in the spaceborne flow velocity measurements and was included in the SMB calculations, and its results are presented in figures 1–3. However, it is not included in the ice dynamics model experiments, is barely discussed in the results and discussion, is not included in the abstract and therefore seems of little significance to the overall story. The authors argue in the introduction that Lugge II is at a different elevation and is therefore difficult to compare to the other two glaciers, but there are many more factors that control the dynamics and mass balance of the glaciers that can complicate the comparison, also between Thorthormi and Lugge, which should be acknowledged. In this light it is also odd that the authors state that the surface mass balance of Thorthormi is 37% more negative than Lugge because it is situated at lower elevation (L360). I would suggest that the authors decide to either remove Lugge II Glacier completely from the manuscript to focus more on a clear comparison of Thorthormi and Lugge, or to consistently include all glaciers in all analyses.

I think more discussion on and comparison with other lake/land terminating glaciers reported in literature is required in the manuscript. This is touched on lightly in the manuscript but needs be more elaborate, especially because at present only two

glaciers are used in draw conclusions and hypothesise on the dynamics of lake/land terminating glaciers. Can the differences in dynamics that are found for these glaciers transfer to others? Why, why not?

Most of the methods deployed by the authors have considerable ranges of uncertainty and those should be addressed and discussed better. Especially the SMB modelling that is largely based on the rather uncertain thermal resistance obtained from ASTER data and a few (risky) assumptions seems to prone to uncertainty. The relatively very large negative SMB of Thorthormi is therefore questionable in my opinion. It has been shown in a number of articles in the last few years that using spaceborne thermal infrared imagery over debris-covered glaciers (e.g. Rounce & McKinney, Foster et al., Mihalcea et al, Gibson et al.) provides opportunities but also numerous difficulties and these should be acknowledged. The accuracy of the '1 m resolution' DEM obtained by IDW interpolation of a seemingly very limited number of moderately well distributed DGPS points (Fig 1a) is also uncertain. Maybe it's spatial variability could be validated/substituted with other DEMs. What about the new High Mountain Asia DEMs available at NSIDC DAAC? I think the manuscript would greatly benefit from a more comprehensive sensitivity analysis (e.g. Monte Carlo) to show the total range of uncertainties affecting the final interpretations.

Line by line comments:

12: Why 'more' than offsets? Rephrase

18: Maybe add Scherler 2011 (10.1038/ngeo1068)

30: There are some more recent papers on this, also by Huss and Hock themselves: 10.1038/s41558-017-0049-x, 10.1038/nature23878, 10.1038/s41558-018-0093-1.

54: 'of most spaceborne DEMs'. DEMs of high resolution stereo satellite imagery (e.g. Pleiades), LIDAR and UAVs are remote sensing methods perfectly capable of deriving several metres (even sub-metre) elevation change

59-66: The aim of the paper is not entirely clear to me from the introduction. This paragraph now is more of a methods summary. Consider changing it to clearly convey the research aim and question.

69: Is the glacier area measurement really accurate to 0.01 km2?. Same for other glaciers.

75: Use separate paragraphs per glacier to improve readability.

77: moraine-dammed

75: No space between '∼' and the number. Throughout.

86: 'were carried out on/around' -> 'were performed for'

92: These are surface flow velocities, right, not integrated over the vertical? Rephrase into something like 'surface flow velocity of Thorthormi'.

101: I've never seen full web URLs in a body of a paper. Use a reference entry for the website in the bibliography instead.

104: Are the elevation variations caused by a person carrying the pack on a debris-covered glacier really only 10 cm? How were these estimated?

106: Mentioning UTM is not quite relevant.

107: Why this 1 m resolution?

122: remove 'the' after calculated.

125-126: There is no info whatsoever on the accuracy of the orthorectification of the ASTER images? Could these maybe be retrieved? This can be quite an issue in steep mountainous regions.

131: Why did you select the statistical correlation mode. I was under the impression that the mode that works on the frequency domain is better and is better suited detect subpixel displacements. Please elaborate on the choice.

134: So no filtering was applied using the signal to noise ratio statistics that are provided by COSI-Corr?

137: The SLC-off gaps in the ETM+ imagery did not provide an issue in the analysis? This should at least be mentioned.

139: 'that possessed the' -> 'with'

141: Use of QGIS in particular is not relevation and, again, the weburl is unnecessary. Just state that delineates were performed in a geographical information system.

143: So there is no user-induced accuracy error?

162-163: These are quite bold assumptions and this should be acknowledged.

260: Preferably also show the off-glacier displacements in figure 3a.

273: Is does not appear that heterogeneous to me. Especially since the actual heterogeneity is likely much higher given the hummocky surface of most debris-covered glaciers. I understand that this is not captured by the ASTER data, and that this is the variability that is obtained, but it should be reworded.

330-333: The authors speak of accelerating mass losses, but the numbers and accompanying year ranges do not show this per se.

341-342: Please elaborate.

344-365: I found this section rather confusing. There are methods and results presented in the discussion section. I strongly suggest relocating this to the appropriate sections.

360: This is not due to differences in debris cover and debris thickness?

375: A difference of 5 ma-1 is a lot. As suggested in the main comments, I think a comprehensive sensitive analysis would be a great addition to the paper and could help to support the conclusions.

---

## Editor Comment (EC1) · F. Pellicciotti (Editor) · 9 Apr 2018

Dear authors,

I have now received three reviews of your paper. I have read them carefully and I would like to thank the reviewers very much for their detailed and thorough comments. All three reviewers agree that substantial revisions of your paper are required, and I agree with their overall ealuation and the main points raised by the reviewers. In particular, you need to address the large uncertainties in all four main steps of calculations: i) estimates of thinning rates from interpolation of DGPS measurements; ii) debris thickness estimation from ASTER thermal imagery; iii) surface energy balance (SEB) modelling;

[Figure]

and iv) flow modelling. This is a concerned shared by all reviewers, related to a lack of validation of those calculations, which questions your final results. All reviewers share a concern for the very high values of SEB obtained over the debris-covered areas (7m w.e./a, Section 4.4.). You should also clarify the different periods that the calculations refer to and how this affects your results. I also agree with the fact that three glaciers are presented but one, Lugge II, is not included in the ice dynamics model experiments, is barely discussed in the results and discussion and is not included in the abstract, and this inconsistency should be resolved.

The reviewers all make important suggestions to improve your paper. A second round of reviews will most likely be required after revision. I encourage you to thoroughly address all concerns raised by the reviewers and look forward to receiving the revised manuscript.

———————————————

---

## Author Comment (AC1) · 15 May 2018

Reply to Referee #1

We would like to thank the referee for thoughtful and useful comments. In the following, we describe our responses (in blue) point-by-point to each referee comment (*italic*).

In this study, the authors combine field measurements (DGPS), satellite image analysis (debris thickness estimation, surface velocity fields) and modelling (2D flow model, surface mass balance (SMB) simulated from an energy balance model) to assess how sensitive the thinning rate of two glaciers in Bhutan is to the presence or not of a proglacial lake. This question is important because many studies have already observed higher thinning rates for lacustrine terminating glaciers than for land terminating glaciers (a complete list of references addressing such observations is available line. But the reasons for this different behavior are still not entirely clear (accelerated flow and calving, flotation).

This study includes different steps:

1. Thinning rates estimation using DGPS measurements;

2. Debris thickness estimation using thermal ASTER images

3. SEB modelling

and 4. Flow modelling.

**Major comments:**

My main concern comes from the fact that each step mentioned above has large uncertainties (see my comments below) that are not possible to quantify because there is almost no validation (except surface velocity fields derived from optical satellite images). As a consequence, the results are rather subjective and are, in my opinion, not supported. This is a pity, because the study is interesting and exhaustive, but in state, it looks like more a theoretical numerical exercise than a field case study. I have some suggestions below to try to evaluate (only qualitatively though) the reliability of some results, but without validation dataset, I doubt that the results can be fully supported. I found interesting the experimental strategy (starting from the present state for experiment 1, and exploring the opposite situation removing or adding a proglacial lake for experiment 2), but according to me, given that the debris thickness spatial variability is likely to be badly reproduced, the SMB is highly uncertain, or the bedrock topography extremely simplified, such experimental exercise concerns more synthetic glaciers than true case studies. If there is no possibility to validate the results at each step, I recommend to stick with the theoretical approach (applying the flow model to a synthetic glacier with and without proglacial lake, prescribing a vertical mass balance gradient observed on Himalayan glaciers, including a debris cover part such as Chhota Shigri Glacier, India for instance – Azam et al, Annals Glaciol. 57, 328–338, 2016) than trying to relate this study to a true case study.

Because some in-situ data (e.g., SMB and bedrock topography) is unavailable for the studied glaciers, validation of the results at each step is difficult. We therefore performed (1) validation of spatial representativeness in thinning rate obtained from DGPS with that from satellite-derived DEM (see reply to the major comment #2). We also performed sensitivity analyses for (2) SMB modelling and (3) glacier flow modelling to estimate uncertainties. For the SMB modelling, we recalculated spatial distribution of debris thermal resistance with considering sensible heat flux (see reply to the major comment #3). Detailed bedrock topography is unavailable for all glaciers, but we alternatively evaluated sensitivity of modelled ice speed against changes in ice thickness and basal sliding coefficient. We also computed RMSE between modelled and observed ice speed as a measure of the uncertainty (see reply to the major comment #4).

Surface elevation changes have been obtained by interpolating points surveyed by DGPS in the field with obviously a limited number of points (approx. 5000 to 26000 surveyed points over glaciers TabS1). Given that those glaciers are rather large (approx. 13, 11 and 3 km2, respectively for thorthormi, Luge and Lugge II), the number of points is not very large (corresponding to a relative coverage

**Figure R1:** Elevation differences in the ice-free area (left) between 2004 ASTER-DEM and DGPS-DEM and (right) between 2011 ASTER-DEM and DGPS-DEM.

**Figure R2:** Rate of elevation change for the 2004–2011 period derived from ASTER-DEMs (background shadings) and DGPS-DEMs (circles filled with the same color scale). Glacier outlines are of December 2011.

---

## Author Comment (AC2) · 15 May 2018

Reply to Referee #2

We would like to thank the referee for thoughtful and useful comments. In the following, we describe our responses (in blue) point-by-point to each referee comment (*italic*).

This manuscript presents measurements of areal and surface elevation change, satellite-derived surface velocity data and modelled mass balance and ice dynamics data for three glaciers in the Bhutan Himalaya. One of these glaciers is land-terminating, another is transitioning between land-terminating and laketerminating, and the third is lake-terminating. The ultimate goal is to be able to test whether proglacial lake development leads to increased glacier thinning rates. The conclusion is that it does, and that the glacier transitioning from land- to laketerminating will accelerate and thin further as the proglacial lake develops. The manuscript is well-written, appropriately and clearly structured and the figures are good quality, but further work is required before it can be published in The Cryosphere.

**Major comments:**

1. My main concern relates to the lack of any real sensitivity testing to the many components that are assumed or estimated in the modelling – particularly relating to the surface mass balance. The stated uncertainty in the thermal resistance calculations are > 60 % alone. . . As a minimum it would be helpful to see the output from the debris thickness modelling to see if it is realistic. There are further assumptions relating to the linear temperature profile and albedo, for example, that need to be accounted for since the estimated mass balances are very negative compared with previous studies. How much impact are these terms having on the results? The ice flow modelling is simple, which is not a problem in itself, but certainly it would help to see some of the input datasets such as the ice thickness map to convince the reader it is somewhat realistic. And what impact does the chosen sliding coefficients have on the modelled results (beyond figure S3)?

Although debris thickness was not measured during the field campaign, ice is exposed from place to place over Thorthormi and Lugge Glaciers (Figs R1a and R1b), suggesting that debris-cover is rather thin than that of Lugge II Glacier (Fig. R1c). In addition, few supraglacial ponds and ice cliff exist over Thorthormi and Lugge Glaciers. So we emphasize that spatial variability of elevation change, thermal resistance and SMB are less than those the reviewers supposed. Anyhow, following the referees suggestion, we recalculated thermal resistance with considering sensible heat, for which pressure level temperature and geopotential height of NCEP2 are taken into account (Fig. R2). Scatter plot and spatial distribution of thermal resistances derived from the original method (net radiation only) and from recalculated one (net radiation + sensible heat) are shown in Figs R3 and R4. Spatial distribution of the difference between the two results is also shown in Fig. R4c. Thermal resistance significantly increased after the consideration of sensible heat (Fig. R3). However, large difference appeared only near the western margin (Fig. R4) probably because of relatively thick debris covering the area. We will recalculate the SMB distribution with the revised thermal resistance in the revised manuscript. We evaluated sensitivity of calculated meltwater against meteorological parameters (Fig. R5). We chose the meltwater instead of SMB to quantify the uncertainty in percentage. The tested parameters are surface albedo, air temperature, precipitation, relative humidity, solar radiation, thermal resistance and wind speed. Uncertainty of thermal resistance and albedo were assumed to be 100% and 40% based on Figs R2b and R2d. Uncertainties of each meteorological variable were assumed to be RMSEs of ERA-Interim reanalysis data against the observational data (Fig. R6). Variations in meltwater within a possible parameter range are estimated by quadratic sum of results from each parameter shown in Fig. R5. Estimated uncertainty of meltwater is less than 50% at a large part of Thorthormi and Lugge Glaciers (Fig. R7). We will replace figures by the recalculated results and add Figs R5, R6 and R7 to the revised supplement.

Detailed bedrock topography is unavailable for all the glaciers. Available bedrock topography data were shown in Figs 6a and 6b in the discussion paper. Because our model geometry is not based on data, we performed sensitivity analysis using the broader range  $(\pm 30\%)$  of the sliding coefficient and ice thickness (Fig. R8). The RMSE between the modeled and measured flow velocities were computed as a measure of the model performance (Fig. R9). For Thorthormi Glacier, the model is similarly sensitive to sliding coefficient and ice thickness. For Lugge Glacier, the model is more sensitive to ice thickness than sliding coefficient. Figs R8 and R9 will be added to the revised supplement.

**Figure R1:** Photographs showing surface condition near the termini of (a) Thorthormi (18 September 2011), (b) Lugge (20 September 2011) and Lugge II Glaciers (21 September 2011).

---

## Author Comment (AC3) · 15 May 2018

Reply to Referee #3

We would like to thank the referee for thoughtful and useful comments. In the following, we describe our responses (in blue) point-by-point to each referee comment (*italic*).

The manuscript by Tsutaki et al. presents a comparison of three glaciers in the Bhutan Himalaya. Two of the three glaciers are studied to determine differences in glacier dynamics, retreat and mass wastage between land-terminating and laketerminating glaciers, and whether the presence of a proglacial lake increases dynamics and ice wastage. To do this the authors: (a) present in situ measurements of surface elevation made using DGPS in 2004 and 2011 and compare them with remotely sensed elevation changes reported in literature; (b) derive glacier surface flow velocities using feature tracking on ASTER satellite imagery; (c) manually delineate retreat of the tongues using Landsat 7 imagery; (d) model surface mass balance of the debris covered glaciers, and (e) present a two-dimensional ice dynamics model and two model experiments. The results show that the lake-terminating glacier (Lugge) has considerable higher thinning rates than the land-terminating glacier (Thorthormi), but that this is mainly caused by differences in ice dynamics and not by differences in surface mass balance. A strong emergence is present for Thorthormi due to its longitudinally compressive flow regime that offsets its much more negative surface mass balance, and this is largely absent for Lugge. The manuscript in generally well-written besides some style issues, and the subject is of interest to the readers of the Cryosphere. There are, however, some technical issues and uncertainties with the modelling. At least moderate revisions are required before the manuscript can be published.

**Major comments:**

The authors present three glaciers in the manuscript: Thorthormi, Lugge and Lugge II. Lugge II was measured using the DGPS, was included in the spaceborne flow velocity measurements and was included in the SMB calculations, and its results are presented in figures 1–3. However, it is not included in the ice dynamics model experiments, is barely discussed in the results and discussion, is not included in the abstract and therefore seems of little significance to the overall story. The authors argue in the introduction that Lugge II is at a different elevation and is therefore difficult to compare to the other two glaciers, but there are many more factors that control the dynamics and mass balance of the glaciers that can complicate the comparison, also between Thorthormi and Lugge, which should be acknowledged. In this light it is also odd that the authors state that the surface mass balance of Thorthormi is 37% more negative than Lugge because it is situated at lower elevation (L360). I would suggest that the authors decide to either remove Lugge II Glacier completely from the manuscript to focus more on a clear comparison of Thorthormi and Lugge, or to consistently include all glaciers in all analyses.

We excluded Lugge II Glacier from the detailed discussion because lack of the bed topography hampered the flow modelling. Spatial variability in surface elevation change derived from ASTER-DEMs is greater than those of Thorthormi and Lugge Glaciers (Fig. R1). Therefore, it is unsure whether the elevation change derived from DGPS-DEMs is representative. For these reasons we excluded Lugge II Glacier from the surface velocity measurements, SMB modelling and the detailed discussion. We will remove descriptions and figures related to Lugge II Glacier from the revised manuscript but we will leave the rate of elevation change of the glacier as an observational fact.

Comparison of Thorthormi and Lugge Glaciers are not easy. We hypothesize that the emergence velocity will decrease at Thorthormi Glacier after the expansion of the supraglacial lake, resulting in an increase in ice thinning rate as observed in Lugge Glacier. To test this hypothesis, we will discuss the influence of lake expansion on the emergence velocity based on lake- (Experiment 1) and land-terminating (Experiment 2) simulations for Lugge Glacier in the revised manuscript.

**Figure R1:** Rate of elevation change for the 2004–2011 period derived from ASTER-DEMs (background shadings) and DGPS-DEMs (circles filled with the same color scale). Glacier outlines are of December 2011.

I think more discussion on and comparison with other lake/land terminating glaciers reported in literature is required in the manuscript. This is touched on lightly in the manuscript but needs be more elaborate, especially because at present only two glaciers are used in draw conclusions and hypothesise on the dynamics of lake/land terminating glaciers. Can the differences in dynamics that are found for these glaciers transfer to others? Why, why not?

More rapid thinning of lake-terminating glacier than neighboring landterminating glacier has been revealed by satellite remote sensing observations in Bhutan (Maurer et al., 2016), Nepal (Nuimura et al., 2012; King et al., 2017), Pamir-Karakoram-Himalaya (Gardelle et al., 2013) and Alaska (Trüssel et al., 2013). However, the previous studies have not quantified contributions of ice dynamics and SMB to those contrasted glacier thinning. This point is a unique approach of this study. In the revised manuscript, we will discuss the differences in ice dynamics affecting ice thinning by comparing a ratio of thinning rates between land- and lake-terminating glaciers reported by the previous studies. Most of the methods deployed by the authors have considerable ranges of uncertainty and those should be addressed and discussed better. Especially the SMB modelling that is largely based on the rather uncertain thermal resistance obtained from ASTER data and a few (risky) assumptions seems to prone to uncertainty. The relatively very large negative SMB of Thorthormi is therefore questionable in my opinion. It has been shown in a number of articles in the last few years that using spaceborne thermal infrared imagery over debris-covered glaciers (e.g. Rounce & McKinney, Foster et al., Mihalcea et al, Gibson et al.) provides opportunities but also numerous difficulties and these should be acknowledged.

Although debris thickness was not measured during the field campaign, ice is exposed from place to place over Thorthormi and Lugge Glaciers (Figs R2a and R2b), suggesting that debris-cover is rather thin than that of Lugge II Glacier (Fig. R2c). In addition, few supraglacial ponds and ice cliff exist over Thorthormi and Lugge Glaciers. So we emphasize that spatial variability of elevation change, thermal resistance and SMB are less than those the reviewers supposed. Anyhow, following the referees suggestion, we recalculated thermal resistance with considering sensible heat, for which pressure level temperature and geopotential height of NCEP2 are taken into account (Fig. R3). Scatter plot and spatial distribution of thermal resistances derived from the original method (net radiation only) and from recalculated one (net radiation + sensible heat) are shown in Figs R4 and R5. Spatial distribution of the difference between the two results is also shown in Fig. R5c. Thermal resistance significantly increased after the consideration of sensible heat (Fig. R4). However, large difference appeared only near the western margin (Fig. R5) probably because of relatively thick debris covering the area. We will recalculate the SMB distribution with revised thermal resistance in the revised manuscript. We evaluated sensitivity of calculated meltwater against meteorological parameters (Fig. R6). We chose the meltwater instead of SMB to quantify the uncertainty in percentage. The tested parameters are surface albedo, air temperature, precipitation, relative humidity, solar radiation, thermal resistance and wind speed. Uncertainty of thermal resistance and albedo were assumed to be 100% and 40% based on Figs R3b and R3d. Uncertainties of each meteorological variable were assumed to be RMSEs of ERA-Interim reanalysis data against the observational data (Fig. R7). Variations in meltwater within a possible parameter range are estimated by quadratic sum of results from each parameter shown in Fig. R6. Estimated uncertainty of meltwater is less than 50% at a large part of Thorthormi and Lugge Glaciers (Fig. R8). We will replace figures by the recalculated results and add Figs R6, R7 and R8 to the revised supplement.

---

## Author Comment (AC4) · 15 Jun 2018

Dear Editor Francesca Pellicciotti

We are grateful for the opportunity to revise our manuscript (tc-2018-15) entitled "Contrasting thinning patterns between lake- and land-terminating glaciers in the Bhutan Himalaya", and the helpful comments from three referees. Please find enclosed our point-by-point responses to three referees comments as well as the revised marked-up manuscript showing the revised sentences highlighted in blue and red. We hope that our manuscript was substantially improved and now it will be suitable for publication in The Cryosphere. We look forward to hearing from you concerning your editorial

decision.

Yours sincerely,

Shun Tsutaki on behalf of co-authors

---

## Referee Report (RR1)

The manuscript by Tsutaki et al. combines surface elevation changes, surface flow velocities, changes in glacial area, and surface mass balance and ice dynamic modelling to investigate the difference in thinning rates of two neighbouring glaciers in the Bhutan Himalaya. One of these glaciers is lake-terminating, the other is defined as land-terminating; the aim of the study is to examine the influence of a proglacial lake on the thinning rates of the glaciers. The study is interesting and the figures are clear and well presented. The manuscript is mostly well-written, but various sections need to be better discussed, explained and justified so that it is clear how the authors have achieved their results. Although the results and conclusions appear to be well supported, there are large uncertainties in the modelling that make it hard to determine how well supported these conclusions are. It would be very interesting for the modelling predictions to be compared with what has occurred to Thorthormi Glacier since 2011, as suggested by a previous reviewer, although it is acknowledged that this would involve a lot more work.

General comments:

1. I find it quite difficult to follow the state of a proglacial lake (or not) at the terminus of Thorthormi Glacier, as contradicting information is drip-fed through the manuscript. Is there a proglacial lake, and when does it form? The timeline (to date) needs to be presented coherently in Section 2 and other mentions in the manuscript then need to correspond. This could also be helped by delineating the lakes in one of the panels in Figure 1. Although the authors made a case against this in their response as the lake area varies over time, even a visualisation of the minimum/maximum lake area for each glacier would be beneficial. A brief investigation of historical Google Earth imagery showed me that Thorthormi Glacier has a proglacial lake that has expanded up both sides of the glacier tongue, along the lateral moraines. I had not grasped this from the manuscript, having understood that there were a few limited supraglacial ponds at the margins.

2. The discussion in Section 5.3 of whether Lugge Glacier has reached floatation seems to be too little, too late. If the glacier tongue was near floatation in 2002, is it not possible it reached floatation during the study period? And if so, surely that may have a large impact on the results: how is it known whether changes in surface elevation are due to the glacier thinning, or changes in the lake volume? This is rather a large assumption to make, and needs justifying or debating earlier in the manuscript.

3. I'm not sure I see the value in including the change in glacial area as delineated from Landsat images. It involves quite large uncertainties, and in my opinion, detracts from a study that is assessing how proglacial lakes affect glacial thinning rates.

4. The comparison of the DGPS-DEM with the ASTER-DEM, after the ASTER-DEM has been calibrated with the DGPS data seems circular, and thus contradictory. If the ASTER-DEM has been calibrated with the DGPS data, it would be fairly obvious that the changes in elevation from each would be very similar. The suggestion from a previous reviewer was to compare the DGPS data with a DEM that has been externally verified, but ASTER does not have any ground-truthing, so it is not an ideal product for this. I suggest that the ASTER-DEM (calibrated with the DGPS data) is instead used for subsequent analysis of surface

elevation changes, which would give elevation changes over a larger area and thus mitigate the large uncertainties involved with interpolating the DGPS data.

5. A number of the uncertainties that are reported are extremely large, and in some cases the errors are significantly larger than the values presented. Furthermore, the calculation and reporting of uncertainties is not consistent through the manuscript (e.g. DGPS DEM uncertainty calculations in Section 3.1 which are reported as something different in Section 4.1), some have not/cannot be given a value (such as the interpolation of the DGPS DEM), uncertainties are given as a range of values without explanation (L437), and other uncertainty ranges are discussed but not defined (L344), but yet the data do not lie within any uncertainty (one standard deviation?) of each other. Similarly, in one instance, calculated and observed velocities are reported as being within 7% of each other, yet in the Author Response this is reported as a mean and therefore only half the data will lie within this %. Other specific issues include the use of a quadratic sum (L134 & L239), which is not defined nor what Bolch et al. use, and the undefined extrapolation of the bedrock upglacier in the model inputs (L278), which must also have large uncertainties.

6. Debris cover is mentioned sporadically throughout the manuscript, but without any real weight or meaning. The authors clearly consider it important as it is mentioned in the abstract, a debris melt model is included in the SMB calculations, and debris-covered areas must have been calculated for different SMB values to be presented for debris and debris-free regions in Section 4.4. These values are certainly interesting, but without an idea of the debris thickness in the debris-covered surfaces, are not so useful (L368-371). The manuscript would benefit from a greater discussion of how the debris cover varies (expand the paragraph from L99 and refer to Figure S1), and what influence this might have on the conclusions drawn about ice dynamics.

Specific comments (by line number):

1: Suggest singularising the title and every other occurrence of "lake- and land-terminating glaciers" to "a lake- and a land-terminating glacier", as you now only consider one of each type.

14: "supraglacial lakes" is not directly relevant, unless you specify that supraglacial lakes can evolve into proglacial lakes, which is what you consider in this study. Perhaps change to "proglacial lakes".

23: "over half" – later in the manuscript, this value is reported as three quarters?

33: What do you mean by "the Bhutanese glaciers"? All Bhutanese glaciers?

37: Change "monsoon influenced humid" to "monsoon-influenced, humid"

38-40: This sentence would read better if it were condensed.

L43-46: This section would also benefit from a brief description of how proglacial lakes can be formed from supraglacial lake formation, growth, coalescence and downcutting, which is likely what has occurred at Thorthormi Glacier in recent years?

L51: No space between number and % sign, throughout manuscript.

L65-67: Two points are made here about the superiority of remote-sensing techniques: 1. that the surface of debris-covered glaciers can be highly variable due to cliffs and ponds, making access difficult to get large amounts of data; and 2. that UAVs can potentially obtain higher-resolution imagery than satellite-imagery, and thus resolve the highly variable thinning rates more accurately. Suggest restructure to make these points clear.

L68-69: DGPS is not remotely sensed, so this doesn't follow on from the previous sentence unless it is explained why it is superior (higher resolution data/no permits required compared to UAVs/etc).

L69: Change "is" to "can be".

L76-77: This sentence sounds like the only reason the glaciers were selected was because of their similar elevations. Any other reasons – debris cover, size, access? Suggest adding the sentence about safety and proximity to trekking routes, but surely the most important reason is the lake/developing-lake scenarios?

L88: If something thins at a negative rate, it is thickening. Remove negative sign. Same on L95.

L99: Needs more discussion of the ponds and cliffs on each glacier to later prove these don't contribute to thinning rates. Google Earth shows a heavy amount of crevassing on both glaciers, which could be a reason why few lakes are found in the ablation regions.

L103-106: These are methodological details and would fit better in Section 3.4

L118: Move website to reference list, and reference instead. Same for L125.

L122-123: Refer to Figure S1.

L135: Change "In previous studies" to "Following previous studies"

L163: Is the 0.9 threshold picked according to information given by COSI-Corr, or some other method? Please specify. This and the next few lines are not clear: how many filters do you apply or are they all the same? Suggest rewrite to state, step-by-step, what you did and why. Should "attitude" be "altitude"?

L173-174: Remove "in a geographical information system" – manual delineation is sufficient to describe the method. Also change "Following to the previously" to "Following previously".

L185: 'Thick' debris that suppresses ice melt is anything deeper than ~5 cm.

L248: "glaciers" should be singular.

L275: Why use yet another DEM when the first section describes four different DEMs? What year was this DEM from, and why was it necessary to "filter the elevations"?

L299-301: It is not clear how the conditions for creating a proglacial lake at Thorthormi Glacier were decided upon; they currently seem entirely random. How was it decided where to 'put' the proglacial lake for Thorthormi Glacier? It doesn't look to be at the current terminus, so this needs some explanation. Similarly, how was a calving front thickness of 106 m chosen? And why would the surface level of the

proglacial lake be identical to the supraglacial ponds in 2004? I understand that this is hypothetical and likely a good place to start, but surely it depends on the time of year, amount of melt, how well the hydrological system has developed, etc, as to the level of the ponds. Additionally recommend changing "assumed" to "simulated" to make it clear this is a hypothetical modelled situation.

L305: What years do you actually run the model from and to? The input data seems to be from a large number of different years… Also needs clarifying in Section 4.5

L331: Is this from the DGPS or ASTER DEMs? Needs specifying, and also in the caption for Figure 1a.

L335: Looking at Figure 2b, it seems that the most negative values are actually found (just) within the upper elevation bands.

L353: Remove sentence beginning "In this region…" as it is an interpretation, not a result, and belongs in Discussions. Same applies for all other instances, particularly of methodological details that are mentioned in the results, but not actually in the methods section.

L355: Report "0.99 years" in number of days to be consistent with start of paragraph.

L357: Change "and accelerated loss" to "which accelerated"

L374: The debris cover must have a more significant influence than this in order to produce 2 m w.e. less melt per year, as reported in the previous paragraph?

L381: I think the experiment names would be clearer as "Present terminus conditions" and "Reversed terminus conditions" as this is what you are changing (not the geometry).

L386: Change "In contrasted to the observed decrease" to "In contrast to the downglacier decrease"

L387: At what distance from the terminus is the 40 m/y velocity reached?

L402-404: Change "Although we assumed" to "Due to the assumption that" – by assuming the glaciers to be temperate, of course most of the ice flow will be due to basal sliding. But how can most of the ice flow be due to basal sliding, but also a "moderate amount" be due to ice deformation? Suggest rewording these two sentences to be clearer as to what you found, and whether the assumption of temperate ice had a significant influence or not.

L450: The repetition of these exact same measurements looks like a mistake; suggest remove the second and change "for the period" to "for both the periods".

L501-503: Remove the reference to making hypotheses and just state what has been shown.

L506: Was the thinning accelerated?

L515: Change "less thinning" to "thickening"

L517: In Section 4.2, Lugge Glacier is described as having a convergent flow where the glacier width narrows. How might this contribute to, or affect, the stretching flow regime?

L531: Should the first 2000-2010 be 2000-2011?

L540: What are the implications of this?

Figure 1: The red and white centrelines of each glacier need more explanation, both in the figure caption and in the text. What is the difference between the red and white, and how is it used?

Figure S6: Needs error bars for the DGPS data – if these are already present, they are too faint and cannot be seen so need to be a darker colour than the ASTER error bars.

---

## Author Response (AR2)

Reply to referee comments

We would like to thank two referees for thoughtful and useful comments. In the following, we describe our responses (in blue) point-by-point to each referee comment (*italic*).

Referee #4

*The manuscript by Tsutaki et al. combines surface elevation changes, surface flow velocities, changes in glacial area, and surface mass balance and ice dynamic modelling to investigate the difference in thinning rates of two neighbouring glaciers in the Bhutan Himalaya. One of these glaciers is lake-terminating, the other is defined as land-terminating; the aim of the study is to examine the influence of a proglacial lake on the thinning rates of the glaciers. The study is interesting and the figures are clear and well presented. The manuscript is mostly well-written, but various sections need to be better discussed, explained and justified so that it is clear how the authors have achieved their results. Although the results and conclusions appear to be well supported, there are large uncertainties in the modelling that make it hard to determine how well supported these conclusions are. It would be very interesting for the modelling predictions to be compared with what has occurred to Thorthormi Glacier since 2011, as suggested by a previous reviewer, although it is acknowledged that this would involve a lot more work.*

General comments:
1. *I find it quite difficult to follow the state of a proglacial lake (or not) at the terminus of Thorthormi Glacier, as contradicting information is drip-fed through the manuscript. Is there a proglacial lake, and when does it form? The timeline (to date) needs to be presented coherently in Section 2 and other mentions in the manuscript then need to correspond. This could also be helped by delineating the lakes in one of the panels in Figure 1. Although the authors made a case against this in their response as the lake area varies over time, even a visualisation of the minimum/maximum lake area for each glacier would be beneficial. A brief investigation of historical Google Earth imagery showed me that Thorthormi Glacier has a proglacial lake that has expanded up both sides of the glacier tongue, along the lateral moraines. I had not grasped this from the manuscript, having understood that there were a few limited supraglacial ponds at the margins.*

**R#4_1:** We added below description to state the timeline of glacial lake development at Thorthormi Glacier as: " Large supraglacial lakes, which are inferred to possess a high potential for outburst flood (Fujita et al., 2008, 2013), have formed along the western and eastern lateral moraines in the

ablation area by merging multiple supraglacial ponds since the 1990s (Ageta et al., 2000; Komori, 2008). The front of Thorthormi Glacier was still in contact with the terminal moraine during our field campaign in September 2011, but the glacier was completely detached from the moraine in the Landsat 7 image of 2 December 2011. Thorthormi Glacier is therefore termed as a land-terminating glacier here since the glacier terminus was grounded during the studied period of 2004–2011." (L96-102). We added the outlines and shadings for the supra/proglacial ponds and lake of Thorthormi and Lugge glaciers in December 2009 (Fig. 1a). We delineated outlines of the lakes from 2000 to 2017 (Fig. 4).

2. *The discussion in Section 5.3 of whether Lugge Glacier has reached floatation seems to be too little, too late. If the glacier tongue was near floatation in 2002, is it not possible it reached floatation during the study period? And if so, surely that may have a large impact on the results: how is it known whether changes in surface elevation are due to the glacier thinning, or changes in the lake volume? This is rather a large assumption to make, and needs justifying or debating earlier in the manuscript.*

**R#4_2:** This comment is difficult to catch. We intend to mean that Lugge Glacier was near floatation in 2002, and then has been in the floatation condition throughout the study period. We do not assert any change in the floatation status nor acceleration of terminus retreat. In this paragraph, we have tried to explain that the continuous retreat of Lugge Glacier was attributed to the floatation condition at the terminus. We deleted the description about expansion of Lugge Glacial Lake enhancing the thinning of Lugge Glacier in the paragraph.

3. *I'm not sure I see the value in including the change in glacial area as delineated from Landsat images. It involves quite large uncertainties, and in my opinion, detracts from a study that is assessing how proglacial lakes affect glacial thinning rates.*

**R#4_3:** I (K. Fujita) personally agree to this suggestion (excluding lake expansion). On the other hand, development of Thorthormi Glacial Lake is an issue discussed in the latter part of this study. We afraid that the description about change in Thorthormi Glacier (from land-terminating to lake-terminating) is not convincing without this data. Therefore we want to keep the description about glacial lake expansion as the present one We merged Figs. 4 and 5 into one figure..

4. *The comparison of the DGPS-DEM with the ASTER-DEM, after the ASTER-DEM has been calibrated with the DGPS data seems circular, and thus contradictory. If the ASTER-DEM has*

*been calibrated with the DGPS data, it would be fairly obvious that the changes in elevation from each would be very similar. The suggestion from a previous reviewer was to compare the DGPS data with a DEM that has been externally verified, but ASTER does not have any ground-truthing, so it is not an ideal product for this. I suggest that the ASTER-DEM (calibrated with the DGPS data) is instead used for subsequent analysis of surface elevation changes, which would give elevation changes over a larger area and thus mitigate the large uncertainties involved with interpolating the DGPS data.*

**R#4_4:** We disagree to the suggestion. The calibration was made over the off-glacier terrain while the comparison of elevation changes was made over the glacier surface, so that the analysis is NOT circular. The former reviewers questioned the representativeness of DGPS-DEMs because these cover limited tracks, and then we show that the DGPS-based elevation changes fall within those from ASTER-DEMs (Fig. S7 and S8). In addition, we disagree to the use of ASTER-DEMs for the subsequent analysis because it detracts the originality of this study. Relative comparison of satellite-based DEMs is widely performed but they have large uncertainty while the DGPS-DEMs have high accuracy. In the same line, we would not like to incorporate the ASTER-derived dh/dt for the recent years because of large uncertainty.

5. *A number of the uncertainties that are reported are extremely large, and in some cases the errors are significantly larger than the values presented. Furthermore, the calculation and reporting of uncertainties is not consistent through the manuscript (e.g. DGPS DEM uncertainty calculations in Section 3.1 which are reported as something different in Section 4.1), some have not/cannot be given a value (such as the interpolation of the DGPS DEM), uncertainties are given as a range of values without explanation (L437), and other uncertainty ranges are discussed but not defined (L344), but yet the data do not lie within any uncertainty (one standard deviation?) of each other. Similarly, in one instance, calculated and observed velocities are reported as being within 7% of each other, yet in the Author Response this is reported as a mean and therefore only half the data will lie within this %. Other specific issues include the use of a quadratic sum (L134 & L239), which is not defined nor what Bolch et al. use, and the undefined extrapolation of the bedrock upglacier in the model inputs (L278), which must also have large uncertainties.*

**R#4_5:** We have checked the values shown in the manuscript. Some discrepancies are attributed to different domains. For instance, surface mass balance (SMB) over the ablation area was firstly addressed in the results section, but different values are used for calculating dh/dt with emergence velocity in the discussion section. This is because SMB along the center flowline was re-averaged in

the latter discussion. We determine to use a consistent value for one parameter in the revised manuscript. All related values are listed in Table 1.

In the previous manuscript, we have used quadratic sums of bias and standard error as uncertainty. We changed this policy because uncertainty in all other parameters except dh/dt is expressed by standard deviation. In addition, because our DGPS points are so many, use of standard error makes uncertainty too small unnaturally.

6. *Debris cover is mentioned sporadically throughout the manuscript, but without any real weight or meaning. The authors clearly consider it important as it is mentioned in the abstract, a debris melt model is included in the SMB calculations, and debris-covered areas must have been calculated for different SMB values to be presented for debris and debris-free regions in Section 4.4. These values are certainly interesting, but without an idea of the debris thickness in the debris-covered surfaces, are not so useful (L368-371). The manuscript would benefit from a greater discussion of how the debris cover varies (expand the paragraph from L99 and refer to Figure S1), and what influence this might have on the conclusions drawn about ice dynamics.*

**R#4_6:** We dealt with the debris cover effects because this study would be underappreciated without it. On the other hand, the distribution of debris thickness, which we suppose that the reviewer intended to mean by "how the debris cover varies", cannot be measured by sole in-situ observation practically because it varies too heterogeneously. We therefore adopted an idea of "thermal resistance" in this study. Large thermal resistance appeared only near the western margin of Lugge Glacier, where a relatively thick debris cover is expected and calculated SMB was less negative ($\sim-3$ m w.e. a$^{-1}$) than the other areas (Fig. 1c). Nevertheless, in-situ photographs (Fig. S1) suggest the debris covers over Thorthormi and Lugge glaciers are sparse. Although the calculated SMBs for debris-covered and -free surface are largely different by 2 m w.e., area of thick debris cover is limited and thus the averaged SMBs over the ablation area are similar to those for debris-free surface (Sect. 4.4). Thus, we considered the influence of debris cover variation on the simulated SMB and the rate of surface elevation change to be limited.

Specific comments (by line number):

*1: Suggest singularising the title and every other occurrence of "lake- and land-terminating glaciers" to "a lake- and a land-terminating glacier", as you now only consider one of each type.*

**R#4_7:** We changed the main text, but not the title for our preference. We believe that the present

title is not wrong even though the reviewer may argue that it looks misleading.

*14: "supraglacial lakes" is not directly relevant, unless you specify that supraglacial lakes can evolve into proglacial lakes, which is what you consider in this study. Perhaps change to "proglacial lakes".*

**R#4_8:** We changed to "proglacial lakes" (L15)

*23: "over half" – later in the manuscript, this value is reported as three quarters?*

**R#4_9:** We changed to "The magnitude of dynamic thickening compensates for approximately one-thirds of the negative SMB of Thorthormi Glacier." (L23-24).

*33: What do you mean by "the Bhutanese glaciers"? All Bhutanese glaciers?*

**R#4_10:** We changed from "the Bhutanese glaciers shrank by" to "the glacier area loss in Bhutan was" (L33).

*37: Change "monsoon influenced humid" to "monsoon-influenced, humid"*

**R#4_11:** We changed (L38).

*38-40: This sentence would read better if it were condensed.*

**R#4_12:** We shorten the sentence to "Mass loss of Gangju La Glacier in central Bhutan was much greater than those of glaciers in the eastern Himalaya and southeastern Tibet for the recent decade (Tshering and Fujita, 2016)." (L39-40)

*L43-46: This section would also benefit from a brief description of how proglacial lakes can be formed from supraglacial lake formation, growth, coalescence and downcutting, which is likely what has occurred at Thorthormi Glacier in recent years?*

**R#4_13:** We added a description about proglacial lake formation caused by supraglacial lake development after the sentence as "Proglacial lakes can be form by expansion and coalescence of suplaglacial ponds, which are formed in topographic hollows formerly occupied with ice, by being

fed with both precipitation and glacial meltwater." (L46-48). But we did not add "downcutting" here because we think that this issue is not generally recognized phenomena and this may confuse readers.

*L51: No space between number and % sign, throughout manuscript.*

**R#4_14:** We changed it throughout the manuscript.

*L65-67: Two points are made here about the superiority of remote-sensing techniques: 1. that the surface of debris-covered glaciers can be highly variable due to cliffs and ponds, making access difficult to get large amounts of data; and 2. that UAVs can potentially obtain higher-resolution imagery than satellite-imagery, and thus resolve the highly variable thinning rates more accurately. Suggest restructure to make these points clear.*

**R#4_15:** We changed the sentence as "because the surface of debris-covered glaciers can be highly variable, making access difficult to get large amounts of data (e.g., Gardelle et al., 2013; Maurer et al., 2016; Brun et al., 2017). In particular, unmanned autonomous vehicle (UAVs) is a powerful tool to obtain higher-resolution imagery than satellite, and thus resolves the highly variable topography and thinning rates of debris-covered surface more accurately (e.g., Immerzeel et al., 2014; Vincent et al., 2016)." (L70-74).

L68-69: DGPS is not remotely sensed, so this doesn't follow on from the previous sentence unless it is explained why it is superior (higher resolution data/no permits required compared to UAVs/etc).

**R#4_16:** The DGPS description follows the UAV in context of "high resolution". So we did not change the sentence. (L74-76)

*L69: Change "is" to "can be".*

**R#4_17:** We changed (L76).

*L76-77: This sentence sounds like the only reason the glaciers were selected was because of their similar elevations. Any other reasons – debris cover, size, access? Suggest adding the sentence about safety and proximity to trekking routes, but surely the most important reason is the lake/developing-lake scenarios?*

**R#4_18:** In the previous version of the manuscript, we suggested three reasons for selecting the glaciers as 1) they are located at similar elevations, 2) contrasting geometrical conditions of the termini (lake/land), and 3) their relatively safe ice-surface conditions and proximity to trekking routes. We modified the reasons as "Thorthormi and Lugge glaciers were selected for analysis because they have contrasting termini, grounding and fully contacting lake at similar elevations. These contrasting conditions at the similar elevations make them suitable for evaluating the contribution of ice dynamics to the observed ice thickness changes. The glaciers are also suitable for field measurements because of their relatively safe ice-surface conditions and proximity to trekking routes." (L84-88).

*L88: If something thins at a negative rate, it is thickening. Remove negative sign. Same on L95.*

**R#4_19:** We removed minus sign on both places (L95 and 106).

*L99: Needs more discussion of the ponds and cliffs on each glacier to later prove these don't contribute to thinning rates. Google Earth shows a heavy amount of crevassing on both glaciers, which could be a reason why few lakes are found in the ablation regions.*

**R#4_20:** We changed the description as "Furthermore, few supraglacial ponds and ice cliffs were observed across the glaciers. Satellite imagery show that the surface is heavily crevassed in the lower ablation areas, suggesting that glacier meltwater immediately drain into the interior of the glaciers." (L112-114).

*L103-106: These are methodological details and would fit better in Section 3.4*

**R#4_21:** We moved the sentence "Automatic weather station (AWS) observations from the terminal moraine of Lugge Glacial Lake (4524 m a.s.l., Fig. 1a) showed that the annual mean air temperature during 2002–2004 was ~0 °C, and annual precipitation was 900 mm in 2003 (Suzuki et al., 2007b)." to Section 3.4 (L219-222).

*L118: Move website to reference list, and reference instead. Same for L125.*

**R#4_22:** This format was used in a paper recently published in The Cryosphere (e.g., Friedl et al., 2018). Therefore, our manuscript also follows this format.

*L122-123: Refer to Figure S1.*

**R#4_23:** We referred Figure S1 "We neglected influence of change in debris thickness in the DGPS surveys because the debris cover across the glaciers is sparse and thin (Fig. S1)" (L133-134).

*L135: Change "In previous studies" to "Following previous studies"*

**R#4_24:** See R#4_5. We changed policy for the uncertainty.

*L163: Is the 0.9 threshold picked according to information given by COSI-Corr, or some other method? Please specify. This and the next few lines are not clear: how many filters do you apply or are they all the same? Suggest rewrite to state, step-by-step, what you did and why. Should "attitude" be "altitude"?*

**R#4_25:** We changed the description more clearly "We used a statistical correlation mode, with a correlation window size of 16 × 16 pixels and a mask threshold of 0.9 for noise reduction (Leprince et al., 2007). The obtained ice flow velocity fields were filtered to remove residual attitude effects and miscorrelations (Scherler et al., 2011b; Scherler and Strecker, 2012). We applied two filters to eliminate those flow vectors that deviated in magnitude (greater than $\pm 1\,\sigma$) or direction ($> 20°$) from the mean vector within the neighbouring 21 × 21 data points." (L171-175). "attitude" is the term in Scherler et al. (2011b) and Scherler and Strecker (2012)

*L173-174: Remove "in a geographical information system" – manual delineation is sufficient to describe the method. Also change "Following to the previously" to "Following previously".*

**R#4_26:** We removed "in a geographical information system", and changed to "Following previously" (L182).

*L185: 'Thick' debris that suppresses ice melt is anything deeper than ~5 cm.*

**R#4_27:** We changed the description "(generally more than ~5 cm)" (L195).

*L248: "glaciers" should be singular.*

**R#4_28:** We changed to "glacier" (L277).

*L275: Why use yet another DEM when the first section describes four different DEMs? What year was this DEM from, and why was it necessary to "filter the elevations"?*

**R#4_29:** DEM used in this analysis was obtained in 2001 (Fujita et al., 2008), because no other data were available when we constructed the ice flow model in 2011. However, we utilized the 30-m-grid DEMs obtained in 2004 and 2011 to calculate the thinning rates of glaciers as responses to previous review comments. We added the year of ASTER-DEM used to model domain as "The surface geometry was obtained from the 90-m-grid ASTER GDEM version 2 obtained in January 2001 after filtering the elevations with a smoothing routine at a bandwidth of 1000 m." (L303-304). The reason why applying filter to the surface and bed elevations is to avoid the influence of the small-scale surface roughness on the computation.

*L299-301: It is not clear how the conditions for creating a proglacial lake at Thorthormi Glacier were decided upon; they currently seem entirely random. How was it decided where to 'put' the proglacial lake for Thorthormi Glacier? It doesn't look to be at the current terminus, so this needs some explanation. Similarly, how was a calving front thickness of 106 m chosen? And why would the surface level of the proglacial lake be identical to the supraglacial ponds in 2004? I understand that this is hypothetical and likely a good place to start, but surely it depends on the time of year, amount of melt, how well the hydrological system has developed, etc, as to the level of the ponds. Additionally recommend changing "assumed" to "simulated" to make it clear this is a hypothetical modelled situation.*

**R#4_30:** We decided the position of hypothetical calving front at the place where the depth of the lake was only acquired from bathymetry survey in September 2011. We added this explanation for deciding the position and depth (as ice thickness) of calving front as "Position of the hypothetical calving front was determined at the place where only one lake depth was acquired from bathymetry survey in September 2011." (L331-332). Lake level was measured by our DGPS surveys in 2004 and 2011, which consist within 3 m in elevation. We disagree that the surface level of supraglacial ponds could be variable from year to year because proglacial lake water generally drains by overflowing from its outlet. Seasonal change in lake level observed in Tsho Rolpa in Nepal showed it limited with 2-m (Yamada, 1998, not cited in the manuscript). The aim of our ice flow model is to simulate the mean ice velocity field of the glaciers for the period of 2002–2010, and thus, this assumption is considered to be sufficient. We changed "assumed" to "simulated" as suggested (L331).

Yamada T (1998) Glacier lake and its outburst flood in the Nepal Himalaya. Monograph No. 1, Data Center for Glacier Research, Japanese Society of Snow and Ice.

*L305: What years do you actually run the model from and to? The input data seems to be from a large number of different years… Also needs clarifying in Section 4.5*

**R#4_31:** As mentioned in R#4_30, we do not intend to simulate any specific year but to reproduce conditions of glacier dynamics corresponding to the satellite-based mean ice velocity field over the period of 2002–2010. We also assumed boundary condition of surface elevation in 2001 and water level of supraglacial ponds in 2004. We added the description as "We employed glacier surface elevation in 2001 and water level of supraglacial ponds and proglacial lake observed in 2004 as boundary conditions (Fujita et al., 2008)." in Section 3.5.2 (L328-329).

*L331: Is this from the DGPS or ASTER DEMs? Needs specifying, and also in the caption for Figure 1a.*

**R#4_32:** This elevation change rate is from DGPS DEMs. We added the description "derived from DGPS-DEMs." (L366), which is also added to the caption for Fig. 1a.

*L335: Looking at Figure 2b, it seems that the most negative values are actually found (just) within the upper elevation bands.*

**R#4_33:** "The lower elevation band" means not lower portion of observed area but of Lugge Glacier. We do not change here.

*L353: Remove sentence beginning "In this region…" as it is an interpretation, not a result, and belongs in Discussions. Same applies for all other instances, particularly of methodological details that are mentioned in the results, but not actually in the methods section.*

**R#4_34:** We removed.

*L355: Report "0.99 years" in number of days to be consistent with start of paragraph.*

**R#4_35:** We changed to number of days as "(362 days)" (L384).

*L357: Change "and accelerated loss" to "which accelerated"*

**R#4_36:** Because we changed analysis from changes in glacier area to glacial lake area, the description here was changed as "The glacial lake area near the front of Thorthormi Glacier progressively increased from 2000 to 2017, at a mean rate of 0.09 km$^2$ a$^{-1}$ (Figs. 4)." (L386-387).

*L374: The debris cover must have a more significant influence than this in order to produce 2 m w.e. less melt per year, as reported in the previous paragraph?*

See the response R#4_6.

*L381: I think the experiment names would be clearer as "Present terminus conditions" and "Reversed terminus conditions" as this is what you are changing (not the geometry).*

**R#4_37:** We changed as the section titles (L405 and 417).

*L386: Change "In contrasted to the observed decrease" to "In contrast to the downglacier decrease"*

**R#4_38:** We changed (L410-11).

*L387: At what distance from the terminus is the 40 m/y velocity reached?*

**R#4_39:** We changed the description more clearly as "the computed velocities of Lugge Glacier are up to ~40 m a$^{-1}$ within 500–2200 m of the terminus, and it sharply increases to ~80 m a$^{-1}$ at the calving front (Fig. 5f)." (L411-412).

L402-404: Change "Although we assumed" to "Due to the assumption that" – by assuming the glaciers to be temperate, of course most of the ice flow will be due to basal sliding. But how can most of the ice flow be due to basal sliding, but also a "moderate amount" be due to ice deformation? Suggest rewording these two sentences to be clearer as to what you found, and whether the assumption of temperate ice had a significant influence or not.

**R#4_40:** The ice flow modelling indicated that ice deformation accounts for 3% and 25% of the observed surface flow velocity for Thorthormi and Lugge glaciers, respectively (Fig. 5). We believe

the influence of the assumption of temperate ice is insignificant because most of the deformation occurs near the glacier bed where ice is most likely temperate. We rewrote the sentence as "Basal sliding accounts for 97 % and 75 % of the simulated ice flow velocity in the ablation area of Thorthormi and Lugge glaciers, respectively (Figs. 5e and 5f), suggesting that ice deformation is insufficient to represent the ice flow regardless of assumption of ice temperature." (L427-429).

*L450: The repetition of these exact same measurements looks like a mistake; suggest remove the second and change "for the period" to "for both the periods".*

**R#4_41:** This is not repetition. Latter is of Thothormi Glacier while former is of all Bhutanese glaciers. We do not change here.

*L501-503: Remove the reference to making hypotheses and just state what has been shown.*

**R#4_42:** We changed the description as "On the other hand, Experiment 2 demonstrates that the emergence velocity was less negative ($-0.78 \pm 0.28$ m a$^{-1}$) in the absence of a glacial lake in Lugge Glacier, resulting in a decrease in the thinning rate by 12 % as compared with the lake-terminating condition. The negative emergence velocity suggests that Lugge Glacial Lake could have been inevitably formed, and the more negative emergence velocity caused by the development of the lake would have accelerated the thinning of Lugge Glacier." (L503-507).

*L506: Was the thinning accelerated?*

**R#4_43:** This is described in Sect. 5.1. (L463-465 of the former manuscript). However, responding to the comment from Reviewer #5, we deleted the description about acceleration of thinning. See R#5_1.

*L515: Change "less thinning" to "thickening"*

**R#4_44:** We changed (L517).

*L517: In Section 4.2, Lugge Glacier is described as having a convergent flow where the glacier width narrows. How might this contribute to, or affect, the stretching flow regime?*

**R#4_45:** We removed the speculative sentence "In this region…" as the referee suggested in

R#4_34.

*L531: Should the first 2000-2010 be 2000-2011?*

**R#4_46:** The period of 2000–2010 was correct. But, because we analyzed changes in lake area instead of the glacier area in the revised manuscript, we removed comparing glacier area with the reported value from Bajracharya et al. (2014).

*L540: What are the implications of this?*

**R#4_47:** We suggested that less accumulation area might result in less ice flux supply to the lower part, making less dynamic thickening. We changed the description more clearly as "also suggesting that a less ice flux supplied cannot counterbalance the ongoing ice thinning." (L540-541).

*Figure 1: The red and white centrelines of each glacier need more explanation, both in the figure caption and in the text. What is the difference between the red and white, and how is it used?*

**R#4_48:** The red lines indicate computed range of lake-terminating condition for each glacier (Experiment 1 for Lugge and Experiment 2 for Thorthormi), while the red + white lines show the range of computed area of land-terminating condition (Experiment 2 for Lugge and Experiment 1 for Thorthormi). These lines are not efficient to explain these ranges, so that we changed to use only white lines to cover entire range of white + red lines.

*Figure S6: Needs error bars for the DGPS data – if these are already present, they are too faint and cannot be seen so need to be a darker colour than the ASTER error bars.*

**R#4_49:** We re-depicted Figs. S7 and S8 with re-evaluated error bars but the error of DGPS is too small to be clearly seen.

**Referee #5**
*This is a very interesting paper showing the difference in thinning between neighboring glaciers that have contrasting boundary conditions (lake or no-lake). The numerical study hints at some of the reasons for the differences and a second numerical experiment explores how the glaciers would develop under different conditions. The paper is generally well written and clear and comes to significant conclusions that warrant publication in TC. I have a few general comments that should*

*be addressed, that revolve mostly around the use of calculated emergence velocities. The required revisions are somewhere between minor or major.*

General comments:

*1) When discussing causes of thinning, a bit more care could be taken. The actual values of emergence velocities do not tell you whether the cause for thinning is dynamic or not (unless they are negative in the ablation area). Generally, I believe, attribution is really only possible when comparing to a steady state. In steady state, emergence velocities compensate for SMB and those values can be large or small depending on climate. When thinning starts one can then say whether it was more negative SMB or less positive emergence. Without that context attribution becomes much trickier. This could be explained better.*

**R#5_1:** We carefully discussed this comment with coauthors. We agree with the reviewer that; "without knowing the mass balance and ice flow conditions under the steady state, it is not possible to attribute observed thinning to the mass balance nor dynamics." We understand that out texts in Sect. 5.1 and 5.2 was not accurate in this context. Our intention was to discuss quantitatively the reason why the lake-terminating Lugge Glacier thins faster than the land-terminating Thorthomi Glacier based on the SMB and ice flow modelling. As Fig. 7 shows (new number), SMBs of these two neighboring glaciers are both negative in similar degree. Ice flow model shows strain thinning at Lugge Glacier and strain thickening at Thorthomi Glacier. Further, additional "reversed" experiments of land-terminating Lugge Glacier and lake-terminating Thorthomi Glacier indicated drastic change in the flow regime before and after the lake formation. We believe that these experimental results strongly suggest that the change in the flow regime after the lake formation accelerated the thinning of Lugge Glacier. We deleted the description about thinning acceleration in Sect. 5.1 and modified Sect. 5.2 (L483-492) so that readers correctly understand our argument.

*2) Calculating emergence velocities in numerical models is fraud with potential errors. Errors in ice thickness or sliding coefficients can result in errors in velocities, which can be amplified in the emergence velocities, because those involve the divergence of horizontal. It is therefore important to compare resulting elevation change rates with measured ones. In places where SMB and emergence velocities don't sum to observed thickness changes, one needs to be extremely careful with interpreting emergence velocities. Fig. 8b indicates that there can be several m/a difference between observed and computed dh/dt and this indicates that v_e has large errors. As a matter of fact, it might be possible to adjust the model to better fit observations by adjusting both ice thickness and the sliding coefficient to fit both velocities and thickness change rates.*

**R#5_2:** We understand the emergence velocity computation is very sensitive to small changes in the conditions. To clearly show the potential errors in the modelling results, we estimated uncertainties in emergence velocities by considering error propagation of (1) accuracy of flow velocity measurements, (2) interannual variability in measured surface velocity over the period of 2002–2010, and (3) RMSE between modelled and measured surface flow velocities. To investigate the robustness of the computed emergence velocity, we performed sensitivity tests by taking ±30% variations in ice thickness and sliding coefficient. We added the description addressed above as " Uncertainty of the emergence velocity is affected by factors such as accuracy of flow velocity measurement, interannual variability, and RMSE between modelled and measured flow velocity so that we performed sensitivity test by changing ±30 % of ice thickness and sliding coefficient." in Sect. 4.5.3. (L434-436)

*3) Some global estimates of mass gain and loss terms would be useful. For example, how much flux enters the numerical domain at the upstream boundary and how does that compare to the mass flux at the terminus (for lake calving), the integrated SMB, and the integrated dh/dt? This would provide a good assessment of the overall role of ice dynamics, without relying on details of emergence velocity calculations.*

**R#5_3:** Because the emergence velocity is estimated from a 2D model, if we calculate the flux budget, we calculate them with depth of the studied domain (no width / parallel shape). In this case, to avoid the confusion about unit, we convert them into changes in surface elevation by dividing with length of the domain. These values are not consistent with those observed/estimated independently. This is because the values discussed in the main text are calculated over the ablation "area" while those for this calculation are calculated over the simulated "central flowline". We afraid the different values make confusion. Contribution can be discussed by the independently derived values so that we did not add the suggested flux discussion (L498-501).

Some detailed comments (in order in which they occur):

*l.14: it is unclear why you mention supraglacial lakes, as the title is about proglacial lakes*

We changed. Please see the response R#4_8.

*l.22-24: I would rewrite this sentence as: The magnitude of dynamic thickening compensates for*

*more than half of the negative surface mass balance.*

The sentence is modified after a comment from another reviewer. Please see the response R#4_9.

*l.194: a linear temperature profile implies steady state, and not 'no heat storage'*

**R#5_4:** Here we do NOT assume "linear profile for expressing no heat storage" BUT assume "no heat storage", "linear temperature profile", and "melting point temperature" as parallel meanings. Because we believe that we correctly use the oxford comma here, we do not think that here should be changed.

*l.205/206: What are the ASTER images used for?*

**R#5_5:** We used 90-m resolution ASTER Level 3A1 data. Surface albedo is calculated using three-visible near-infrared sensors (VNIR; bands 1–3), and surface temperature is obtained from an average of five sensors in the thermal infrared (TIR; bands 10–14). We changed the description as: "Using eight ASTER images (90 m resolution, Level 3A1 data) obtained between October 2002 and October 2010 (Table S4), along with the NCEP/NCAR reanalysis climate data (NCEP-2, Kanamitsu et al., 2002), we calculated the distribution of mean thermal resistance on the two target glaciers. Surface albedo is calculated using three-visible near-infrared sensors (VNIR; bands 1–3), and surface temperature is obtained from an average of five sensors in the thermal infrared (TIR; bands 10–14)." (L215-219).

*eqn 5: I don't understand this equation. Is M_d only non-zero over debris? What is the motivation for the first term (t_D H_L/l_m)? The equation looks like discharge is counted separately from melt? What is the source for the accumulation data? And, finally, should the opening parenthesis be after the summation sign?*

**R#5_5:** Here is the simple mistake of K. Fujita who made mass balance calculation and writing the corresponding paragraphs. Thanks a lot for this comment. We will replace the present the equation (5) and explanation by the following paragraph. We have checked the model code too.

"where $t_D$ is the length of a day in seconds (86400 s), and $l_m$ is the latent heat of fusion of ice $(3.33 \times 105 \text{ J kg}^{-1})$. Annual mass balance of debris-covered part ($b$, m w.e. a$^{-1}$) is expressed as:

$$b = \sum_{D=1}^{365} \left( P_s + P_r + \frac{t_D H_L}{l_m}_{\ for\ debirs} + \frac{t_D H_L}{l_m}_{\ for\ snow} - D_d - D_s \right)/1000$$

here $P_s$ and $P_r$ are snow and rain, respectively, which are distinguished from precipitation depending on air temperature. Evaporation from debris and snow surfaces is expressed in the same formula, but they are calculated in different schemes because temperature and saturation conditions of the debris and snow surfaces are different. $D_d$ and $D_s$ are the daily discharge from the debris and snow surfaces, respectively. Discharge and evaporation from the snow surface was calculated only when snow layer was formed on the debris. Because snow layer does not exist at the end of melting season in the current climate condition and at the elevation of debris-covered area, snow accumulation ($P_s$) is compensated with evaporation and discharge from snow surface during a calculation year. Discharge from debris ($D_d$) is expressed as:

$$D_d = M_d + P_r + \frac{t_D H_L}{l_m}_{\ for\ debirs}$$

and then the mass balance can be simplified as:

$$b = -\sum_{D=1}^{365} M_d/1000$$

This implies that the mass balance of debris covered area is equivalent to the ice melting under the debris. Further details on the ... are described by Fujita and Sakai (2014)." (L234-252)

*l.256: delete: 'vertical component of the' (gravity only has a vertical component.*

**R#5_6:** We removed.

*l.273: Make it clear that the modeling domain is in the ablation area (this caused me some confusion at first, because I assumed it was the whole glacier).*

**R#5_7:** We modified the description as "Model domain was within 5100 m and 3500 m from the termini of Thorthormi and Lugge glaciers, respectively (white lines in Fig. 1b), including the ablation area and the lower accumulation area. We only interpret results in the ablation area (0–4200 and 700–2500 m from the termini of Thorthormi and Lugge glaciers, respectively), where surface

flow velocity was obtained from ASTER imagery. The lower accumulation area was included in the domain to supply ice into the studied area, thus excluded from analysis presentation of the results." (L297-301).

*l.284/85: Give some detail about this. Do you assume a parabolic velocity profile and assign it as a Dirichlet condition?*

**R#5_8:** The ice flux through the upper boundary was prescribed from the surface velocity from the satellite analysis. We assumed a velocity profile as 4th order of quadratic function without basal sliding. We clarified the description as ", through which the ice flux was prescribed from the surface velocity obtained by the satellite analysis. We assume no basal sliding and quadratic function (4th order) for the velocity profile from the surface to the bed." (L313-315).

*eqn (12): It's a bit strange to convert emergence velocity to water equivalent and then convert it back in the next equation. It would seem more logical to just state the ice balance b in ice equivalent.*

**R#5_9:** We change to calculate ice equivalent SMB (L344-353), and the equations (13) and (14) are simplified as:

$$v_e = v_z - v_h \tan \alpha \qquad (13)$$

and

$$\frac{dh}{dt} = b + v_e. \qquad (14)$$

*l.370: specify that this is not the mean SMB for the whole glacier (I think)*

**R#5_10:** We added "in the ablation area of" (L393).

*l.386: contrasted -> contrast*

Corrected. (L410)

*l.404: indicated -> indicates*

The sentence is modified after a comment from another reviewer. Please see the response R#4_40.

*l.465-468: it seems this could be more usefully displayed in a graph, otherwise there are too many numbers to make sense of.*

**R#5_11:** We deleted the description about thinning acceleration in Sect. 5.1. See R#5_1.

*l.546-548: I do not understand this sentence.*

**R#5_12:** We changed the description as "A numerical study suggested that lake water currents driven by valley wind over the lake surface could enhance thermal undercutting and then calving when a proglacial lake expands to a certain longitudinal length (Sakai et al., 2009)." (L546-548).

*l.555: a decrease in effective stress does not enhance shear stress, it reduces shear strength of sediments*

[revised manuscript text omitted]

---

## Author Response (AR3)

**Reply to referee comments**

We would like to thank two referees for thoughtful and useful comments. In the following, we describe our responses (in blue) point-by-point to each referee comment (*italic*). The revised manuscript was edited by Stallard Scientific, an English editing company in New zealand (https://www.stallardediting.com/). The manuscript with editing history is attached at the end of the reply, in which the light blue text denotes change by the authors while the light green text denotes the editing by the company.

**Reviewer #4**

The manuscript by Tsutaki et al. uses field- and satellite-derived measurements of a neighbouring lake- and land-terminating glacier in the Bhutan Himalaya to assess the impact of a proglacial lake on the thinning rates of each glacier. I very much enjoyed reading this version of the manuscript; it is much improved from the previous iteration, and is now a very neat study. The writing is generally good, but needs some correction to clearly communicate the content. I have provided examples of minor grammatical mistakes through the first two sections but this needs checking throughout; after this I only note specific instances where meaning is confused. I also have small queries about the discussion of the debris covers and a few other details. Otherwise, I consider it worthy of publication in The Cryosphere.

[Reply] Thanks a lot for the detailed comments and suggestions.

**Specific comments**

L21: Change "would be" to "is"
L24: "One-thirds" to "one-third"
Rev5: l.24 one-thirds -> one third
L39: Add "For example," before "Mass loss"
L41: Remove "the" before "more insight for"
Rev5: l.46 form -> formed
L46: "Suplaglacial" to "Supraglacial"
L72: Change to "unmanned autonomous vehicles (UAVs) are"
L73: "satellite" to "satellites," and "resolves" to "resolve" and "debris-covered

surfaces"

L76: "the repeated DGPS" to "repeat DGPS"

*Rev5: l.85/86: ... contrasting conditions at similar ... (delete 'the')*

L113: "show" to "shows" and "drain" to "drains"

*Rev5: l.114: drain -> drains*

*Rev5: 1.133: We neglected THE influence ...*

L144: Repeated word "change"; suggest remove second instance.

L157: Change to "...survey are so many (n = 3893) that the standard error could appear too small."

L170: Do you mean that the horizontal flow velocities are "at 60 m resolution"?

*Rev5: l.212: comma after resistance*

L218: "three-visible" to "three visible"

*Rev5: l.218: three-visible -> three visible*

Equations 4 & 5: "debirs" to "debris"?

*Rev5*: eqn 5: debirs -> debris

L243: Change to "a snow layer" ... and "climate condition nor at the elevation of the debris-covered area"

Rev5: 1.242: ...when A snow layer ...

L251: "debris covered" to "the debris-covered"

Rev5: 1.290: described -> fixed

*Rev5:* 1.300/301: *Reword this sentence. For example: ...into the studied area, and it therefore excluded from the analysis of the results.*

L297: Would read more clearly as: "Model domain was from 5100 m and 2500 m to the termini ... including the ablation and lower accumulation areas."

Rev5: 1.328: employed -> used

Rev5: 1.331: THE position ...

L369: I understand your comment in the author response, but to be clearer in the text, suggest you do change this to: "found in the lower elevations of the glacier"

*Rev5: l.436: test -> tests*

L447: "and greatly negative value" doesn't make sense; perhaps "becoming more negative"

L449: "within 4200 m of the terminus of Thorthormi"? Rev5: l.453: delete 'into' L483: Suggest changing estimated to modelled through this section, and being very clear which dh/dt is being discussed at any one time

L488: Suggest start new paragraph after "would be plausible"

Rev5: 1.490: ... the glacier BEING situated ...

L495: Remove "rather" and change "accelerates" to "accelerate"

L503: Doesn't this second sentence agree with the first? Suggest change "On the other hand" to something like "Furthermore"

L510: Proglacial rather than supraglacial lake?

*Rev5: l.510: supraglacial -> proglacial*

*Rev5: l.519: delete 'accelerated' (negative emergence velocities imply thinning, but it doesn't imply accelerated thinning)*

Rev5: 1.536: and THE glacier ...

L566: Just use DGPS instead of spelling out acronym

Rev5: 1.572: ... the MODELED SMB was ...

L830: "enotes" to "denotes"

L846: Do you mean over "off-glacier areas"? Also after (c), "terminus" should be "termini"

[Reply] All comments above (including those by reviewer #5 [*Rev5:*]) were corrected according to the reviewers' suggestions.

**L22 and throughout: "dynamically induced" to "dynamically-induced"**

[Reply] The English editing company suggested "Please note that compound adjectives that consist of an adverb ending in "ly" should not be hyphenated." so we did not change this.

**L55: Add "the" after "total ice thinning at"**

[Reply] In this manuscript we do not use "the" for proper name, which is here Yakutat "Glacier". The English editing company also suggested NOT to add "the" here so we did not change this.

L60 & 62: The phrases "beneficial to compute the ice flow velocity field" and "which require the ice flow velocity field" seem to contradict? Should the second be the surface velocity field? Or perhaps remove the first sentence of this paragraph.

**[Reply] We removed the first sentence.**

L85: I don't understand "grounding and fully contacting lake". Perhaps either remove, or explain that one is lake-terminating and the other not. [Reply] We deleted this because their terminal features are described above.

L133: I don't think you mean a change in debris thickness, just that the debris didn't affect the surveys because the layer was very thin. Suggest change to "We neglected the influence of debris thickness"

[Reply] This part was added through the previous revision by responding to a comment, in which the reviewer #4 wrote "how the debris cover varies". If we changed here to the suggested one, it would be more unclear what "the influence of debris thickness" is. So we removed the sentence.

L178: "Between November 2000" and when? Needs a date here. Also check consistency through manuscript – the Results section says delineation was between 2000-2017, but Figure 4 still says 2000-2012.

[Reply] We corrected the end year to 2011.

L314: I don't understand the second part of "We assume no basal sliding and quadratic function...". I assume you did apply a quadratic function? Perhaps change to "and applied a quadratic.."?

*Rev5: l.313/314: Perhaps you could split this sentence, and explain it a bit better. Upper surface can mean the surface of the glacier or the upstream boundary condition. The surface of the glacier has a stress free boundary condition. The upstream boundary has a Dirichlet condition with an assumed parabolic velocity and no sliding.*

[Reply] Yes, this was confusing. We corrected here as suggested by Reviewer #4.

L332: It is still not clear where this lake depth measurement was made, considering that Thorthormi wasn't lake-terminating at the time of this measurement. Could the position be marked on Figure 1? Or described here?

[Reply] We added the position in Fig. 1.

L374: The two vertical uncertainties aren't height uncertainties, but the uncertainty in the elevation change rate, I think? This should be noted ("Vertical elevation change rate uncertainties...")

[Reply] We corrected both as "uncertainty in the elevation change rate".

L390: As you report mass balance values for both types of surface, why not just have this section title as "Glacier mass balance" [Reply] We corrected this as "Surface mass balance".

L398 and throughout: I'm still not convinced by the description of the debris cover as "thin and sparse". I understand that a thick debris layer could reduce the SMB values by 2 m w.e. a-1. However, the images provided in Figure S1 actually show a continuous and very thin debris layer, which instead would greatly increase the melt rate. Where does this thin debris layer come into the debris-free/-covered categories for SMB – does it contribute to the high SMB for the debris-free areas? That would be wrong, in my opinion.

A "sparse" debris layer implies a discontinuous layer of debris (mostly clean ice), not what is shown in Figure S1. This thin debris layer is also the sort of debris layer I imagine from the methods, where the authors state that the debris cover was not thick enough to influence the surface elevations measured. Perhaps the thermal resistance results need reporting or showing in a figure to clear this up. If the debris layer is as shown in Figure S1, I suggest removing all instances of "sparse", and just referring to the majority of the debris cover as thin.

[Reply] We removed "sparse" or replaced it by "thin", and added the following figure showing distribution of thermal resistance in the supplement, and added the following description: "The distribution of SMBs is well consistent with that of thermal resistance (Fig. S10), the larger thermal resistance, suggesting the thicker debris, and then the more suppressed SMB."

*L416:* What does "within 16%" mean:  $\pm$  16% or  $\pm$  8% or that the calculated velocities were within 16% of the satellite-derived velocities? [Reply] This is  $\pm$ 16%. Corrected.

L429: I still have issue with the phrase "regardless of ice temperature assumptions" – the ice deformation is small because of the assumption that the ice is temperate, so you can't discard this assumption. I suggest you rephrase this either to follow the comment in the author response (that ice deformation is near the bed, so negligible) or simply that ice deformation plays a minor role in movement. [Reply] Corrected with the second option.

**L441: Mean uncertainty of what?**

[Reply] This was mistakenly embedded due to miscommunication between the first and second authors. This sentence was removed.

L480: Much greater than which values? The difference for Nepali glaciers is reported as 4x greater earlier in this paragraph, so not sure this sentence is valid? [Reply] We intended to compare with that by King et al. (2017). We corrected here to "similar to those previously reported in ...".

L484-5: I'm not sure what this sentence adds here? Nor do I understand the next sentence ("Although both SMB..."); could it be explained a little clearer?

L485: The argument would be better supported if the values were restated, rather than "Lugge < Thorthormi"; this makes a lot of work for reader. Write out the point in full and add important values in; same for instances below.

Rev 5: 1.486-488: Awkward sentence, rewrite.

[Reply] We changed the sentence (from L454) as "However, differences in  $\Delta z_s/\Delta t$  between the two glaciers are similar; i.e., Lugge is more negative by 3.27 m a-1 (observation) and 2.58 m a-1 (model) than Thorthormi.", removed the following one (L486), and merged with the next paragraph.

L506: I don't understand the start of this sentence as written – do you mean that the lake would have formed from this land-terminating condition if the model was run for longer?

*Rev5: l.506: I don't understand why the negative emergence velocity would have led to Lugge Glacier Lake being 'inevitably formed'?*

[Reply] We removed this sentence to avoid confusion.

L522: While some of this paragraph is necessary, I don't think it fits well here; the previous paragraph would run into the next section well. Perhaps move, or condense this paragraph to a sentence or two and include in previous paragraph.

[Reply] We shortened the paragraph into sentences, and include it in the previous one.

L541: Change "a less ice flux supplied" to "its smaller ice flux" Rev5: l.541: .. that less ice flux cannot counterbalance ... [Reply] We followed the reviewer #4's suggestion.

**Reviewer #5**

This is a second review of the manuscript by Tsutaki et al. The revision has much improved the presentation and it is now much clearer in what has been done in the paper. I find the material interesting and relevant. On the other hand I still have several comments that need addressing. I apologize that some of these comments are things I did not clearly point out in the first review. The paper could also use more editing for My main issue with the paper is that the proposed models for both SMB and ice flow will necessarily come with very large errors; potentially much larger than what the impression is from the error analysis. I will explain in more detail below. As such I propose that the paper be reworded a bit. Mostly it requires a better explanation of the purpose of the model, which should be given at the end of the introduction. The way I see it is that this paper provides solid data that Lugge and Thorthormi Glaciers have different thinning rates and also quite different dynamics. This is observationally well constrained. The modeling serves more as an idealized case of how the presence or absence of a lake can alter thinning rates. The answers from the authors to the previous reviews make it clear that the models' purpose is NOT an accurate represention of these glaciers; the necessary model complexity (2 vs 3D, for example) and input data is missing. I therefore suggest adding a paragraph at the end of the Introduction that explains that these models are there to illustrate the differences between a lake-calving and a land terminating terminus and that the model set up is meant to approximate the situation at Thorthormi and Lugge without making an attempt at accurate representation.

I think this would set a different tone for the paper. In particular, it would mean that the reader does not have to be worried about the very large differences in observed and modeled thinning rates.

*Here is why I think these models have larger errors than stated:*

1) SMB model: There are no observations that could be used for model validation, whether debris thickness or any measure of melt. As such, the model is entirely 'floating'. There is an error analysis in the paper, but it is very difficult to assess whether the reanalysis data works well for this purpose. The model serves well for the purpose of comparison between the two glaciers, because it is at least reasoable that the errors introduced from applying reanalysis data would apply to both situations.

2) Flow model: There are some validation data (surface velocities), but the model is severely under-constrained because of the lack of thickness data and the necessary restrictions from a 2D model in a valley glacier situation.

*l.88/89: Here I would add several sentences explaining the purpose of the modeling exercise (as explained above).*

[Reply] Thanks a lot for the constructive suggestions. We added the following sentence at the end of introduction as "However, due to lack of observational data for model validation, the models were only used to demonstrate the differences between lake- and land-terminating glaciers using the idealised case of how a proglacial lake can alter glacier thinning rates.".

*List of comments (in order that they occur):*

*l.69-71: I would split this into two sentences. Also, DEM differencing is routinely done, not just because the terrain is difficult to access*

[Reply] We deleted "because the surface ... large amount of data", and changed the start of next sentence from "In particular" to "Recently".

*l.156/158: Explain this a bit better: The issue is that using standard error assumes uncorrelated noise (which goes as 1 over square root n), while you assume systematic error. The truth is probably in between, where noise is correlated on some spatial scale (see e.g. Rolstad et al., 2009, J.Glac. or Motyka et al., 2010, J.Glac.)*

[Reply] Acknowledging to the comments, we added phrases "assuming uncorrected noise" for standard error, and "assuming systematic error" for standard deviation. One more sentence above was also added at the end of section. Thank you so much.

*l.204: I don't understand this assumption of a linear temperature profile. This assumes steady state. But a thin debris layer can never reach steady state when exposed to diurnal and seasonal boundary conditions. Also, what does the assumption of 'no heat storage' mean? Is it no 'change in heat storage'? The debris layer is at a certain temperature, that implies a certain amount of heat storage?*

[Reply] Surface temperature of the debris changes day by day (model time step is daily), but temperature profile is "linear" between surface temperature and the melting point (0 °C) at the debris-ice interface. We realized that "a" makes this confusion. "a linear temperature profile" is changed to "linear temperature profiles". Also changed to "no change in heat storage".

eqn 2: should the H\_L in the 'for debris' part be G\_d? Also: spelling of 'debirs'. Finally: use rho\_w instead of the number 1000 in the equation

[Reply] Latent heat (H\_L) is independent from heat flux into the debris (G\_d). Others were corrected.

*l.238/39: How is the solid/liquid determination done? Is it a step function at 0 deg C?* [Reply] Probability of solid/liquid precipitation is linearly changed between 0 (100% snow) and 4 °C (100% rain) (Fujita and Ageta, 2000). We added this sentence.

eqn 6: again, use rho\_w instead of 1000. Also, do you need to go through this derivation? Couldn't you simply write eqn (6) and be done with it? Where do you actually need runoff values?

[Reply] Water density was revised. In the model simulation, all components shown in Eq. 4 are calculated. But for the debris-covered ablation area, these components can be finally simplified into equation 6. Uncertainty in the SMB calculation was evaluated using melting amount instead of mass balance (otherwise we cannot express the uncertainty by percentage), so we believe that it is necessary to show that mass balance of debris-covered ablation area is equivalent to melting amount. Eq. 5 is required for this simplification from Eq. 4 to Eq. 6. We do not change here.

*l.296:* Assuming temperate because no information is available seems like a bad justification. Is there other supporting evidence? For example, the occurrence of melt high in the accumulation area, which would lead to annual warming of the firn through refreezing.

[Reply] We assumed the glaciers were temperate. This assumption was based on approximately 0 °C annual mean air temperature measured near the front of Lugge Glacier (Suzuki et al., 2007b). We added this information in the main text.

*l.304/305:* This requires more detail: The Farinotti method requires an 'apparent mass balance', which is SMB - dh/dt. What did you use here? The SMB calculated above and observed dh/dt? Also, there is an assumption about rheology, did you use a literature value for flow rate factor and no sliding? There is some circularity here, because the assumptions in the derivation of the ice thickness distribution affect the calibration of the slipperiness used to match surface velocities. This is one reason that the model results have to be treated with caution when applied to the glaciers, although the 'lake -

**no lake' comparison is still valid.**

*l.319/320: Are those values of C consistent with the thickness inversion (see my comment above)?*

[Reply] Treatment of sliding is different in the ice thickness and ice flow models, i.e. sliding is implicitly included in a correction factor in the Farinotti's model, whereas a stress dependent sliding low was used in the flow model. It is hard to adjust the sliding conditions in the two completely different models. Nevertheless, both models attribute a certain portion of ice motion to sliding. We believe the influence of this detail is insignificant.

We recalculated all the simulations because the rate factor (A) was inconsistent in the ice thickness and flow models. We corrected the text and replaced the figures after running the models with the same rate factor.

**1.311/312: Did you do any convergence tests under element refinement?**

[Reply] We tested finer mesh resolution (1224 elements) for the Thorthormi Glacier model. This test confirmed that the difference in computed velocities was within 4%. We addressed this additional experiment.

*l.325/326:* Do you need to prescribe 'zero horizontal velocity'? Does this not come naturally at the land terminating boundary, due to ice thickness going to zero? [Reply] The text is corrected to avoid the confusion.

*l.339: Same as earlier comment, do you need to prescribe this?* [Reply] Corrected as above.

eqn 14: dh/dt should really be partial derivatives, otherwise this equation is not correct. Actually, I don't like the use of dh/dt anywhere in the manuscript (and in many other manuscripts as well). When you measure surface differences you measure Delta h, or actually really Delta z\_surf (since you don't know anything that could happen at the base). If you want to put this as a rate it would be far better to write Delta h / Delta t (or Delta z\_surf / Delta t). This clearly indicates that this is a measurement over a certain finite time span (you never directly measure a rate) and it avoids the issue that partial and total derivatives are not identical. Usually, this is clear from context, but why not be accurate? Also, in the equation it bothers me, because it's technically wrong. [Reply] We replaced all dh/dt by  $\Delta z_s/\Delta t$  and added description at the first appearance as "which is usually expressed as dh/dt in other previous studies".

*l.354-357:* The result in Truffer et al. (2009) that emergence velocities appear to be proportional to horizontal velocities are not stated in a universal way. That seems to be an observation at the terminus of the Taku Glacier. I do not believe this can be used as justification for assuming that errors in emergence velocities are proportional to errors in horizontal velocities. The vertical velocities are calculated from eqn (9), which involves derivatives of horizontal velocities. Because derivatives amplify noise, they can be large. Furthermore, the equation then needs to be integrated over the ice thickness, which also has large errors. That's why I believe the model might have much larger errors than stated here. Again, that's a problem if you claim to accurately model Lugge and Thorthormi Glaciers. I don't think it's a problem for comparing the two situations and for doing that 'lake - no lake' comparisons.

[Reply] We removed the uncertainty estimation for emergence velocity, and confirmed that this removal does not affect the following discussion because we also removed the description for this uncertainty from the section 4.5.3 (see reply to 1.431/432 addressed below).

*l.374/375: rewrite this sentence* [Reply] We rewrote it.

*l.391:* As stated earlier, I have a hard time believing this error estimate. In fact, this is almost at the level of measurement uncertainty if you had a small stake network.[Reply] These are not uncertainty but spatial variability of SMB. We added one sentence to address it as "The errors in SMBs are of spatial variability over the calculated domains".

*l.431/432:* I think the RMSE between modelled and measured is meaningless, given some of my earlier comments. The model cannot be an accurate representation of these glaciers, so one should not worry too much about matching velocities. Furthermore, the velocities were used for model calibration, so it makes no sense to also use them for

**validation.**

[Reply] We removed the description for the uncertainty of emergence velocity from the main text, but we remained the description for the uncertainty and sensitivity tests of surface flow velocity.

*l.493-496:* You first say that the flow is surface parallel and then that the emergence velocity is negative, this seems contradictory.[Reply] Corrected to avoid the contradiction.

**Contrasting thinning patterns between lake- and land-terminating glaciers in the Bhutan Himalaya**

Shun Tsutaki1,a, Koji Fujita1, Takayuki Nuimura1,b, Akiko Sakai1, Shin Sugiyama2, Jiro Komori1,3,c, and Phuntsho Tshering1,3,d

[revised manuscript text omitted]
  $\Delta z_s / \Delta t$  between Lugge and Thorthormi glaciers derived from the DGPS-DEMs (3.3 times) is similar to those previously reported in the Nepal Himalaya, suggesting that ice dynamics play a more significant role here.

**5.2 Influence of ice dynamics on glacier thinning**

The modelled  $\Delta z_s / \Delta t$  values are 63 % more negative than the DGPS observations for Thorthormi Glacier and 79 % more negative than the DGPS observations for Lugge Glacier (Table 1). However, the differences in  $\Delta z_s / \Delta t$  between the two glaciers are similar; as Lugge Glacier is only 3.27 (observation) and 6.08 m a-1 (model) more negative than Thorthormi Glacier. The mean SMB of Thorthormi Glacier is 40 % more negative than that of Lugge Glacier. Since there is only a thin debris mantle across the ablation areas of both glaciers (Fig. S1), the more negative SMB of Thorthormi Glacier could be explained by the glacier being situated at lower elevations (Fig. 2b). The modelled SMBs (Thorthormi < Lugge) and observed  $\Delta z_s / \Delta t$  values (Lugge < Thorthormi) suggest that the glacier dynamics of these two glaciers are substantially different. The horizontal flow velocities of Lugge Glacier are nearly uniform along the central flowline (Fig. 5d), and the computed emergence velocity is negative ( $-0.83 \pm 0.30$  m a-1), which means the ice dynamics accelerate glacier thinning. Conversely, the flow velocities of Thorthormi Glacier decrease toward the terminus (Fig. 5c), resulting in thickening under a longitudinally compressive flow regime. The emergence velocity of Thorthormi Glacier is positive  $(6.89 \pm 0.34 \text{ m s}^{-1})$ , indicating a vertically extending strain regime. The calculated  $\Delta z_s/\Delta t$  of Thorthormi Glacier is equivalent to 28 % of the negative SMB, implying that two-third of the surface ablation is counterbalanced by ice dynamics. In other words, dynamically induced ice thickening partly compensates the negative SMB.

Experiment 1 demonstrates that the difference in emergence velocity between land- and lake-terminating glaciers leads to contrasting thinning patterns. Furthermore, Experiment 2 demonstrates that the emergence velocity was less negative  $(-0.09 \pm 0.30 \text{ m a}^{-1})$  in the absence of a proglacial lake at the front of Lugge Glacier, resulting in a decrease in the thinning rate by 9 % compared to the lake-terminating condition. For Thorthormi Glacier, the emergence velocity under the lake-terminating condition is negative  $(-2.38 \pm 0.77 \text{ m a}^{-1})$ , resulting in a 3.5 times greater thinning rate  $(2.28 \text{ to } 8.02 \text{ m a}^{-1})$ . Our ice flow modelling demonstrates that thinning will accelerate with the development of a proglacial lake at the front of Thorthormi Glacier.

Contrasting patterns of glacier thinning and horizontal flow velocities between land- and lake-terminating glaciers are consistent with satellite-based observations over lake- or ocean-terminating glaciers and neighbouring land-terminating glaciers in the Nepal Himalaya (King et al., 2017) and Greenland (Tsutaki et al., 2016). A decrease in the down-glacier flow velocities over the lower reaches of land-terminating glaciers suggests a longitudinally compressive flow regime,

which would result in a positive emergence velocity and therefore thickening to compensate for the negative SMB. Conversely, for lake-terminating glaciers, an increase in the down-glacier flow velocities suggests a longitudinally tensile flow regime, which would yield a negative emergence velocity, resulting in ice thinning. The contrasting flow regimes modelled in this study suggest that the mechanisms would not only be applicable to Thorthormi and Lugge glaciers, but also to other lake- and land-terminating glaciers worldwide where contrasting thinning patterns are observed. The modelled thinning rates are more negative than the observed rates for both glaciers (Fig. 7b), probably 
[revised manuscript text omitted]
 \ ({\rm m \ a^{-1}})$ | DGPS                  | $-1.40 \pm 0.27$ | $-4.67 \pm 0.27$ |
|                                            | ASTER                 | $-1.61 \pm 2.75$ | $-2.24 \pm 2.75$ |
| SMB (m w.e. $a^{-1}$ )                     | Ablation area         | $-7.36 \pm 0.12$ | $-5.25 \pm 0.13$ |
|                                            | Debris-covered area   | $-7.30 \pm 0.13$ | $-5.41 \pm 0.18$ |
|                                            | Debris-free area      | $-9.31\pm0.68$   | $-7.33 \pm 0.41$ |
| Exp. 1 (m a -1 )                | b ie       | $-8.09 \pm 0.13$ | $-5.77 \pm 0.14$ |
|                                            | Ve                    | $+6.89 \pm 0.34$ | $-0.83 \pm 0.30$ |
|                                            | $\Delta z_s/\Delta t$ | $-2.28 \pm 0.66$ | $-8.36 \pm 0.73$ |
| Exp. 2 (m $a^{-1}$ )                       | b ie       | $-8.09 \pm 0.13$ | $-5.77 \pm 0.14$ |
|                                            | v e        | $-2.38 \pm 0.77$ | $-0.09 \pm 0.30$ |
|                                            | $\Delta z_s/\Delta t$ | $-8.02 \pm 1.10$ | $-7.63 \pm 0.73$ |

---

## Author Response (AR4)

**Editor's comment**

*First of all, apologies for my delay in handling this manuscript.*
*I have now received two reviews of your re-revised manuscript. Both reviewers find the paper improved, and I agree with their evaluation. Inclusion of the idealised case (as suggested by reviewers) has also benefited the paper.*

*The first reviewer, who is not a modeller, finds the paper improved and recommends minor revision. The second reviewer, however, raises some major issues regarding the errors in both the surface mass balance and ice flow model, and I agree with his main three points. These are major issues that jeopardise the quality of your paper.*

*I therefore recommend major revisions.*

*Since this is the third iteration in a process that has taken a long time, I suggest that you make a real effort to address those points. I would be happy to discuss with you the needed revisions, and I am sure that the reviewer, Martin Truffer, who has decided from the beginning not to be anonymous, would also be happy to discuss this with you.*

*I really think this is a nice paper that would be worth publishing, but the main issues related to errors should be addressed in a satisfactory manner.*
*Please get back to me and/or the reviewer if we can help you with the revision, so that the paper will get to a prompt conclusion soon.*

*Non-public comments to the Author:*
*Dear Koji and co-authors,*

*This is an addition for you only. Again, apologies for my delay in handling this manuscript.*

*Please see above for my recommendation, which is a major revision again, based on the comments of the second reviewer, who did a very thorough job and is an expert on the flow modeling and the main topics of the paper. His three main issues are all important and you need to address them properly.*

*Since this is the third iteration in a process that has taken a long time (partly because of my absences, and partly because not all issues were always satisfactorily addressed), I suggest that you make a real effort to address those points. I would be happy to discuss with you the needed revisions, and I am sure that the reviewer, Martin Truffer, who has decided from the beginning not to be anonymous, would also be happy to discuss this with you.*

*If the revision is not satisfactory this time, my suggestion will be to reject the paper and encourage a new resubmission here or elsewhere.*

*You might remember that I had sent a separate email to you after the last revision asking to improve the way you responded to some of the issues raised by the reviewers.*

*I really think this is a nice paper that would be worth publishing, but the main issues related to errors should be addressed in a satisfactory manner.*

*Please get back to me and/or the reviewer if we can help you with the revision, so that the paper will get to a prompt conclusion soon.*

**Reply to referee comments**

We would like to thank two referees for thoughtful and useful comments. In the following, we describe our responses (in blue) point-by-point to each referee comment (*italic*). The revised manuscript was edited by Stallard Scientific, an English editing company in New Zealand (https://www.stallardediting.com/).

**Reviewer #4**

*This is the third review of Tsutaki et al.'s manuscript. The manuscript is much improved again. As I am not a modeller, I have been largely unable to comment on the modelling section, however, I think the changes made based on the suggestions of Reviewer 5 to present the modelling as an idealised case has helped mitigate issues with the large uncertainties I had previously pointed out. I do think the idealised case should be restated in the conclusions section. Otherwise I have only a few minor comments, which I list below.*
[Reply] Thanks a lot for the detailed comments and suggestions.

Specific comments:

*L16: change "glacier" to "glaciers"*
*L38: "monsoonal" rather than "monsoon-influenced"?*
*L55 and thereon: no space between a number and % sign*
*L80: suggest combining these sentences – "contrasting termini at similar elevations, which makes them…"*
*L131 and thereon: no dash between a number and m (denoting metres) – this has become inconsistent through the text and figure captions*
*L172: remove "the areal" (repetition of area)*
*L204-5: shortwave and longwave should be one word*
*L219 and thereon: no space between a number and degree sign*
*L230: add "the" before "debris-covered"*
*L240: add "the" before "melting"*
*L304: "smoothing the elevations" rather than "filtering the elevations with a smoothing routine"?*
*L315: remove "and" before "several", and "numbers" should be "number"*
*L404: "reversed" instead of "prescribed" for consistency with earlier in manuscript*
*L415: remove "within"*
*L430: add "to be" after "considered"*
*L540: hyphenate "water-saturated"*
[Reply] All comments above were corrected according to the reviewers' suggestions.

*L295: I still have issue with: i) the assumption of temperate conditions, and ii) the apparent (and later contradictory) assumption of no basal sliding in the modelling (L319). First, I don't see how the air temperature over a lake determines the temperature of a glacier? Lakes obviously have a temperature > 0°C unless they are frozen through (which still wouldn't represent the glacier's thermal regime), and can additionally alter the local microclimate (Carrivick and Tweed, 2013), and I therefore don't think this is a valid assumption. Second, if you do assume temperate conditions, there WILL be (at least some) basal sliding – which your results show (L427). The clause "We assumed no basal sliding" (L319) is thus unclear (at least to a non-modeller) and needs to be altered.*

[Reply] We calculate ice thickness of both glaciers to reproduce observed surface velocity distribution with assuming a frozen bed (no basal sliding and warm and soft ice), and then obtain 800 and 300 m for Thorthormi and Lugge glaciers, respectively. These are much greater than the observational lake depth (~100 m). In particular, with almost flat surface and fast flow, Thorthormi would have an extremely thick ice. We briefly address this in the text.

For issue ii), we removed the sentence "We assumed no basal sliding, and" and changed here as "We applied a fourth-order..." to be clear the fact that we applied basal sliding in the model.

*L380: is this flow uncertainty ± 12.1 or ± 6.05 m/a?*
[Reply] Uncertainty is estimated to be ±12.1 m a$^{-1}$. We added plus-minus sign before 12.1 in the revised manuscript.

*L475-6: if the difference between these glaciers and others in Nepal are similar, how can the role of ice dynamics be different?*
[Reply] We deleted this sentence because a point of discussion is unclear here.

*L505: I'd be careful with this use of "thickening", as the overall thinning rate is still negative – this clause implies to me that the thickening is greater than the negative SMB. This is expressed better in the final sentence of this paragraph, so I would suggest either wording more carefully (akin to L512) or just removing this final clause after "emergence velocity". Similar for L23 – perhaps change "suppressed" to "minimised"?*
[Reply] We deleted this sentence "and therefore thickening to compensate for the negative SMB" because this is described more clearly in the latter part of this section as reviewer suggested. We changed "suppressed" to "minimized" in the abstract.

*Carrivick, J. L. and Tweed, F. S.: Proglacial Lakes: Character, behaviour and geological importance, Quat. Sci. Rev., 78, 34–52, doi:10.1016/j.quascirev.2013.07.028, 2013.*

**Reviewer #5**

*This paper has improved a lot since its last iteration, particularly in grammar and language, which helps a lot with understanding. As I've stated before, the topic of the paper is interesting, the findings are significant, and I do hope that this will be published.*

*Unfortunately I still think that there are some issues that need to be addressed, and these issues revolve around the treatment of errors in both SMB and the flow model. I am not certain what the correct way is to address these errors, but they way it is done here is incorrect.*

*For the SMB the authors use the standard deviation of the SMB over the modelled area as an estimate of error. This is clearly incorrect, the spatial variability of modelled SMB is entirely unrelated to the error. A better way to estimate error would be if there are any independent estimates to compare to (even over short times), but this might not exist? Or a comparison of integrated SMB to geodetic balance for a land terminating glacier.*

[Reply] We agree with this comment. However, there is no data to compare with our SMB estimate. Although we have conducted DGPS surveys at a land-terminating glacier (Lugge 2) for the studied period, the domain is limited to the debris-covered ablation zone where the surface lowering is also affected by glacier dynamics. We will encounter the same problem argued by reviewers because of no thickness data. If this "geodetic balance" means "glacier-wide mass balance", on the other hand, we can compare our SMB estimate with some remotely sensed mass balance. However, we concern that large uncertainty for the accumulation zone in both remote sensing analysis and model estimate would not support justification of the model. Therefore, we will simply replace the standard deviation (spatial variability of SMB) by the error estimated by changing related parameters (Figs. S12 and S13). Because SMB of the debris-covered ablation zone is equal to melt amount with negative sign, we will show spatial averages of uncertainties (Fig. S13) as SMB errors (±2.92 m w.e. for Thorthormi and ±2.41 m w.e. for Lugge glaciers).

*For the flow model, the stated error is clearly not correct, because measurements of velocity and simulated thickness change are outside the error range. There are a number of issues with modeled ice velocities that have yet to be addressed:*

*1) The issue of 'apparent mass balance' has not been addressed in the revision. This quantity is the SMB-dh/dt in the Farinotti paper. A negative thickness change contributes to mass flux and works opposite the SMB. In this case thickness changes are almost of the same magnitude as SMB, so the 'apparent mass balance' is much closer to zero, which would have an important effect on the calculated ice thickness.*

[Reply] In the former version of the revised manuscript, we calculated apparent mass balance by the simulated SMB and *dh/dt* observed by ASTER DEMs. However, we described in the manuscript as "with … and the above-mentioned SMB model (Sect. 3.4)", leading to misunderstanding for reviewer that we did not calculate apparent mass balance. In the revised manuscript, we changed a description to "above-mentioned SMB model (Sect. 3.4) and satellite-based ice thickness change (Sect. 3.1)".

*2) The calculated emergence velocities show some clear errors, for example, the large positive values in Fig. 5b. If those were correct, the glacier would be thickening in those places at fast rates; this is not observed. I think that errors in emergence velocities are simply too large to say something meaningful about simulated thickness changes. It would make more sense to calculate emergence velocities from observed thickness change and modelled SMB.*

[Reply] We agree that errors in emergence velocities are still too large to discuss simulated thickness changes for experiments 1 and 2. As reviewer pointed out in the latter comment, comparing the results between experiments 1 and 2 is important for our conclusion. We addressed spatial variation of sliding coefficients in the flow model to reproduce observed velocities more accurately. Sliding coefficients were determined by minimizing of RMSE between modelled and measured surface flow velocities over the area within 4100 m and 500-1900 m of the termini of Thorthormi and Lugge glaciers, respectively. Obtained distribution of the sliding coefficient (*C*) is shown in Figure S5. Based on these new results, emergence velocities were re-calculated for experiments 1 and 2. We added Figure 8 to compare longitudinal distribution of emergence velocities calculated from experiments 1 and 2.

*3) Both Thorthormi and Lugge Glaciers show step changes in observed velocites that are not at all reproduced in the model.*
[Reply] There is one possibility that step changes in observed surface velocities are due to miscorrelation in feature tracking process caused by surface ogives along the center of both glaciers. In the revised manuscript, we simulated emergence velocities over the area within 4100 m of the terminus of Thorthormi and within 500-1900 m of the terminus of Lugge Glacier, where step changes in velocities were not observed. We added the description "Measured surface velocities show step changes at 800–1200 m and 1900–2000 m from the termini of Thorthormi and Lugge glaciers, respectively (Fig. 3). It is likely that these step changes are due to miscorrelation in feature tracking process caused by surface ogives along the centre of the glaciers. Consequently, we only interpret the simulated velocities within 500–1900 m of the terminus of Lugge Glacier. For Thorthormi Glacier, we considered a mean value of observed velocities at the area as a reference value of simulation." in Sect. 4.5.3.

*These inadequacies are perhaps not such a big surprise. There are many things that the model does not consider, such as the influence of lateral stresses or spatial variation of sliding coefficients. The latter could be addressed, but in some sense the goal here is not to reproduce velocities. Rather, the authors should stress the changes that occur between experiment 1 and 2.*
[Reply] We removed simulated thickness changes in the revised manuscript because those uncertainties are still too large to compare with observed thickness changes. We addressed spatial variation of sliding coefficients in the flow model as described above reply.

Smaller comments:

The thinning rate is sometimes stated as a positive number (l.114/115) and sometimes as a negative number (l.17/18). It is generally clear from the context what is meant, but you should at least be consistent. My preference is to always use thickness change, rather than thinning rate.
[Reply] We changed "thinning rate" to "thickness change" throughout the manuscript.

My previous comment about partial derivatives was slightly misunderstood: In eqn 14, you should use partial derivatives for h; this is an equation that is valid at each point in time. Otherwise, I like the changes, that is, measurements are now all refered to by Delta z/ Delta t. I think that's the most accurate representation. Basically what you do is to take a measurement Delta z/ Delta t as an estimate for the quantity partial h / partial t.
[Reply] We agreed using Delta z/ Delta t throughout the manuscript.

These comments are all addressable and I do hope that the paper can be published. I also apologize for the delay with the review; it came at a very busy time.
[Reply] Thanks a lot for your careful review for many times. Our manuscript is now much improved from the original one.

[revised manuscript text omitted]

---

## Author Response (AR5)

**Reply to referee comments**

We are very happy to hear that our revised manuscript was accepted for publication in The Cryosphere. We would like to thank the referee for review of the manuscript and suggestion of editorial corrections. All editorial corrections were corrected according to the reviewers' suggestions.

Line 535: We changed acknowledgement for support as "(grant numbers 17H06104 and 18K18176)".

[revised manuscript text omitted]

**Figure 1:** Glaciers and glacial lakes in the Lunana region, Bhutan Himalaya, superimposed with (a) rate of elevation change ($\Delta z_s/\Delta t$) for
the 2004–2011 period derived from DGPS-DEMs, (b) surface flow velocities (arrows) with magnitude (colour scale) between 30 January
2007 and 1 January 2008, and (c) simulated surface mass balance (SMB) for the 1979–2017 period. Inset map in (a) shows the location of
the study site. The $\Delta z_s/\Delta t$ in (a) is depicted on a 50 m grid, which is averaged from the differentiated 1 m DEMs. Note that bathymetry of
Thorthormi Lake was measured at a limited point due to icebergs (red cross). Light blue hatches indicate glacial lakes in December 2009
(Ukita et al., 2011; Nagai et al., 2017). Background image is of ALOS PRISM scene on 2 December 2009. White lines in (b) indicate the
central flowline of each glacier.

[Figure]

**Figure 2:** (a) Histogram of elevation differences over off-glacier area at 0.5 m elevation bins. The rate of elevation change for Thorthormi
(blue) and Lugge (red) glaciers is compared with (b) elevation in 2011, and (c) distance from the glacier termini in 2002 along the central
flowlines (Fig. 1b). The red dashed line in (c) denotes the location of the calving front of Lugge Glacier in 2011.

[Figure]

**Figure 3:** Surface flow velocities along the central flowlines of (a) Thorthormi and (b) Lugge glaciers for the 2002–2010 study period. The
black lines are the mean flow velocities from 2002 to 2010, with the shaded grey regions denoting the standard deviation. The distance from
each respective 2002 glacier terminus is indicated on the horizontal axis.

[Figure]

**Figure 4:** Glacial lake boundaries in (a) Thorthormi and (b) Lugge glaciers from 2000 to 2011, and (c) cumulative lake area changes of the
glaciers since 17 November 2000. The background image is an ALOS PRISM image acquired on 2 December 2009.

[Figure]

**Figure 5:** Ice flow simulations in longitudinal cross sections of Thorthormi (left panels) and Lugge (right panels) glaciers, with the present geometries of the glaciers employed in the models. (a and b) Finite element meshes used for the simulations, with red markers indicating the bedrock elevation based on a bathymetric survey. The light blue shading in (b) indicates Lugge Glacial Lake. Simulated (c and d) two-dimensional flow vectors (magnitude and direction) and (e and f) horizontal components of the flow velocity. The blue and black curves are the simulated surface ($u_s$) and basal velocities ($u_b$), respectively. The red curves are the observed surface flow velocities for 2002–2010.

[Figure]

**Figure 6:** Ice flow simulations in longitudinal cross sections of Thorthormi Glacier under the lake-terminating condition (left panels), and
Lugge Glacier under the land-terminating condition (right panels). (a and b) Finite element meshes used for the simulation. The light blue
shading in (a) indicates the proglacial lake in front of Thorthormi Glacier. Simulated (c and d) two-dimensional flow vectors (magnitude
and direction) and (e and f) horizontal components of the flow velocity. The blue and black curves are the simulated surface ($u_s$) and basal
velocities ($u_b$), respectively. The red curves are the observed surface flow velocities for 2002–2010.

[Figure]

**Figure 7:** Rate of elevation change ($\Delta z_s / \Delta t$), from survey and ASTER-DEMs during 2004–2011, simulated surface mass balance (SMB),
emergence velocity ($v_e$) calculations along the central flowlines of (a) Thorthormi and (b) Lugge glaciers. Shaded regions denote the
simulated SMB uncertainties.

[Figure]

**Figure 8:** Calculated emergence velocity ($v_e$) for experiment 1 and 2 along the central flowlines of (a) Thorthormi and (b) Lugge glaciers.